# Regional and institutional trends in assessment for academic promotion

B. H. Lim[1], C. D'Ippoliti[2,3], M. Dominik[4], A. C. Hernández-Mondragón[5], K. Vermeir[6,7,8], K. K. Chong[1], H. Hussein[9], V. S. Morales-Salgado[10], K. J. Cloete[11,12], J. N. Kimengsi[13,14], L. Balboa[15], S. Mondello[16], T. E. dela Cruz[17], S. Lopez-Verges[18,19], I. Sidi Zakari[20], A. Simonyan[21], I. Palomo[22], A. Režek Jambrak[23], J. Germo Nzweundji[24], A. Molnar[25,26], A. M. I. Saktiawati[27], S. Elagroudy[28], P. Kumar[29], S. Enany[30], V. Narita[31], M. Backes[32,33], V. Siciliano[34], D. Egamberdieva[35,36] & Y. Flores Bueso[37,38 ✉]

The assessment of research performance is widely seen as a vital tool in upholding the highest standards of quality, with selection and competition believed to drive progress. Academic institutions need to take critical decisions on hiring and promotion, while facing external pressure by also being subject to research assessment[1–4]. Here we present an outlook on research assessment for career progression with specific focus on promotion to full professorship, based on 314 policies from 190 academic institutions and 218 policies from 58 government agencies, covering 32 countries in the Global North and 89 countries in the Global South. We investigated how frequently various promotion criteria are mentioned and carried out a statistical analysis to infer commonalities and differences across policies. Although quantitative methods of assessment remain popular, in agreement with what is found in more geographically restricted studies[5–9], they are not omnipresent. We find differences between the Global North and the Global South as well as between institutional and national policies, but less so between disciplines. A preference for bibliometric indicators is more marked in upper-middle-income countries. Although we see some variation, many promotion policies are based on the assumption of specific career paths that become normative rather than embracing diversity. In turn, this restricts opportunities for researchers. These results challenge current practice and have strategic implications for researchers, research managers and national governments.

The pervasiveness of evaluation and the obsession with metrics in modern society often come at the cost of sensible judgement[10], and academia is no exception[11]. Performance assessment is widely regarded as essential for upholding high standards, and selective processes and competition are believed to drive progress. However, performance indicators can become ends in themselves, and assessments lose effectiveness when misaligned with their original purpose[7,12]. Moreover, one may question whether competition as a core value suits a global research ecosystem that thrives on diversity and depends on collaboration for impact[13]. If our goal is to advance society through knowledge generation, we need to understand how research assessment, from the global level to individual researchers, can contribute positively to the research ecosystem.

Claiming the promotion of research 'excellence' and priding oneself in the record of 'excellence' has become commonplace, but what this excellence is concretely about is unclear[14,15]. It might not be problematic if 'excellence' varies across contexts. However, increased marketization subjects research institutions to competitive pressure[16], under which research managers face the challenge of building efficient teams delivering long-term value while maintaining external recognition linked to financial support, which is frequently related to flawed university rankings[3,4]. These management decisions affect researchers at all career stages, especially in recruitment, evaluations, retention and promotion.

The widespread use of scientometrics, particularly bibliometrics[1], has fostered the perception of a universal research assessment system, but such a view fails to recognize the complexity and diversity of actual practices. Metrics are attractive owing to their simplicity, low cost and perceived objectivity, which is thought to mitigate favouritism[17,18]. However, citation-based bibliometrics reflect social networks and accumulate subjective decisions[19]. Rather than objectivity, transparency is key for maintaining integrity in assessments, which inevitably involve human judgement. Research outputs offer important evidence of progress but present only a narrow view of the broader research ecosystem. Evaluating them solely by productivity and popularity fails to capture both value and rigour. In the Leiden Manifesto, experts on scientometrics raised concerns that evaluation has increasingly become led by data rather than by judgement and warned of the misapplication of indicators, with the journal impact factor being a prime example[20,21]. The SCOPE guide for research evaluation stresses that performance should be measured against the mission goals of institutions, groups or individuals, respecting relevant contexts[22]. Assessment processes that do not meet their purpose are invalid.

The Hong Kong Principles for assessing researchers emerged from acknowledging the need for trustworthiness of knowledge[23], emphasizing the importance of recognizing behaviours that promote research integrity. However, the pursuit of 'excellence' through quantitative metrics often drives unethical behaviour[24,25]. The United Nations Educational, Scientific and Cultural Organization (UNESCO) Recommendation on Open Science highlights that fostering a global research culture aligned with open science requires adequate evaluation processes that reward good practices[26], as previously stressed, for example, by the Global Young Academy and the European University Association[27,28].

Change is on the horizon as more signatories to the San Francisco Declaration on Research Assessment[29] update their procedures, the Latin American Council of Social Sciences (CLACSO) launches the Latin American Forum for Research Assessment (FOLEC)[30], and collaborations such as that between the InterAcademy Partnership, the Global Young Academy and the International Science Council publish studies on research evaluation and plan joint future initiatives[31]. All the while a large community originally gathered by the European Commission builds a Coalition for Advancing Research Assessment[32,33]. Wide discontent with existing practices has also led to discourse at national levels about ways forwards with better approaches[34,35].

Our analysis of promotion practices across the world aims at overcoming some biased perceptions and illuminating the connection with competencies, skills, productivity, impact and benefits. We anticipate that our findings will prove valuable to both researchers and research managers for understanding career options and opportunities and offer guidance on how to build a robust and diverse research ecosystem driven by responsible actors contributing through their strengths.

## Study design

To study how researchers are evaluated worldwide, we conducted a cross-sectional analysis examining the assessment criteria used in promotion policies. We systematically identified and analysed selected promotion criteria by capturing their presence or absence in promotion policies and comparing differences and similarities across disciplines, fields, tracks, types of institution and countries, considering their socioeconomic contexts. Rather than following a predefined protocol, we developed our methodology through an initial pilot study, which then evolved into a comprehensive framework (Supplementary Information section 1.1), which allowed us to compare and quantify qualitative data from institutional documents worldwide.

Researchers operate in diverse environments across countries, regions and institutions, resulting in substantial variability in career progression roles. Mapping all career paths is beyond our aims, so we focused on promotion policies for (full) 'professor'—the most senior academic role, widely recognized and comparable across countries. We distinguish the standard academic track, a research-focused track, a teaching-focused track and a clinical track. Additionally, we focus on assessment within academic institutions, and do not include roles beyond universities such as those in research institutes, clinical settings or commercial environments.

Between May 2016 and November 2023, we drew on the Global Young Academy membership and alumni network to collect documents outlining promotion policies, including criteria and procedures (Methods section "Data acquisition"). In a study pilot phase (2016–2018), we sourced 46 policies to inform our methodological framework. Subsequently, in a first data collection phase (2018–2021), we built a dataset of 196 policies from 55 countries. In 2023, we updated the data for all policies collected and broadened the scope of our effort to include policies from under-represented Global South regions, adding a further 440 policies. With this last round, we sourced policies set by 190 academic institutions ('institutional policies') and 58 government agencies ('national policies') across 121 countries. Additionally,

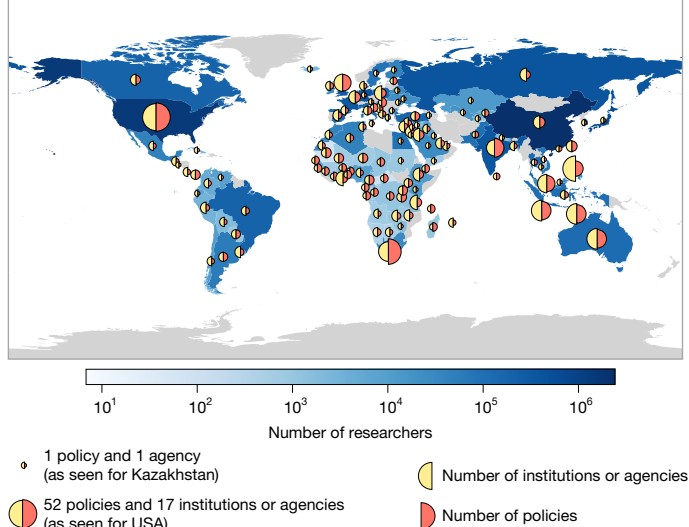

**Fig. 1 | Sample of promotion policies.** Map showing the geographical distribution of the data used for analysis. Blue colour shades (on logarithmic scale) indicate the number of estimated active researchers in each of the 121 countries from which we sourced policies[36–42]. Other countries and territories are shown in grey. The area of the semicircles is proportional to, respectively, the number of institutions or agencies (yellow) and the number of policies (orange) from a given country. The map is based on geodata openly available at the Natural Earth repository, using the 50-m land polygon GeoJSON dataset (https://www.naturalearthdata.com/). The plot was generated with geopandas and matplotlib wedge shape patch, as per the annotated code shared in the Code availability section. The authors do not endorse any position over any disputed area or contested border. The number of researchers was obtained from the UNESCO Institute for Statistics and UNESCO Science Reports (Supplementary Information section 1.7).

53 organizations' (institutions' or agencies') documents in our sample defined multiple policies that distinguish between career tracks and/or disciplines, which brought the total of collected track-specific and discipline-specific policies to 532 (Fig. 1; more in Methods sections "Translation" to "Clustering by disciplines, tracks, global region and economic status").

Most of the organizations in our sample (73%) are based outside Europe and North America, offering a more diverse perspective than previous studies[5–9]. Our study covers 32 countries in the Global North and 89 countries in the Global South, although we do not provide country-level analyses. To better reflect the realities of an average applicant, we applied post-sampling weights, so that each policy from the same organization had equal weight, each organization within a country had equal weight, and each country was weighted proportionally to its number of active researchers[36–42] (Supplementary Information section 1.7 and Supplementary Tables 4 and 9). Our geographical coverage is illustrated in Fig. 1, and a breakdown of the data is provided in Extended Data Table 1. Full data and code are available via the Data and Code availability sections.

## General outlook of promotion criteria

We included all policies for promotion to professorship that clearly specified evaluation criteria, covering both research and teaching (see eligibility in Methods section "Data cleaning (eligibility criteria)"). Policy documents varied widely in scope, structure and level of detail; 28% of the policies in our sample were brief guidelines, whereas 72% included detailed application or evaluation forms with points-based systems (Extended Data Tables 2 and 3 and Supplementary Information sections 1.4 and 1.5). Generally, policies defined three domains:

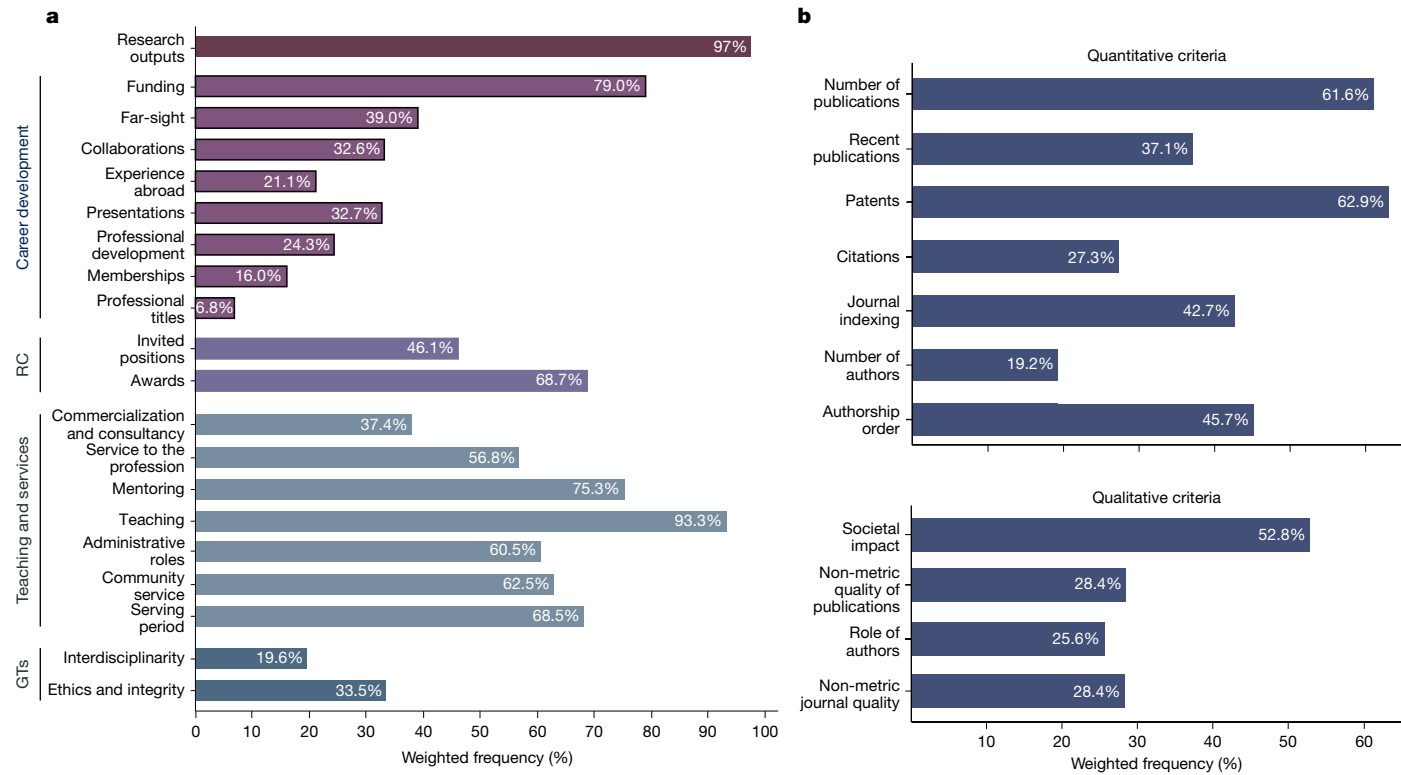

**Fig. 2 | Trends in research assessment. a,b,** Frequency with which each of the 30 assessment criteria is estimated to affect researchers in the 121 countries surveyed. **a** shows 19 criteria along with the general category of research outputs, which is expanded into 7 quantitative (top) and 4 qualitative (bottom) measures shown in **b** ($n = 532$). RC, recognition; GTs, general traits.

research, teaching and services. We identified 30 key criteria across 5 categories: research outputs; career development; recognition; teaching and service; and general traits. Within research outputs, 11 criteria were identified as either 'quantitative' (metrics-based; for example, bibliometrics) or 'qualitative' (narrative descriptions and/or peer-review) descriptors (Methods section 2.3 "Criteria and categories", Extended Data Table 3, Supplementary Information section 1.3 and Supplementary Table 1).

As shown in Fig. 2, policies in our sample reflect an assessment system that prioritizes research outputs (97%), teaching (93%), funding (79%) and mentoring (75%), followed by criteria relating to professional services and recognition, such as administrative roles, awards and societal service (each between 60% and 70%). Research outputs are more often assessed through quantitative measures (92%) than through qualitative measures (77%) ($F(1, 531) = 7.88$, $P = 0.0052$, $n = 532$; in this section and the section below, all Wald tests for equal proportions account for sample weights and are based on an $F(1, 531)$, with $n = 532$). The use of the single quantitative and qualitative methods also varied, with a frequent reliance on patents (63%) and number of publications (61%) among the quantitative measures, and on societal impact (53%) among the qualitative ones.

## Regional and institutional differences

A fair distribution in our sample of national (41%) and institutional (59%) policies from both the Global North (31%) and Global South (69%; United Nations Statistics Division 2018 classification)[43] enabled us to analyse policy types by region. As shown in Fig. 3a, in our sample, both policy types assess teaching with no evidence for a statistical difference in frequency (national: 91%, institutional: 94%, $F = 0.305$, $P = 0.5814$), and both use bibliometrics for assessing research outputs (national: 85%, institutional: 89% of documents using at least one quantitative criterion; Extended Data Table 3 and Extended Data

Fig. 1), such as the number of publications (national: 76%, institutional: 59%, $F = 1.353$, $P = 0.246$) or citations (national: 33%; institutional: 26%, $F = 0.258$, $P = 0.612$), for which no evidence of significant differences was found, but their focus diverged on some specific criteria. National policies prioritized research output metrics, such as journal indexing ($F = 6.059$, $P = 0.015$) and recent publications ($F = 4.065$, $P = 0.045$). By contrast, institutional policies reflected a broader scope, with greater emphasis on qualitative measures ($F = 4.95$, $P = 0.0265$, for a test on the use of at least one policy criterion defined as qualitative; Extended Data Table 3), such as non-metric quality of publications ($F = 19.622$, $P = 0.000$), and valued interdisciplinarity ($F = 4.818$, $P = 0.029$) and career development aspects such as long-term scientific prospects (far-sight; $F = 36.016$, $P = 0.000$).

At the simple level of observed frequencies, we find a higher proportion of policies in the Global South (95%) relying on quantitative measures compared to the Global North (84%). However, high variability within each group (with 95% confidence intervals of the estimated average frequencies ranging between 0.71 and 0.92 for the Global North, and between 0.76 and 0.99 for the Global South) renders this difference statistically nonsignificant ($F = 2.44$, $P = 0.1188$), although a one-sided equivalence test for a difference of at least 5 percentage points yields $F = 9.04$, $P = 0.0028$. Similarly, we observed a difference that is not statistically significant at the conventional level in the average frequency of reliance on qualitative measures in the Global North (83%) and the Global South (61%; $F = 3.27$, $P = 0.0712$), yet again, the high variability within each group yields a one-sided equivalence test for a difference of at least 5 percentage points resulting in $F = 5.22$, $P = 0.0227$ (in this case, the 95% confidence intervals are 0.70–0.91 for the Global North and 0.39–0.79 for the Global South; rationale for one-sided test in Methods section "Frequency statistics"). A disaggregated regional analysis, shown in Fig. 3b, revealed that most differences between types of policy are found in the Global North, for which qualitative criteria were more frequently used in institutional than in national policies (in

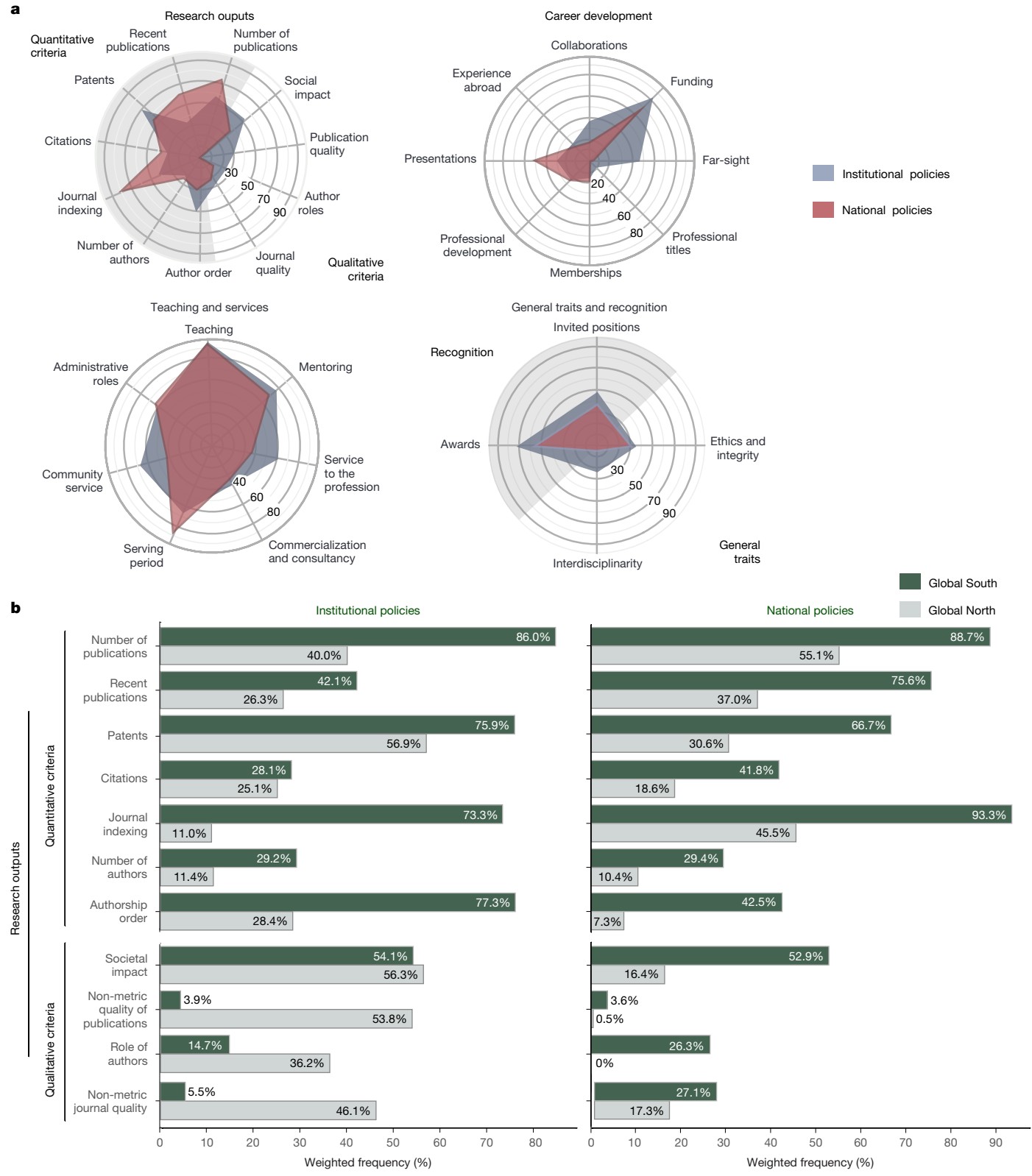

**Fig. 3 | Frequencies by policy type and global regions. a**, Spider plots showing the frequency of 30 criteria among institutional (grey) and national (red) policies, grouped into four spider plots according to the class of the criteria: research outputs; career development; teaching and services; and general traits and recognition. **b**, Bar plots comparing the frequency of qualitative and quantitative criteria for assessing research outputs within national (right) and institutional (left) policies, distinguishing between Global South (green) and Global North (grey). Institutional $n = 314$ (North, 141; South, 173); national $n = 218$ (North, 24; South, 194).

89% and 34% of policies, respectively; $F(1, 86) = 14.7623$, $P = 0.0002$; in this subsample, $n = 165$). By contrast, the Global South does not exhibit significant differences between policy types in the use of qualitative measures (both at 61%, with $F(1, 181) = 0.0001$, $P = 0.9926$; in this subsample, $n = 367$). Detailed results for Pearson's $\chi^2$ and design-based $F$-tests of equal proportions for each criterion by policy type, world

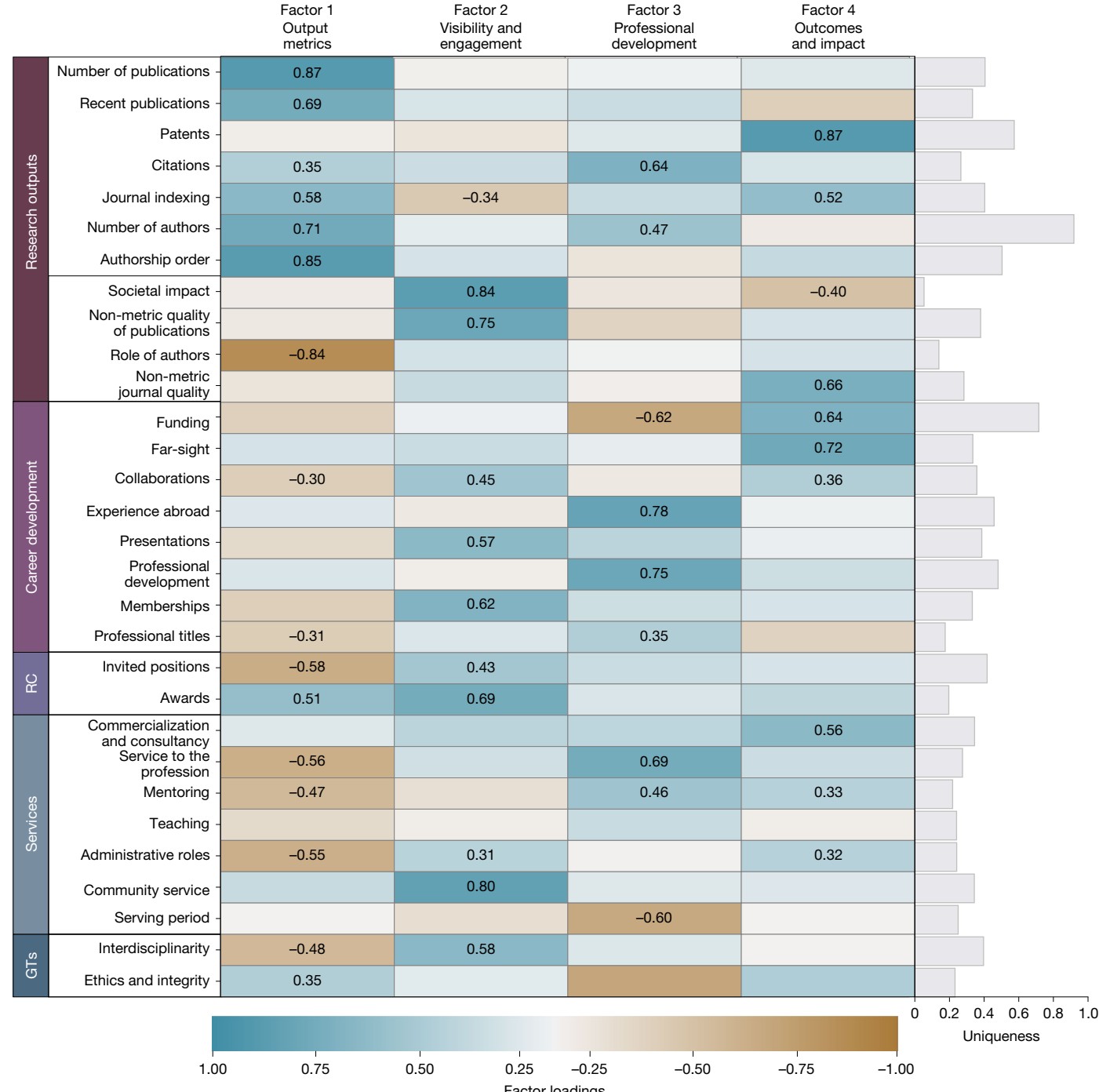

**Fig. 4 | Factor analysis of the assessment criteria for promotion to professorship.** The heat map shows the factor loadings (a measure of correlation; Methods) of each assessment criterion in the 532 policies on each of the 4 latent factors (factors 1–4) predicted after principal factor analysis and rotated with the oblimin oblique method. 'Uniqueness' is the fraction of the variance that a given criterion does not share with others. Blanks denote loading <0.3 in absolute value, and other values are highlighted with a colour scale; all loadings are shown in Supplementary Table 5. We assigned factor interpretation labels to the four factors, to describe the set of criteria they cover.

region, discipline, track and economic status of the country are provided in Supplementary Table 5.

## Main trends in assessment

Beyond measuring frequencies, we examined co-occurrence patterns to examine whether any correlations between policy choices could be indicative of the standpoints of individual institutions and/or national governments. Using principal factor analysis, we found four latent factors that collectively accounted for 65% of the cumulative variance in the data (see scree plot in Extended Data Fig. 2). As can be observed in the loading plots (Extended Data Fig. 3), each factor represents a pattern of distinct clusters of criteria found together in the same policies. Notably, quantitative and qualitative criteria for assessing research outputs are clearly separated across factors, emphasizing their role as key differentiators in policy design. This suggests that organizations often make distinct choices between quantitative or qualitative methods and then consistently apply them throughout the

policy. By contrast, Fig. 4 illustrates how other categories of predefined criteria were intertwined, with some showing loadings greater than 0.3 across multiple factors. For ease of interpretation, all factors were standardized to range from 0 to 1. A detailed description of the analysis is presented in the Methods section "Factor analysis"; the matrix of pairwise tetrachoric correlations of the single criteria is presented in Extended Data Table 4, along with the visualization of the distribution of factors and tests on the differences in the distribution of each factor between categories of policies and additional results (Extended Data Fig. 4 and Extended Data Table 5, accordingly). As shown in Fig. 4, each criterion aligned with at least one of the four factors (listed in order of variance explained), which can be interpreted as follows–(factor 1) output metrics: quantitative assessment of publications and awards; (factor 2) visibility and engagement: engagement with the academic and wider communities, interdisciplinary efforts, and recognition in terms of invited positions and awards; (factor 3) career development: experience abroad, professional development, service to the profession and mentoring, coupled with citations as an element of recognition; (factor 4) outcomes and impact: patents, funding, societal impact, far-sight, and commercialization and consultancy.

## Factors influencing policy criteria

To investigate which policies placed greater emphasis on specific assessment criteria, we performed separate regression analyses on each of the four factors presented in Fig. 4 (as per Methods section "Multivariate regression"). Policies were classified on the basis of contextual factors, such as global region, national per-capita income (World Bank[44]) and continent; and policy- or job-related attributes, such as career track, discipline (Organisation for Economic Co-operation and Development[45]) or whether the policy was specific to full professorships or applied more broadly to scholars. Concerning the contextual factors, in a second set of regressions, we further differentiate policy type (institutional versus national) by global region. For both specifications, the coefficients along with robust standard errors and test statistics are detailed in Supplementary Table 7. Figure 5 summarizes the main results for the specification with region-specific policy types (as seen in Supplementary Table 7, the coefficients of the other variables do not change between the two specifications). Although direct comparisons across factors are impossible owing to the use of different dependent variables in each regression, the figure clearly demonstrates that the extent to which policy categories align with each factor varies substantially.

A key result of our study is that job characteristics–such as discipline, career track or rank–although intuitively relevant, often exhibited coefficients that were not statistically significant. For example, among disciplines, only engineering-specific policies differ significantly (with $P < 0.05$) from those applicable across disciplines (the reference group), showing a higher emphasis on outcomes and impact (coefficient (Coeff) = 0.17, $t$ = 3.212, $P$ = 0.002; see Supplementary Table 7). Similarly, among the various tracks, only research-focused ones showed differences significant at the 5% level, placing greater emphasis on candidates' visibility (Coeff = 0.127, $t$ = 2.797, $P$ = 0.006) and outcomes and impact (Coeff = 0.094, $t$ = 2.086, $P$ = 0.038), and less on career development (including criteria such as experience abroad and mentoring; Coeff = −0.206, $t$ = −2.554, $P$ = 0.012). Policies specific to full professorships were also less likely to prioritize career development (Coeff = −0.191, $t$ = −2.437, $P$ = 0.016). All other coefficients related to job characteristics were not statistically significant at 5% level in all four regressions (see also Extended Data Tables 4–6 for univariate analyses of differences across disciplines and tracks).

Regarding contextual characteristics of the policy documents, rather than characteristics of the job, we found statistically significant differences across continents compared to Europe. Policies from Asia placed less emphasis on visibility (Coeff = −0.201, $t$ = −3.363, $P$ = 0.0009) and career development (Coeff = −0.163, $t$ = −4.020, $P$ = 7.59 × 10$^{-5}$)

while focusing more on outcomes and impact (Coeff = 0.126, $t$ = 1.99, $P$ = 0.0471). Latin American policies relied less on output metrics (Coeff = −0.297, $t$ = −4.328, $P$ = 2.13 × 10$^{-5}$) and visibility (Coeff = −0.224, $t$ = −3.149, $P$ = 0.002), whereas Oceania's policies focused more on outcomes and impact (Coeff = 0.202, $t$ = 2.829, $P$ = 0.0051).

Overall, policies from the Global South exhibit a reliance on outcomes and impact that is not statistically different from those from the Global North (Coeff = −0.0636, $t$ = 0.4550, $P$ = 0.3420), but the two regions differed in a statistically significant way in the other three factors, with the Global South relying more on output metrics (Coeff = 0.219, $t$ = 4.803, $P$ = 2.61 × 10$^{-5}$), visibility (Coeff = 0.211, $t$ = 3.94, $P$ = 0.000104) and career development (Coeff = 0.114, $t$ = 1.99, $P$ = 0.0476).

Across regions, national policies placed a stronger emphasis on output metrics compared to institutional policies (Coeff = 0.0699, $t$ = 2.7260, $P$ = 0.0068). However, more differences by policy type emerge when distinguishing the two main world regions (for all intersections, the reference group is the institutional policies in the Global South). National policies in the Global North emphasized output metrics (Coeff = 0.101 $t$ = 2.89, $P$ = 0.0043) but showed a nonsignificant, negative association with visibility (Coeff = −0.122, $t$ = −1.959, $P$ = 0.0512). Institutional policies from the Global South placed greater emphasis on output metrics (Coeff = 0.223, $t$ = 4.84, $P$ = 0.000), visibility (Coeff = 0.193, $t$ = 3.81, $P$ = 0.0002) and career development, although the last of these did not reach statistical significance (Coeff = 0.110, $t$ = 1.842, $P$ = 0.0667). Finally, national policies in the Global South placed more emphasis on output metrics (Coeff = 0.263, $t$ = 4.712, $P$ = 0.000), visibility (Coeff = 0.289, $t$ = 3.68, $P$ = 0.0003) and career development (Coeff = 0.183, $t$ = 2.688, $P$ = 0.0076).

Finally, our analysis revealed a significant association between average national income and promotion criteria. Visibility is a key focus in higher-income countries, whereas in comparison upper-middle-income countries place more emphasis on metrics (Coeff = 0.123, $t$ = 3.627, $P$ = 0.0003) and less on visibility (Coeff = −0.148, $t$ = −3.211, $P$ = 0.002). Similarly, lower-middle-income (Coeff = −0.197, $t$ = −2.992, $P$ = 0.003) and low-income (Coeff = −0.243, $t$ = −3.067, $P$ = 0.002) countries show a decreased emphasis on visibility, with low-income countries also showing a reduced focus on outcomes and impact (Coeff = −0.362, $t$ = −2.732, $P$ = 0.0067).

Beyond examining the statistical significance of individual coefficients, we compared likelihood-based information criteria for our models (Fig. 4 and Extended Data Table 6) with those of alternative models that exclude all job characteristics (disciplinary field, career track and full professorship specificity) or only exclude disciplinary fields. As shown in Supplementary Table 8, the models that include policy document characteristics (continent, global region and economic status of the country) show a substantial improvement in log-likelihood (ranging from 19% for career development to 127% for output metrics) over a simple intercept-alone model across all four factors. However, when it comes to including job characteristics, we see different preferred model specifications for each factor. The best-fitting models for output metrics and for visibility and engagement exclude job characteristics (or at least the disciplinary field), whereas models for outcomes and impact and for career development perform best with job characteristics included.

To investigate the geographical representativeness of our analysis, and understand whether researchers based in countries with smaller research systems face different conditions, we examined whether the results were influenced by a few countries with large research systems (Supplementary Information section 2.5). We repeated the principal factor and regression analyses on a subsample excluding the ten largest countries, which account for 72% of the estimated global researcher population. In the resulting subsample, representing 28% of the research population, based in the smaller research systems, policies followed patterns similar to those of the full sample. The same four main factors of Fig. 4 emerged from the co-occurrence of criteria (see also Extended Data Table 6). These factors aligned with

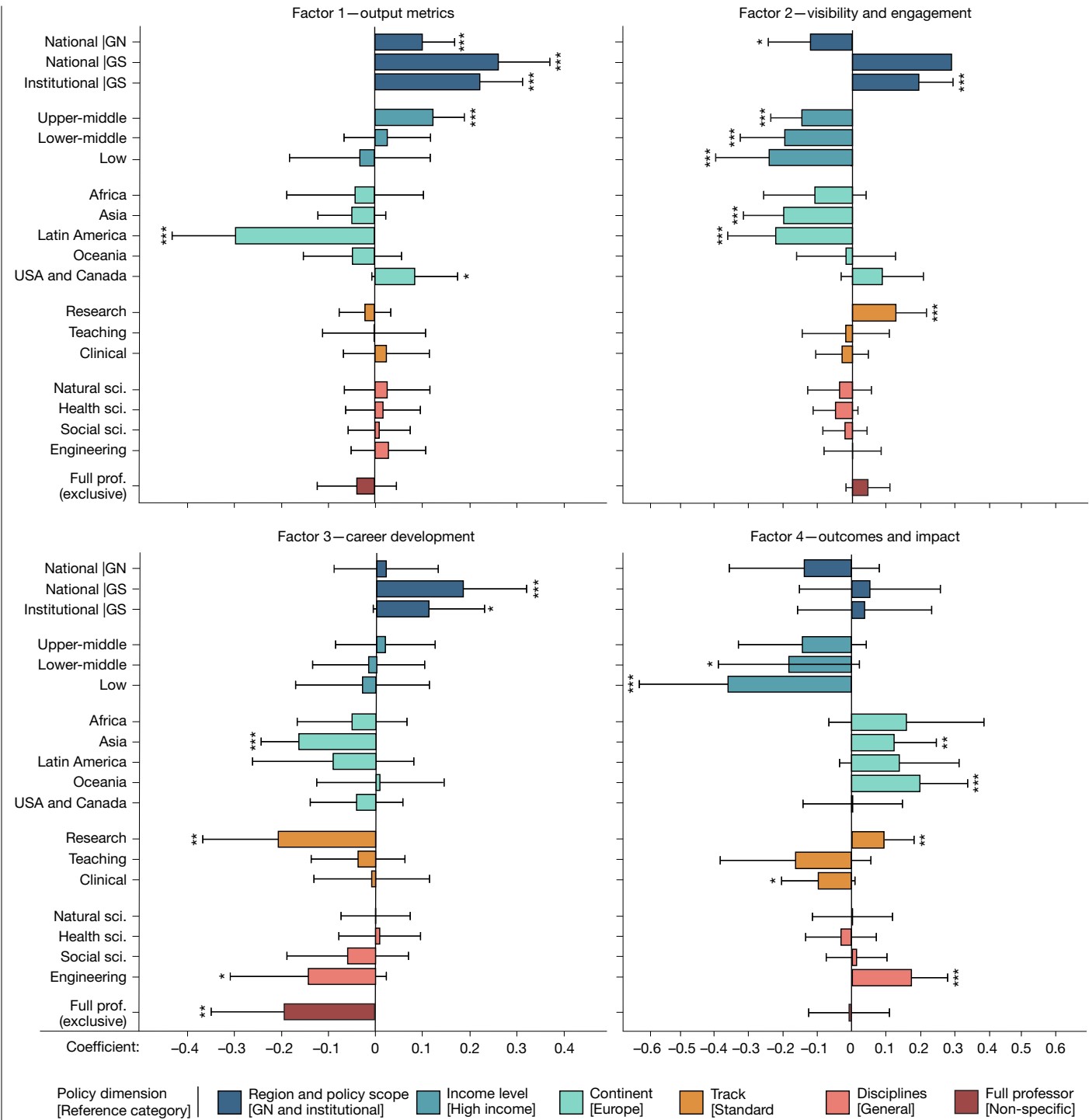

**Fig. 5 | Coefficients of the regression analyses.** Relation between the four predicted factors and policy or country characteristics noted in this study. For the categorical variables, the relation is measured in terms of deviation from a reference category shown in brackets in the figure legend. From top to bottom, categories are: region and policy scope (dark blue), income level (blue), continents (cyan), tracks (orange), disciplines (salmon), and exclusive to full professor (red). Variables with a statistically significant coefficient are indicated; \*\*\*$P < 0.01$, \*\*$P < 0.05$, \*$P < 0.1$; two-sided $t$-tests of difference from zero. Exact $P$ values for each variable are provided in Supplementary Table 7. The number of policies within each category can be found in Extended Data Table 1. The length of the bars denotes the size of the coefficient, and the length of the lines denotes the 95% confidence intervals based on robust standard errors. Sample size ($n$) = 531. The values for each coefficient, their standard errors and statistical significance are reported in Supplementary Table 7. GN, Global North; GS, Global South; sci., science; Prof., professor.

policy characteristics similarly to the full sample (as seen in Fig. 5). However, as emerges from comparing Supplementary Tables 7, 10 and 11, some regression coefficients showed differences in statistical significance—most notably, the three coefficients highlighting differences between the Global North and Global South are not statistically significant in the smaller subsample, as happens for the coefficients denoting upper-middle-income countries (see Supplementary Information section 2.5 for further details).

## Discussion

Our study provides a broad, inclusive dataset capturing promotion policies across under-represented regions, presenting, to our knowledge, one of the most diverse views available on promotion practices. Through rigorous data coding and analysis methods, including post-sampling weighting and factor analysis, we reveal distinct patterns in assessment practices and explore factors that may influence them, offering insights into global policy alignment and diverse institutional needs. However, the scope of our analyses is limited to the presence or absence of criteria, without any regard to the process of hiring or promotion. We relied on snowball sampling rather than a randomized global sample, and our data are not representative at the country level. Additionally, our methodology was initially developed through a pilot study, with no established protocol at that stage.

Our analysis reveals key findings that both institutions and researchers should consider, and which are essential for shaping national research policies.

First, promotion criteria are not identical across institutions. We identified substantial variation around common profiles, with no universally applied criteria. Many institutions have the flexibility to adapt criteria to their needs, accommodating diversity among institutions and researchers' career paths. However, we did not observe a free mix-and-match of criteria, but rather diversity around characteristic clustering of criteria. This allows researchers to align their skills with suitable institutions, although not every institution will be a good fit. However, differences between assessment systems that are uniform at the national level pose challenges for international mobility[46], particularly for early-career researchers from countries in the Global South[47].

Second, scientometrics are most popular in upper-middle-income countries. Although these aim to close the gap with stronger economies, high-income countries rely more on in-depth assessments of researchers' qualities. This raises the question of the effectiveness of catch-up strategies. There may be a misunderstanding about what drives the success of top-performing economies. Metrics appeal owing to their perceived simplicity and objectivity, but the true meaning of 'progress' and 'success' may be unclear. If metrics are not aligned with societal goals, they lose purpose. Many metrics systemically disadvantage lower-income countries and their researchers[48]. Focusing on these measures risks staying behind and missing opportunities to leap ahead. We note that this result is driven by few countries with large research systems and is not replicated in our subsample of smaller countries. This once again points to the variety of approaches among different contexts.

Third, national and institutional policies show different preferences, with divergent regional trends. Quantitative measures for assessing research outputs are generally more popular, but national policies more often emphasize publication counts and venues, whereas institutional policies focus on author order and roles. Quantitative measures are more popular in national policies and generally in the Global South, for which visibility and career development are emphasized too. It is an open question whether these trends are supply-led (for example, there may simply be more highly visible candidates in the upper-income countries, and this makes that group of criteria more feasible), or demand-led (the use of metrics in national policies is often defended on the basis of their scale and cost[49]).

Fourth, the pronounced differences are not between disciplines. We found more variation in assessment criteria within disciplines than between them, at least for two of our four main policy factors, and with geographic, regional and income group differences being often statistically significant. This contrasts with calls for discipline-specific assessments. Outcomes and impact resonate most with engineering, but many challenges and solutions lie between disciplines, as researchers rarely fit into one field. Discoveries in one area influence others, affecting society. Research assessment belongs to the social sciences and should be informed by them, although the popular h-index was introduced by a physicist in 2005 (ref. 50). A one-size-fits-all approach will not work; we need a framework that respects diversity and encourages cross-disciplinary connections.

Fifth, a bibliometric profile is not a key to success everywhere. Whether to apply quantitative or qualitative approaches for assessing research outputs is a most distinctive feature of promotion policies. Bibliometrics are used frequently but not universally. Our study exhibits 39% of policies not mentioning publication counts, 57% not mentioning journal indexing, and 73% not mentioning citations. Policies covering 11% of candidates explicitly warn against the misuse of bibliometrics. These findings align with studies covering the USA and Canada[6]. A strong bibliometric profile is often insufficient, especially in competitive promotions. Committees also value mentoring, administrative work and contributions to the field. For professorial candidates, relying on metrics without meaningful contributions may offer limited success.

Harnessing skills for a globally equitable research ecosystem requires moving beyond normative career tracks that serve as proxies for success. Conforming to standardized profiles harms diversity and limits mobility within and across academia, industry, government and non-profit-making organizations. Skilled researchers risk exclusion owing to rigid policies and biases held by assessors. Outer circumstances or luck should not be mistaken for individual ability. Previous achievement is not necessarily a good predictor of future potential. Models tailored to certain institutions or countries fail to provide meaningful global benchmarks, as researchers face varying conditions[51]. Knowledge advancement should deliver societal benefits, but this is not simply about demonstrating 'impact'. Impact can mean different things, from citations to tangible contributions such as technology transfer or economic returns[52]. Citations mainly reflect social networks[19], and systems built on these metrics can support narcissistic, deceitful or abusive behaviour[53–56]. Flawed methodologies often produce spectacular results, favouring the 'natural selection of bad science'[57]. Research integrity is vital, yet some assessment practices fail to recognize it or even undermine good standards[58].

Meaningful assessment requires clarity about the desired qualities. It is necessary to have an understanding of what distinguishes an academic, and of the quality and value of intellectual achievements. Breakthroughs differ vastly from standardized industrial-scale output production. Moreover, researchers do not function independently, but are part of teams and collaborations, and are highly interdependent actors in an intrinsically complex global research and innovation ecosystem. Individual snapshots do not capture this well. We need team players who create added value by elevating their colleagues[59]. Thus, it is worth building a narrative that captures all relevant dimensions of the researcher profile, considering their context, and potential for growth.

Metrics foster monocultures, whereas the global research ecosystem thrives from diversity across global, national and regional levels down to institutions and research teams[60,61]. We need approaches that foster diversity, rather than imposing norms that limit creativity and impact by promoting a predominant culture[13]. University ranking systems push institutions into unwinnable competition, preventing them from leveraging unique strengths. This raises the question of whether institutions need to break free from these pressures to deliver true value and benefits to society. If so, policies must not allocate funds or other benefits (such as eligibility for hosting students or research visitors, granting of visas and so on) on the basis of rankings.

Research assessment shapes career strategies. Researchers align with promotion criteria, but when measures become targets, they lose their effectiveness ('Goodhart's law')[62,63]. Some criteria remain vague, such as collegiality, which includes both good citizenship and conformity, subject to interpretation[64]. Researchers often wonder how many publications are needed for career progression, but publication counts and journal prestige are not decisive in some processes[65,66]. Researchers face a dissonance between what counts, what is perceived

to count, and what should count[67,68], leading to a dilemma: contribute to society or prioritize career advancement. Excelling depends on standing out, not just performing well on the same criteria as everyone else. Researchers must face the dilemma of building their career on profiting from fitting a popular profile, but potentially failing owing to the lack of distinction and personal ambitions, or by not developing a well-distinguished profile of excellence that is less popular but might appeal strongly to specific teams or institutions[53,61].

The obsession with frequently ill-suited metrics has created inefficiencies in the research ecosystem. Although many countries in the Global North can afford such inefficiency (but should not), it is important that the Global South adopts strategies focused on building appropriate research culture. Our study challenges South–North catch-up strategies based on unsuitable performance indicators. As demonstrated by Latin America's world-leading model for open access publishing[69], building purposeful research environments is not primarily a matter of funding, but mostly about fostering a different kind of culture. Such initiatives can provide key input to platforms such as the Coalition for Advancing Research Assessment[33] that aim at building a global community. Rather than letting the Global North sort out things that the Global South then adapts to, actors from the Global South are well suited to take the lead on global initiatives that show the way forwards.

## Conclusions

Our findings reveal that promotion policies worldwide show considerable variation, with no universal criteria, reflecting diverse institutional and regional needs. Principal factor analysis identified four main assessment clusters—output metrics; visibility and engagement; career development; and outcomes and impact—each reflecting distinctive patterns across policies. Although trends appear when differentiating by policy scope, global region, continent or income group, substantial variability persists within each subsample. Quantitative metrics, particularly in the Global South, frequently underpin assessment frameworks, whereas high-income countries tend to prioritize qualitative attributes, such as visibility and engagement, to assess academic merit. This reliance on metrics in the Global South raises concerns about the effectiveness of 'catch-up' strategies, as metrics alone may reinforce regional inequities. National policies often emphasize output measures such as publication counts, whereas institutional policies, more responsive to local and institutional priorities, focus more broadly on aspects such as societal impact and interdisciplinary work. In contrast to assessment systems that that are uniform at national level and that may hinder international mobility, institutional policies may offer a framework enabling researchers to find institutions that match their skills. These insights suggest that flexible, context-sensitive frameworks are essential to balance global equity with institutional needs, fostering a resilient research ecosystem that values diverse contributions across the academic landscape.

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

¹Lee Kong Chian Faculty of Engineering and Science, Universiti Tunku Abdul Rahman, Kajang, Malaysia. ²Department of Statistical Sciences, Sapienza University of Rome, Rome, Italy. ³Centro Interdisciplinare Linceo Giovani, Rome, Italy. ⁴Centre for Exoplanet Science, SUPA School of Physics & Astronomy, University of St Andrews, St Andrews, UK. ⁵Centro de Investigación y de Estudios Avanzados del IPN, Mexico City, Mexico. ⁶Laboratoire SPHERE UMR 7219, CNRS, Paris, France. ⁷Université Paris Cité, Paris, France. ⁸Université Paris-1 Panthéon-Sorbonne, Paris, France. ⁹Department of Politics and International Relations (DPIR), University of Oxford, Oxford, UK. ¹⁰Centro de Estudios de Derecho e Investigaciones Parlamentarias, Cámara de Diputados, Mexico City, Mexico. ¹¹College of Graduate Studies, University of South Africa, Pretoria, South Africa. ¹²Nanosciences African Network (NANOAFNET), iThemba LABS-National Research Foundation, Somerset West, South Africa. ¹³Department of Geography, Higher Teacher Training College, The University of Bamenda, Bambili, Cameroon. ¹⁴Forest Institutions and International Development (FIID) Research Group, Faculty of Environmental Science, Technische Universität Dresden, Tharandt, Germany. ¹⁵Instituto de Medicina Experimental (IMEX), CONICET-Academia Nacional de Medicina, Buenos Aires, Argentina. ¹⁶Department of Biomedical and Dental Sciences and Morphofunctional Imaging, University of Messina, Messina, Italy. ¹⁷Department of Biological Sciences, College of Science, University of Santo Tomas, Manila, The Philippines. ¹⁸Department of Research in Virology and Biotechnology, Gorgas Memorial Institute for Health Studies, Panama City, Republic of Panama. ¹⁹Sistema Nacional de Investigación, Secretaria Nacional de Ciencia, Tecnologia e Innovacion, Panama City, Panama. ²⁰Department of Mathematics and Computer Science, Faculty of Sciences and Technics, Abdou Moumouni University, Niamey, Niger. ²¹School of Law, Politics, and Sociology, University of Sussex, Brighton, UK. ²²Univ. Grenoble Alpes, IRD, CNRS, INRAE, Grenoble INP, IGE, Grenoble, France. ²³University of Zagreb, Faculty of Food Technology and Biotechnology, Zagreb, Croatia. ²⁴Institute of Medical Research and Medicinal Plants Studies, Yaoundé, Cameroon. ²⁵Department of Computing Technologies, Swinburne University of Technology, Melbourne, Victoria, Australia. ²⁶Institute for Advanced Study, Technische Universität München, Garching, Germany. ²⁷Department of Internal Medicine, Faculty of Medicine, Public Health, and Nursing, Universitas Gadjah Mada, Yogyakarta, Indonesia. ²⁸Egypt Solid Waste Management Center of Excellence, Faculty of Engineering, Ain Shams University, Cairo, Egypt. ²⁹Department of Pharmacy and Pharmacology, School of Therapeutic Sciences, Faculty of Health Sciences, University of the Witwatersrand, Johannesburg, South Africa. ³⁰Department of Microbiology and Immunology, Faculty of Pharmacy, Suez Canal University, Ismailia, Egypt. ³¹Department of Chemical Engineering, International University of Liaison Indonesia, South Tangerang, Indonesia. ³²Department of Physics, Chemistry & Material Science, University of Namibia, Windhoek, Namibia. ³³Centre for Space Research, North-West University, Potchefstroom, South Africa. ³⁴Synthetic and Systems Biology for Biomedicine, Istituto Italiano di Tecnologia, Naples, Italy. ³⁵Institute of Fundamental and Applied Research, National Research University TIIAME, Tashkent, Uzbekistan. ³⁶Faculty of Biology, National University of Uzbekistan, Tashkent, Uzbekistan. ³⁷CancerResearch@UCC, University College Cork, Cork, Ireland. ³⁸Institute for Protein Design, University of Washington, Seattle, WA, USA. ✉e-mail: yensi.floresbueso@ucc.ie

## Methods

This study aimed to identify commonalities and differences in promotion criteria to full professor across global institutions, as outlined in institutional and government documents. We focused on this senior role owing to its comparability, given that career progression pathways and roles vary substantially across countries. Rather than making a priori assumptions about the structure and content of promotion policies, our study design was informed by initial textual analysis described in Supplementary Information section 1.1. In the following, we focus on the methods allowing the reproduction of the manuscript results: data acquisition, data preparation and data analysis. Methods regarding the study design, including the sampling strategy, definition of subsamples, categories and criteria, are detailed in Supplementary Information section 1. The defined protocols are shared via Figshare (see Data availability).

### Data acquisition

Using the network of members and alumni of the Global Young Academy as a platform, we conducted snowball sampling by requesting members, alumni and their networks to source documents describing academic promotion policies from their institutions and broader academic networks. This included collecting both publicly available and confidential documents from countries and regions for which they are familiar with the language and promotion frameworks. This approach enabled us to obtain a representative sample across all world regions without relying on impractical stratified random sampling. As mentioned in paragraph three of the "Study design" section in the main text and Supplementary Information section 1.2.2, we conducted three rounds of policy sourcing, with the latest in 2023. From this exercise, we sourced 440 policies, representing 83% of our data, with 460 (87%) of the policies in our sample being applicable as of December 2023. Throughout this process, we included every sourced document that was clear and comprehensive enough to identify the presence or absence of specific promotion criteria, ensuring suitability for our analysis. More detail is available in Supplementary Information sections 1.1–1.2.1.

### Data preparation

Given our sample of documents describing academic promotion policies, we needed to extract the 'policy' (that is, their content) from these documents and to identify suitable characteristics for statistical analysis. This involved the steps that are described in detail in Supplementary Information section 1.2, and summarized below.

**Translation.** Our sample included documents in 27 languages, which were translated to English using translation software (for example, Google Translate) for consistent analysis. Translations were verified by fluent speakers within the team. Non-machine-readable documents were processed using optical character recognition tools (more details in Supplementary Information section 1.2.4).

**Data cleaning (eligibility criteria).** The documents varied substantially in structure and level of detail (Extended Data Table 2 and Supplementary Information section 1.5). We included only documents with clear, measurable criteria, excluding duplicates such as national policies reported by multiple institutions. Documents relying on vague terms such as 'excellence in research', 'leadership' or 'international visibility' without specific, measurable achievements were excluded. Additionally, only documents addressing the role of full professor and covering both research and teaching responsibilities were considered. We focused on policies related to the sciences and humanities, excluding arts and creative works owing to substantial differences in outputs and achievements.

**Criteria and categories.** This study analyses the presence or absence of these 30 criteria in professorship promotion policies, originally defined in a pilot study to capture key policy features comparably and quantifiably (Supplementary Information sections 1.1 and 1.4). Full definitions and rationale for these criteria are in Supplementary Table 1, with further explanation in Supplementary Information section 1.4.1 and Supplementary Table 2. The criteria are organized into three main categories: research (21 criteria); teaching and services (7 criteria); and general traits (2 criteria). Research is further divided into research outputs (11), career development (8) and recognition (2), with research outputs split into quantitative (7) and qualitative (4). Additional details on the rationale and categorization process are provided in Supplementary Information section 1.4.

**Clustering by disciplines, tracks, global region and economic status.** Each distinct set of assessment criteria was treated as a separate 'policy for promotion' and was clustered for analysis by: disciplines—on the basis of the Organisation for Economic Co-operation and Development classification[45], policies were categorized into natural sciences, engineering and technology, medical and health sciences, and social sciences and humanities, with a 'general' category for non-specific policies; career tracks—policies aligned with four tracks: standard academic, research-focused, teaching-focused and clinical; region—we followed the United Nations Geoscheme[70], defining six regions: Africa, Asia, Europe, Latin America and the Caribbean, Northern America, and Oceania; and economic classification—the World Bank's income groups (low, lower-middle, upper-middle and high)[44] and the United Nations Statistics Division 2018 Global North versus South classification[43] were used. Policies were also categorized by whether they applied specifically to full professors or scholars in general. Full details of these classifications are in Supplementary Information section 1.3, with data splits shown in Extended Data Table 1.

**Data coding.** Our analysis focuses solely on the presence or absence of specific criteria, not the assessment process itself. We did not evaluate the role, weight or interpretation of criteria, or how assessment panels reach decisions, which may include additional, unstated criteria (Extended Data Table 2 and Supplementary Information sections 1.4 and 1.5). A standardized data coding template in Google Sheets was shared with regional teams through Google Drive, organized into region-specific folders with individual subfolders for each team member. Regional teams sourced documents, identified policies and coded attributes, including: policy scope (institutional or national); discipline and career track scope; regional data; and document details (for example, year of implementation and completeness). Data coders assessed each policy for 30 predefined promotion criteria, strictly adhering to definitions in Supplementary Table 1, scoring each as present (1) or absent (0) on the basis of exact definitions. To ensure transparency and reusability, coders documented the reference for each criterion, noting text location and relevant quotes from the policy in adjacent cells in the dataset, shared in the replication package (Data availability). Each policy was independently reviewed by two team members and cross-reviewed by the regional team lead. Team leaders consolidated data, mediated differences and consulted project leads as necessary. Discrepancies were resolved collaboratively among team leads, coders and project leads, with definitions of criteria refined as needed to ensure consistency across the dataset.

**Weighting.** We applied post-sampling weights in the analysis of the dataset obtained from coding the documents. As shown in Supplementary Table 9, the distribution of active researchers by country is highly skewed, with 72% of them based in the ten countries with the largest research systems. Accordingly, policies from these countries substantially influence our findings. Specifically, with $r$ denoting the global number of researchers, $r_k$ and $n_k$, respectively, representing the number of researchers and institutions (or agencies) in country $k$, and $p_j$ representing the number of policies for institution $j$, the weight $w_{jk}$ of

a policy of institution $j$ in country $k$ becomes $w_{jk} = r_k / (p_j n_k r)$. This weight was then applied to each variable captured in the analysis. Information on the number of researchers was primarily obtained from the UNESCO World Data Bank (February 2024), with estimates for countries lacking data derived from alternative sources[36–42]. Further details on data sources, assumptions and the impact of weighting on the results are provided in Supplementary Information section 1.7.

### Data analysis and visualization

**Frequency statistics.** Data coded in Google Sheets was exported to an Excel 2024 workbook, cleaned to remove formatting, and converted to a CSV file before being imported into Stata 17 (both the raw and cleaned datasets are shared in our replication package). We analysed the frequency of criteria appearing in policies by grouping the data into subsets defined by key attributes: policy scope (institutional versus national), Global North versus South, national per-capita income group and disciplinary scope. Proportions within these categories were computed using the weighted data (as described in the "Weighting" section above). Power analysis for our dataset is in Supplementary Information section 2.3, Supplementary Fig. 1 and Supplementary Table 6; with all tests, the estimated power is above 0.95. Pearson chi-squared ($\chi^2$) tests were used to assess systematic differences in criteria across policy categories. Given the weights applied to the data, we also performed a design-based $F$-test, the most appropriate statistical test, as reported in the main text and fully detailed in Supplementary Table 5. All analyses were conducted in Stata 17. For specific code and scripts, see the replication package in CodeOcean (Data availability). One-sided equivalence tests were chosen in our analyses to assess whether differences between groups exceed a predefined threshold of practical relevance (5 percentage points). This approach was considered more parsimonious than a two-sided test as it directly addresses whether observed differences reach a meaningful level rather than testing for any deviation. One-sided tests are also well suited to cases with high within-group variability, focusing on practically relevant differences and enhancing interpretability while conserving statistical power.

**Factor analysis.** To identify potential associations between criteria in the analysed policies and assess their degree of co-occurrence, we computed a matrix of pairwise tetrachoric correlations for the dichotomous variables (present or absent), measuring the degree of co-occurrence between the criteria. The results of this correlation analysis are presented in Extended Data Table 4. We then conducted a principal factor analysis on this correlation matrix to explore the underlying factors or dimensions within the data. The primary trends in the research assessment criteria were synthesised using four continuous variables, all normalized to values between zero and one for ease of interpretation. The number of factors was chosen on the basis of both conceptual concerns (ease of interpretation and relevance of the latent factor) and empirical considerations (summarized by the eigenvalues, as shown in Extended Data Fig. 1). To facilitate interpretation, we applied oblimin oblique rotation to these factors. The full account of factor loadings is presented in Extended Data Table 6 and for more information on the method, see ref. 71.

**Multivariate regression.** The factors derived from the previous analysis were predicted for the full sample and normalized to values between 0 and 1. Separate regression analyses were conducted for each factor to examine how the criteria in the policy documents correlate with job-related characteristics (for example, track and discipline) and environmental characteristics (for example, policy scope, global region and country). The estimation method used was linear regression, applying the post-sampling weights as defined earlier, while controlling for potential heteroskedasticity and clustering of error terms by institution or national authority. We report in Supplementary Table 7 the full results of the regressions of the four factors on the explanatory variables explored in this study, including heteroskedasticity-robust standard errors, two-sided $t$-tests of difference from zero and corresponding $P$ values[72,73].

**Data visualization.** Data from Stata were imported into Python 3 and plotted using Python's Matplotlib, seaborn and geopandas libraries. Panels were assembled and formatted in Adobe Illustrator 2024.

### Reporting summary

Further information on research design is available in the Nature Portfolio Reporting Summary linked to this article.

## Data availability

All the data sourced for this study are available via Figshare at https://figshare.com/s/f8aa5ab402440a9a7933 (ref. 74). Our data package includes the full raw dataset coded by the authors, the metadata in a readme file and the protocols used to source, code and analyse the data in a study protocol file. For institutions for which policies were not public the data were anonymized.

## Code availability

All code used to generate the analysis in the manuscript is shared by the authors for reuse in a CodeOcean capsule available at https://doi.org/10.24433/CO.0942594.v1 and https://doi.org/10.24433/CO.3615162.v1.

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

**Acknowledgements** We thank the Global Young Academy, of which all authors except V.S.M.-S. are current members or alumni, for providing the platform and funding that facilitated the development of this work; our peers for voluntary participation, as much of this work was conducted on a voluntary basis; the Global Young Academy office for support; A.K. Claessens, A. Xuereb, A. Bhadra and G. Bassioni for advice and/or assistance at early stages of the project; our peers within and outside the Global Young Academy for assisting in collecting policies for this study, of whom we can mention (some we cannot): A. Ahmad, A. Samakov, A. Godoy-Faúndez, A. Villarreal Medina, A. Awan, A. Kuuwill, A.K. Claessens, A. Sum, A. Villarino, A. Rich, A. Bhadra, A. Sidorovich, A. Xuereb, A. Betti, A. Bernier, B. Hennig, C. E. Rojas Zenozaín, B.C. Kok, B. Wrobel, C. Choudhury, C.C. Diaz, C. Stanley, C. Rios Rojas, C. Nshemereirwe, E. Rojas Prado, E. Castellanos, E. F. Khor, E. T. Lim, E. Corrales-Aguilar, E. Alisic, F. Ramos Quispe, F. A. Phang, F. Vargas Lehner, F. Edi Soetaredjo, N. Gaab, G. Ferreira, G. Fuente, G. Bassioni, G. Tornaría, H. Shunker, H. Abdalla, H.C. Yang, H. Cheng, H.H. Goh, I. Kurnaz, I. Torres, J. Young, J.D. Romero Carpio, K. Chan, K. Zaafouri, K. B. Tan, K. Binger, K. Fairfax, K. Taman, L.M. Freire, L. Fierce, L. Sokny, M. Nasr, M. Peccianti, M. Vergara Rubio, M. Wieling, M. D. Balela, M. Elhadidy, M. T. Rahman, M. M. Karim, M. Wahajuddin, A.K. Mukong, M. Pieri, N. Nguyen, N. Guerrero González, N. Arenas, N. Kwarikunda, N. Yasuda, N. Meethong, N. Ahmed, O. Nguyen, O. Hod, O. Adeyemo, P. Simpemba, R. Owusu, R. Al Bakain, S. Leonelli, S. Kaur-Ghumaan, S. Maw, S.L. Fernández Valverde, S. Komai, S.T. Tan, S. Hild, S. Bhattarai, T.T.M. Hanh, W. S. Ho, W.S. Chang, W. Ochoa, W. Setthapun, X. Chiriboga, X. B. Tran, Y. F. Chan and Z. Haiguang. A. Simonyan acknowledges his scholarship from the Calouste Gulbenkian Foundation. All authors acknowledge funding support from Taighde Éireann – Research Ireland under Grant number 18/SP/3522, Breakthrough Cancer Research under Precision Oncology Ireland, HORIZON-MSCA-2021-PF-101059124 (BacStar) and the Global Young Academy.

**Author contributions** K.V., B.H.L. and M.D. were involved in the conception, early design and execution of the study. K.V. led the project from 2016 to 2018, B.H.L. led the project from 2018 to 2021, and Y.F.B. led the project from 2021 until the publication of this work. Data coding, involving the collection, translation and tabulation of policy data, was carried out by (in descending order of contributions): B.H.L., K.V., L.B., J.N.K., A.M.I.S., I.S.Z., S. Elagroudy, K.J.C., A.C.H.-M., H.H., A.S., Y.F.B., P.K., V.S.M.-S., A.R.J., J.G.N., K.K.C., T.E.d.C., S. Enany, D.E., S.M., V.N., I.P., S.L.-V. and A.M. Data revision, involving the review of policies, reviewing annotation, discussion and agreement of definitions, was performed by (in descending order of contributions): B.H.L., Y.F.B., A.C.H.-M., H.H., V.S.M.-S., K.K.C., K.J.C., A.R.J., J.G.N., T.E.d.C., S.M., V.N., J.N.K., A.S., S. Enany, I.S.Z., S.L.-V., A.M., V.S., P.K., I.P., L.B., S. Elagroudy, D.E. and A.M.I.S. Refer to our dataset shared on the data section to see the exact policies tabulated and/or reviewed by each author, listed in the columns labelled "tabulated" and "reviewed". B.H.L. and Y.F.B. collated and merged

the dataset. The published version of the manuscript was mainly drafted by M.D., C.D'I., Y.F.B., B.H.L. and A.C.H.-M. All figures were drafted by Y.F.B., with support from C.D'I., M.D. and B.H.L. All statistical analyses were performed by C.D'I., Y.F.B. and B.H.L. Manuscript revisions 3–5 were mainly performed by Y.F.B., C.D'I., M.D., B.H.L., A.C.H.-M., K.K.C., S.L.-V. and V.S.M.-S. (in descending order of contributions). The first submitted manuscript was prepared with substantial contributions from M.D., B.H.L., C.D'I., Y.F.B., M.B., K.K.C., A.C.H.-M., I.P., T.E.d.C., S.M., S.L.-V., P.K. and A.M. (in descending order of contributions). The protocol was drafted by S.M., Y.F.B., V.S.M.-S., S.L.-V., K.K.C., I.P., S.Elagroudy, B.H.L., C.D'I. and M.D., on the basis of guidelines and protocols prepared by Y.F.B. and B.H.L. All authors reviewed and approved the content of the study and are personally responsible for their contributions as stated here.

**Competing interests** The authors declare no competing interests.

**Additional information**
**Correspondence and requests for materials** should be addressed to Y. Flores Bueso.

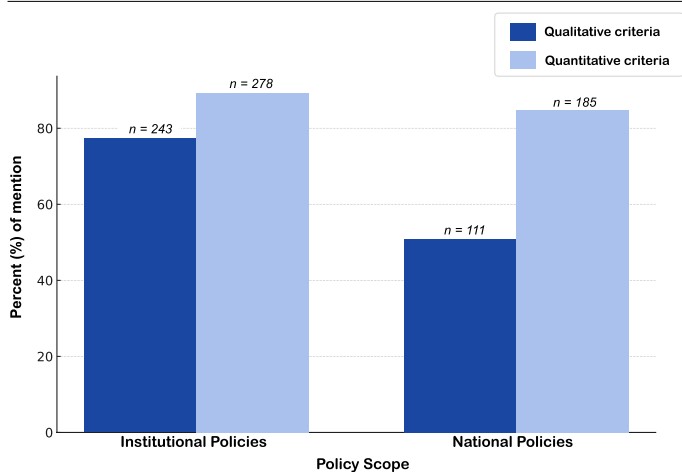

**Extended Data Fig. 1 | Approach to assessment of research outputs by policy scope.** Application of quantitative vs qualitative criteria for the 314 institutional and 218 national policies in our dataset. Result shown is the percentage of policies mentioning each type of assessment. Number of policies (n) for each criterion is presented above each corresponding bar.

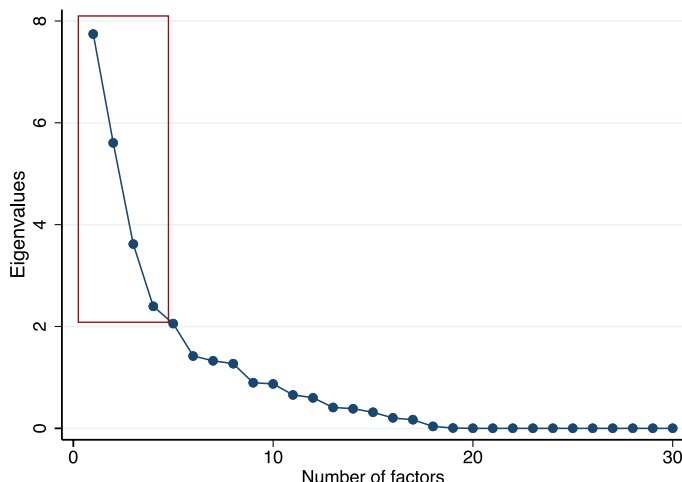

**Extended Data Fig. 2 | Scree plot for the factor analysis.** Showing the factors obtained, ordered by eigenvalue - denoting the level of variability captured by the factors. In the red box are the Eigenvalues for the four factors used.

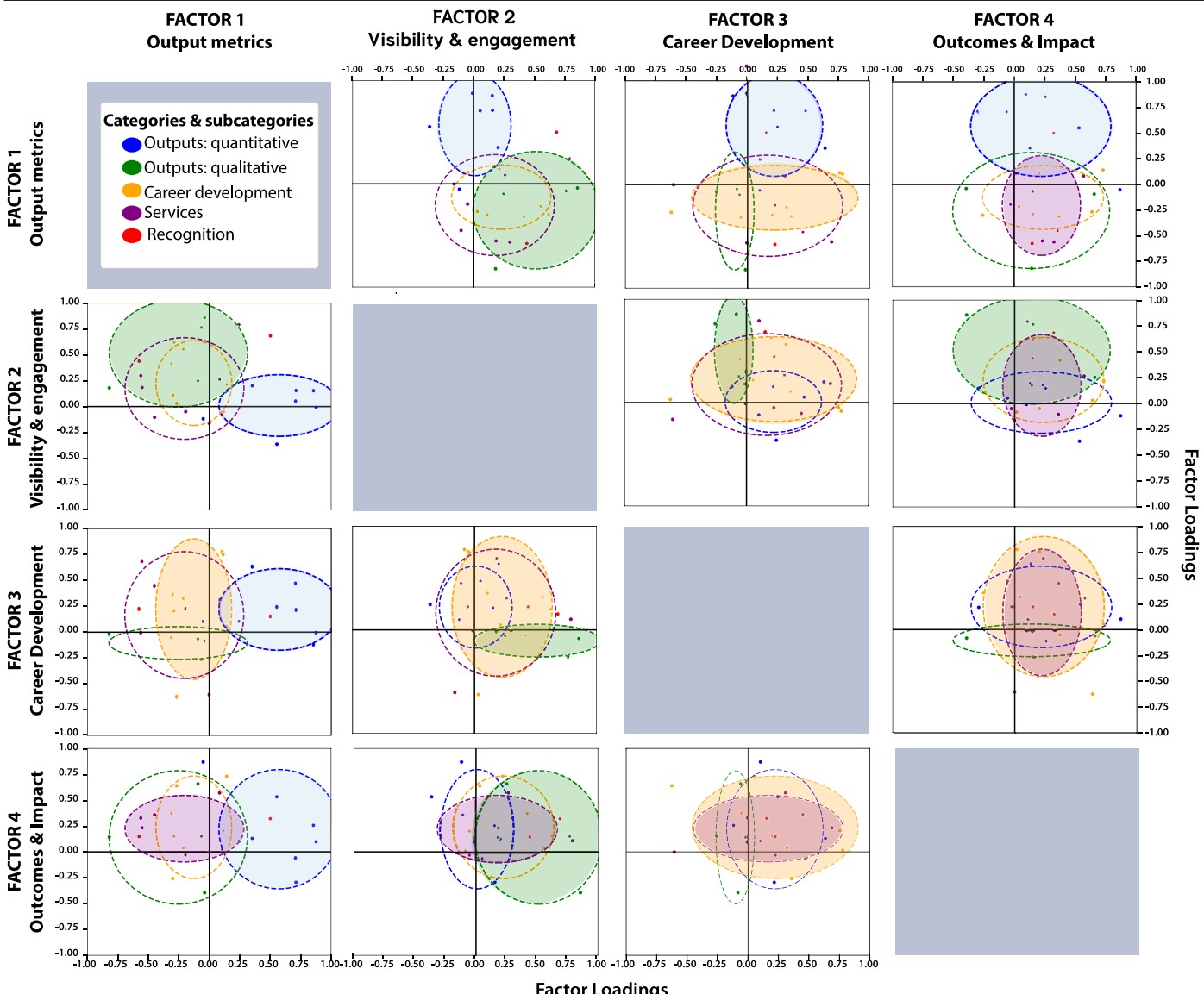

**Extended Data Fig. 3 | Loading plots for each pair of factors.** Each scatter plot in the grid represents a pairwise comparison between factors: Factor 1: *Output Metrics*, Factor 2: *Visibility and Engagement*, Factor 3: Career Development, and Factor 4: Outcomes & Impact. Data points are colour-coded to differentiate between quantitative outputs (blue), qualitative outputs (orange), career development criteria (yellow), services (purple), and recognition (red).

Ellipses represent the concentration and dispersion of data points associated with each group, indicating the variance and co-relationship strength between factors. Ellipses were overlaid using matplotlib, ellipse function, where the width and height of the ellipse were set to reflect the standard deviation for each group along the X and Y axes, respectively. $N = 532$.

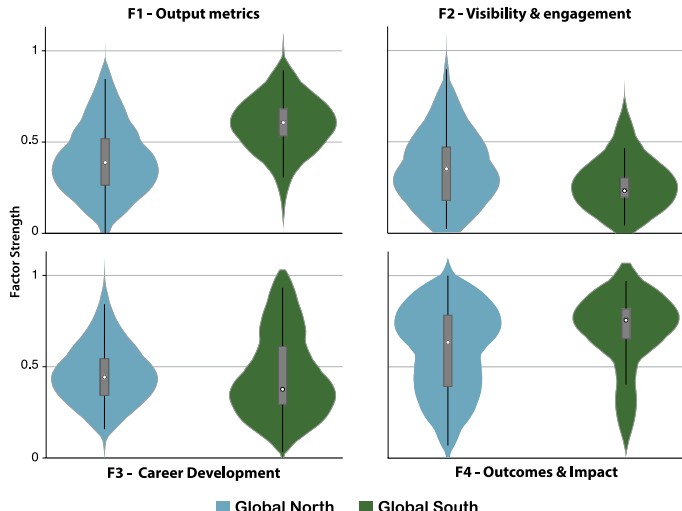

**Extended Data Fig. 4 | Single variable comparison between Global North and South.** Violin plots showing median, quartiles, and the distributions of the families of criteria described by the four latent factors resulting from the principal factor analysis. Here, factor strength is the overall impact of the measured factor in explaining the observed data (factor score for each policy). Note: The correlation among criteria, the factor loadings and their variance, the factor scores and their graphical representation through kernel density and boxplots are all based on the weighted sample. N for Global North = 165 and for Global South = 367.

**Extended Data Table 1 | Description of the sample and its distribution by Region, Economic Status, Disciplines, and Tracks**

| CLASS | Counts | | | | | |
|---|---|---|---|---|---|---|
| | Countries | Government Agencies | Academic institutions | National policies | Institutional policies | Total policies |
| *Global Regions* | | | | | | |
| Global North | 32 | 14 | 73 | 24 | 141 | 165 |
| Global South | 89 | 44 | 117 | 194 | 173 | 367 |
| *Economic status (Income level)\** | | | | | | |
| High-income | 37 | 13 | 84 | 27 | 157 | 184 |
| Upper-middle income | 32 | 18 (2) | 52 | 56 | 83 | 139 |
| Lower-middle income | 33 | 20 (8) | 43 | 68 | 61 | 129 |
| Low-income | 19 | 6 (12) | 11 | 67 | 13 | 80 |
| *Continents* | | | | | | |
| Europe | 28 | 15 | 40 | 25 | 62 | 87 |
| Africa | 41 | 14 | 44 | 145 | 61 | 206 |
| Asia | 33 | 19 | 56 | 28 | 91 | 119 |
| Latin America | 16 | 10 | 20 | 20 | 21 | 41 |
| North America | 2 | 0 | 20 | 0 | 57 | 57 |
| Oceania | 1 | 0 | 10 | 0 | 22 | 22 |
| *Disciplines* | | | | | | |
| General | - | 51 | 165 | 77 | 187 | 264 |
| Natural sciences | - | 9 | 16 | 41 | 24 | 65 |
| Engineering & Technology | - | 9 | 12 | 31 | 25 | 56 |
| Medicine & Health Sciences | - | 10 | 28 | 32 | 46 | 78 |
| Social sciences & humanities | - | 11 | 22 | 41 | 42 | 83 |
| *Tracks* | | | | | | |
| Standard academic track | - | 48 | 178 | 179 | 247 | 426 |
| Research track | - | 15 | 44 | 150 | 53 | 203 |
| Teaching track | - | 1 | 22 | 1 | 38 | 39 |
| Clinical track | - | 5 | 20 | 5 | 25 | 30 |
| Other tracks | - | 0 | 11 | 0 | 13 | 13 |
| *Evaluation exclusive to the role of (full) professor?* | | | | | | |
| Not exclusive for professor | - | 16 | 54 | 42 | 70 | 112 |
| Specifically for professor | - | 43 | 140 | 176 | 244 | 420 |
| *TOTAL* | | | | | | |
| Number of countries | - | - | - | 63 | 58 | 121 |
| Full dataset | 121 | 58 | 190 | 218 | 314 | 532 |

Total counts of policies segmented by global regions (Global North and Global South), economic status (high-income, upper-middle-income, lower-middle-income, and low-income countries), and continents for all out policies. It also breaks down the data by academic disciplines, career tracks, and whether the policies specifically apply to full professors. Both national and institutional policies are shown, with total policy counts for each category. *The number in parenthesis gives the distribution across categories for the 22 countries covered by the supra-national agency CAMES. For the splits by disciplines and career tracks, some policies cover more than one category.*

**Extended Data Table 2 | Attributes of the documents describing promotion policies**

| Level of detail | Score < 3 | Score = 3 | Score >3 |
|---|---|---|---|
| *Tracks* | | | |
| Academic | 27 | 79 | 320 |
| Research | 2 | 16 | 185 |
| Teaching | 3 | 15 | 21 |
| Clinical | 1 | 10 | 19 |
| Other | 1 | 2 | 10 |
| *Disciplines* | | | |
| Natural Sciences | 0 | 8 | 57 |
| Engineering & technology | 0 | 7 | 49 |
| Medicine & health sciences | 3 | 19 | 56 |
| Social sciences & humanities | 0 | 15 | 68 |
| General | 28 | 71 | 165 |
| *Source of policy* | | | |
| National | 12 | 35 | 171 |
| Institutional | 19 | 81 | 214 |
| *Global North and Global South* | | | |
| Global North | 11 | 37 | 117 |
| Global South | 20 | 79 | 268 |
| *Economic status (Income level)* | | | |
| High-income | 12 | 45 | 127 |
| Upper-middle income | 8 | 27 | 104 |
| Lower-middle income | 7 | 35 | 87 |
| Low-income | 4 | 9 | 67 |

Summary of data collected on policy features. The collected documents varied in structure and detail. Key features summarised include document type, inclusion of supplementary materials, and whether criteria applied exclusively to professors or across disciplines/tracks. The table also includes average scores (1-3) for the level of detail, as assessed by the data coders.

**Extended Data Table 3 | Evaluation criteria classified by the sub-categories defined by the authors**

| CRITERIA | Output metrics | Visibility and engagement | Career development | Outcomes & impact | Uniqueness |
|---|---|---|---|---|---|
| Number of Publications | 0.8675 | -0.014 | 0.0104 | 0.0972 | *0.2337* |
| Recent Publications | 0.6978 | 0.1426 | 0.2285 | -0.2918 | *0.3874* |
| Patents | -0.0573 | -0.1178 | 0.092 | 0.8723 | *0.2513* |
| Citations | 0.3456 | 0.2009 | 0.6365 | 0.1415 | *0.3446* |
| Journal indexing | 0.5572 | -0.3516 | 0.2409 | 0.5315 | *0.2402* |
| Number of authors | 0.7047 | 0.0561 | 0.4771 | -0.0561 | *0.2431* |
| Authorship order | 0.8551 | 0.1569 | -0.1146 | 0.2475 | *0.2141* |
| Non-metric journal quality | -0.0458 | 0.8445 | -0.093 | -0.3991 | *0.2717* |
| Role of authors | -0.0693 | 0.7526 | -0.2575 | 0.1521 | *0.3396* |
| Nonmetric publication quality | -0.8313 | 0.168 | -0.0156 | 0.1459 | *0.2062* |
| Social Impact | -0.106 | 0.2397 | -0.0518 | 0.6629 | *0.4207* |
| Farsight | -0.2675 | 0.0272 | -0.63 | 0.6274 | *0.1806* |
| Funding | 0.1424 | 0.2192 | 0.0438 | 0.7258 | *0.3334* |
| Collaborations | -0.3079 | 0.4281 | -0.061 | 0.3718 | *0.4802* |
| Experience abroad | 0.0998 | -0.0792 | 0.775 | 0.0246 | *0.3892* |
| Presentations | -0.2093 | 0.5626 | 0.3029 | 0.0429 | *0.4504* |
| Professional development | 0.1092 | -0.0446 | 0.7519 | 0.2109 | *0.3545* |
| Memberships | -0.2899 | 0.6235 | 0.1939 | 0.1469 | *0.3361* |
| Professional Titles | -0.3051 | 0.1224 | 0.3512 | -0.2671 | *0.7137* |
| Invited positions | -0.5805 | 0.4375 | 0.2149 | 0.1462 | *0.2776* |
| Awards | 0.5102 | 0.6841 | 0.1412 | 0.2883 | *0.142* |
| Commercialisation/Consultancy | 0.0844 | 0.2786 | 0.3006 | 0.5785 | *0.3682* |
| Service to the Profession | -0.5701 | 0.1897 | 0.6828 | 0.2032 | *0.0589* |
| Mentoring | -0.4788 | -0.1457 | 0.4415 | 0.3191 | *0.5135* |
| Teaching | -0.2053 | -0.0367 | 0.2302 | -0.0455 | *0.9124* |
| Administrative roles | -0.5579 | 0.3065 | -0.019 | 0.323 | *0.3967* |
| Community service | 0.2388 | 0.7965 | 0.1087 | 0.1119 | *0.2701* |
| Serving period | 0.0096 | -0.1548 | -0.6141 | -0.0229 | *0.5643* |
| Interdisciplinarity | -0.471 | 0.5808 | 0.104 | -0.0079 | *0.3357* |
| Ethics and integrity | 0.3596 | 0.08 | -0.6288 | 0.3714 | *0.4054* |

Sub-categories and criteria used to analyse promotion policies. The criteria were mainly grouped under three main categories: Research, Teaching & Service, and General Traits. Categories were selected through an exercise by team members as expanded in SI sect. 1.4.

# Extended Data Table 4 | Heatmap of the correlation among the criteria in each policy

| CRITERIA | No. Publications | Recent publications | Patents | Citations | Journal indexing | No. authors | Authorship order | Non-metric jouranl quality | Role of authors | Non-metric quality of publications | Societal impact | Farsight | Funding | Collaborations | Experience abroad | Presentations | Professional development | Memberships | Professional Titles | Invited positions | Awards | Commercialisation / Consultancy | Service to the profession | Mentoring | Teaching | Administrative roles | Community service | Serving period | Interdisciplinarity |
|---|---|---|---|---|---|---|---|---|---|---|---|---|---|---|---|---|---|---|---|---|---|---|---|---|---|---|---|---|---|
| Recent publications | 0.84 | 1.00 | | | | | | | | | | | | | | | | | | | | | | | | | | | |
| Patents | 0.08 | -0.13 | 1.00 | | | | | | | | | | | | | | | | | | | | | | | | | | |
| Citations | 0.35 | 0.40 | 0.21 | 1.00 | | | | | | | | | | | | | | | | | | | | | | | | | |
| Journal indexing | 0.77 | 0.24 | 0.40 | 0.42 | 1.00 | | | | | | | | | | | | | | | | | | | | | | | | |
| No. authors | 0.76 | 0.73 | 0.06 | 0.56 | 0.47 | 1.00 | | | | | | | | | | | | | | | | | | | | | | | |
| Authorship order | 0.78 | 0.46 | 0.11 | 0.38 | 0.65 | 0.66 | 1.00 | | | | | | | | | | | | | | | | | | | | | | |
| Non-metric jouranl quality | -0.19 | 0.08 | -0.27 | 0.06 | -0.61 | 0.02 | -0.09 | 1.00 | | | | | | | | | | | | | | | | | | | | | |
| Role of authors | -0.12 | 0.01 | 0.29 | 0.18 | -0.37 | -0.47 | 0.03 | 0.47 | 1.00 | | | | | | | | | | | | | | | | | | | | |
| Non-metric pulications qual. | -0.74 | -0.57 | 0.26 | -0.18 | -0.79 | -0.62 | -0.74 | 0.29 | 0.40 | 1.00 | | | | | | | | | | | | | | | | | | | |
| Societal impact | -0.14 | 0.03 | 0.62 | 0.20 | 0.02 | 0.01 | 0.07 | 0.05 | 0.52 | 0.55 | 1.00 | | | | | | | | | | | | | | | | | | |
| Farsight | -0.19 | -0.51 | 0.48 | -0.45 | -0.12 | -0.74 | 0.15 | -0.03 | 0.41 | 0.50 | 0.52 | 1.00 | | | | | | | | | | | | | | | | | |
| Funding | 0.19 | -0.16 | 0.58 | 0.23 | 0.36 | 0.52 | 0.28 | 0.30 | 0.20 | 0.58 | 0.68 | | 1.00 | | | | | | | | | | | | | | | | |
| Collaborations | -0.24 | -0.46 | 0.28 | -0.01 | -0.35 | -0.26 | 0.21 | 0.30 | 0.50 | 0.49 | 0.38 | 0.30 | | 1.00 | | | | | | | | | | | | | | | |
| Experience abroad | 0.09 | 0.29 | 0.16 | 0.54 | 0.31 | 0.56 | 0.07 | -0.22 | -0.12 | -0.03 | 0.11 | -0.49 | 0.17 | 0.07 | 1.00 | | | | | | | | | | | | | | |
| Presentations | -0.27 | -0.23 | 0.26 | 0.32 | 0.02 | -0.05 | -0.15 | 0.53 | 0.53 | 0.24 | 0.02 | -0.12 | 0.28 | 0.39 | 0.13 | 1.00 | | | | | | | | | | | | | |
| Professional development | 0.13 | 0.22 | 0.24 | 0.53 | 0.42 | 0.46 | 0.21 | -0.09 | -0.11 | -0.24 | 0.12 | -0.39 | 0.36 | -0.09 | 0.37 | 0.38 | 1.00 | | | | | | | | | | | | |
| Memberships | -0.24 | -0.14 | 0.26 | 0.07 | -0.19 | 0.02 | -0.36 | 0.47 | 0.53 | 0.42 | 0.38 | 0.06 | 0.30 | 0.60 | 0.10 | 0.66 | 0.19 | 1.00 | | | | | | | | | | | |
| Professional Titles | -0.28 | -0.02 | -0.31 | 0.06 | -0.19 | -0.33 | -0.14 | 0.05 | -0.07 | 0.17 | -0.41 | -0.15 | -0.14 | 0.34 | 0.56 | 0.32 | 0.27 | 0.43 | 1.00 | | | | | | | | | | |
| Invited positions | -0.53 | -0.42 | 0.18 | 0.34 | -0.33 | -0.42 | -0.41 | 0.47 | 0.59 | 0.62 | 0.39 | 0.19 | 0.31 | 0.44 | 0.02 | 0.75 | 0.06 | 0.71 | 0.17 | 1.00 | | | | | | | | | |
| Awards | 0.40 | 0.13 | 0.35 | 0.60 | 0.30 | 0.64 | 0.61 | 0.34 | 0.63 | -0.23 | 0.27 | 0.02 | 0.64 | 0.37 | 0.30 | 0.54 | 0.39 | 0.61 | -0.12 | 0.22 | 1.00 | | | | | | | | |
| Commercialisation/Consultancy | 0.07 | -0.03 | 0.60 | 0.46 | 0.42 | 0.22 | 0.11 | 0.08 | 0.18 | 0.03 | 0.52 | 0.06 | 0.64 | 0.60 | 0.31 | 0.35 | 0.32 | 0.52 | 0.02 | 0.21 | 0.58 | 1.00 | | | | | | | |
| Service to the profession | -0.50 | -0.26 | 0.31 | 0.40 | -0.21 | 0.02 | -0.38 | 0.20 | 0.28 | 0.69 | 0.52 | -0.03 | 0.33 | 0.39 | 0.54 | 0.60 | 0.68 | 0.76 | 0.54 | 0.85 | 0.14 | 0.39 | 1.00 | | | | | | |
| Mentoring | -0.17 | -0.15 | 0.43 | 0.22 | -0.11 | -0.15 | -0.34 | -0.04 | 0.23 | 0.49 | 0.17 | 0.19 | 0.38 | -0.02 | 0.20 | 0.20 | 0.43 | 0.53 | 0.26 | 0.45 | -0.04 | 0.11 | 0.71 | 1.00 | | | | | |
| Teaching | -0.04 | -0.06 | -0.25 | -0.15 | 0.09 | 0.20 | -0.08 | 0.17 | -0.38 | 0.13 | 0.01 | -0.15 | 0.03 | 0.02 | 0.02 | -0.02 | 0.19 | 0.05 | 0.46 | 0.15 | -0.19 | 0.32 | 0.39 | 0.17 | 1.00 | | | | |
| Administrative roles | -0.58 | -0.53 | 0.29 | 0.02 | -0.17 | -0.51 | -0.37 | 0.29 | 0.27 | 0.56 | 0.47 | 0.28 | 0.30 | 0.58 | -0.33 | 0.50 | 0.11 | 0.71 | 0.13 | 0.81 | 0.07 | 0.39 | 0.62 | 0.14 | 0.47 | 1.00 | | | |
| Community service | 0.04 | 0.26 | 0.16 | 0.27 | -0.06 | 0.38 | 0.39 | 0.49 | 0.53 | 0.12 | 0.51 | -0.01 | 0.44 | 0.56 | 0.25 | 0.29 | 0.35 | 0.69 | 0.28 | 0.18 | 0.71 | 0.64 | 0.36 | -0.10 | 0.13 | 0.37 | 1.00 | | |
| Serving period | 0.09 | -0.08 | -0.04 | -0.72 | 0.00 | -0.16 | 0.02 | -0.11 | -0.17 | -0.15 | -0.21 | 0.41 | -0.11 | -0.16 | -0.56 | -0.28 | -0.54 | -0.02 | 0.25 | -0.35 | -0.39 | -0.29 | -0.60 | -0.20 | 0.35 | 0.03 | -0.28 | 1.00 | |
| Interdisciplinarity | -0.43 | -0.33 | 0.06 | 0.04 | -0.42 | -0.40 | -0.35 | 0.46 | 0.69 | 0.76 | 0.31 | 0.24 | -0.04 | 0.65 | 0.11 | 0.49 | 0.04 | 0.61 | 0.37 | 0.56 | 0.23 | 0.36 | 0.73 | 0.43 | 0.33 | 0.37 | 0.65 | -0.27 | 1.00 |
| Ethics & integrity | 0.29 | -0.10 | 0.10 | -0.13 | 0.30 | -0.28 | 0.48 | -0.28 | 0.19 | -0.40 | 0.25 | 0.41 | 0.16 | 0.13 | -0.45 | -0.16 | -0.24 | -0.09 | -0.14 | -0.30 | 0.21 | 0.03 | -0.56 | -0.25 | -0.03 | -0.01 | 0.14 | 0.41 | -0.16 |

Matrix of pairwise tetrachoric correlations among all the promotion criteria used in the analysis, in terms of presence/absence in the same policies (for all policies; N=532). The original matrix with pairwise tetrachoric correlations is not positive semidefinite; for the purposes of the subsequent factor analysis, it has been adjusted to be positive semidefinite.

**Extended Data Table 5 | Differences in the distribution of each factor, between categories of policies**

| CRITERIA | Output metrics | Visibility & Engagement | Professional development | Outcomes & Impact | *Uniqueness* |
|---|---|---|---|---|---|
| N. of publications | 0.682 | | | | *0.4123* |
| Recent publications | 0.378 | 0.3895 | | -0.5299 | *0.3792* |
| Patents | 0.657 | | | 0.4493 | *0.4212* |
| Citations | 0.4566 | | 0.4256 | -0.4755 | *0.3798* |
| Journal indexing | 0.8841 | | | | *0.1821* |
| Number of authors | 0.8619 | | | | *0.2568* |
| Authorship order | 0.7279 | | | | *0.2602* |
| Non-metric journal quality | -0.3468 | 0.7604 | | -0.3749 | *0.2467* |
| Role of authors | | 0.3922 | 0.5019 | -0.3868 | *0.4072* |
| Nonmetric publication quality | -0.4369 | 0.3218 | 0.3917 | | *0.3822* |
| Social Impact | | 0.5907 | 0.3856 | | *0.3616* |
| Farsight | -0.4111 | 0.3827 | 0.4096 | | *0.3866* |
| Funding | | | 0.4525 | | *0.7686* |
| Collaborations | -0.3356 | | 0.6163 | | *0.2679* |
| Experience abroad | | -0.4792 | 0.604 | | *0.5202* |
| Presentations | 0.5423 | | 0.3017 | | *0.6059* |
| Professional development | | | -0.3141 | 0.6616 | *0.4992* |
| Memberships | 0.4024 | 0.3363 | | 0.5264 | *0.4151* |
| Professional Titles | | | | | *0.8625* |
| Invited positions | | 0.4003 | 0.521 | | *0.4637* |
| Awards | 0.5332 | | 0.554 | | *0.4277* |
| Commercialisation/Consultancy | | | 0.38 | | *0.7948* |
| Service to the Profession | | | 0.4043 | 0.6572 | *0.2814* |
| Mentoring | | | | 0.6749 | *0.423* |
| Teaching | | 0.6969 | | | *0.4175* |
| Administrative roles | | 0.7128 | | 0.4734 | *0.2194* |
| Community service | | 0.7296 | | | *0.3479* |
| Serving period | 0.3848 | 0.4178 | -0.4774 | | *0.5234* |
| Interdisciplinarity | | | 0.5314 | | *0.5894* |
| Ethics and integrity | | 0.7222 | | | *0.3858* |

Results of Wald tests of equality of the means of each factor between institutional vs. national policies; Global North vs. Global South; countries' economic status; disciplines; and tracks. *Null hypothesis (HO)*: no difference (diff = 0) in the means between the compared categories; alternative hypothesis (H1): the average in one category (reported) is different (≠) from the average in the other(s). **Bold** font and green denote p-values < 0.05. N = 532.

# Extended Data Table 6 | Full Factor loadings

| Level of detail | Score < 3 | Score = 3 | Score >3 |
|---|---|---|---|
| *Tracks* | | | |
| Academic | 27 | 79 | 320 |
| Research | 2 | 16 | 185 |
| Teaching | 3 | 15 | 21 |
| Clinical | 1 | 10 | 19 |
| Other | 1 | 2 | 10 |
| *Disciplines* | | | |
| Natural Sciences | 0 | 8 | 57 |
| Engineering & technology | 0 | 7 | 49 |
| Medicine & health sciences | 3 | 19 | 56 |
| Social sciences & humanities | 0 | 15 | 68 |
| General | 28 | 71 | 165 |
| *Source of policy* | | | |
| National | 12 | 35 | 171 |
| Institutional | 19 | 81 | 214 |
| *Global North and Global South* | | | |
| Global North | 11 | 37 | 117 |
| Global South | 20 | 79 | 268 |
| *Economic status (Income level)* | | | |
| High-income | 12 | 45 | 127 |
| Upper-middle income | 8 | 27 | 104 |
| Lower-middle income | 7 | 35 | 87 |
| Low-income | 4 | 9 | 67 |

Values for each assessment criterion in the 532 policy documents on each of the four latent factors (F1-F4) predicted after Principal Factor analysis and rotated with the oblimin oblique method (SI section 1.8.2). "Uniqueness" is the fraction of the variance that a given criterion does not share with others.

# Reporting Summary

## Statistics

For all statistical analyses, confirm that the following items are present in the figure legend, table legend, main text, or Methods section.

| n/a | Confirmed | |
|---|---|---|
| ☐ | ☒ | The exact sample size (*n*) for each experimental group/condition, given as a discrete number and unit of measurement |
| ☐ | ☒ | A statement on whether measurements were taken from distinct samples or whether the same sample was measured repeatedly |
| ☐ | ☒ | The statistical test(s) used AND whether they are one- or two-sided<br>*Only common tests should be described solely by name; describe more complex techniques in the Methods section.* |
| ☐ | ☒ | A description of all covariates tested |
| ☐ | ☒ | A description of any assumptions or corrections, such as tests of normality and adjustment for multiple comparisons |
| ☐ | ☒ | A full description of the statistical parameters including central tendency (e.g. means) or other basic estimates (e.g. regression coefficient) AND variation (e.g. standard deviation) or associated estimates of uncertainty (e.g. confidence intervals) |
| ☐ | ☒ | For null hypothesis testing, the test statistic (e.g. *F*, *t*, *r*) with confidence intervals, effect sizes, degrees of freedom and *P* value noted<br>*Give P values as exact values whenever suitable.* |
| ☒ | ☐ | For Bayesian analysis, information on the choice of priors and Markov chain Monte Carlo settings |
| ☒ | ☐ | For hierarchical and complex designs, identification of the appropriate level for tests and full reporting of outcomes |
| ☒ | ☐ | Estimates of effect sizes (e.g. Cohen's *d*, Pearson's *r*), indicating how they were calculated |

*Our web collection on statistics for biologists contains articles on many of the points above.*

## Software and code

Policy information about availability of computer code

| Data collection | Data was collected using Google Sheets and compiled into a single file downloaded and transformed in Microsoft Excel 16 (Mac) to a CSV file that was the input for the software for analysis. The raw data was shared in a csv format in CodeOcean. Both, raw and clean datasets are shared here: https://doi.org/10.6084/m9.figshare.23272175. |
|---|---|
| Data analysis | Data was analysed using Stata17 and plotted using python3 pandas and matplotlib. The panels were assembled and formated using Adobe Ilustrator 2024. The code for data analysis was shared in CodeOcean. https://codeocean.com/capsule/0942594/tree . |

For manuscripts utilizing custom algorithms or software that are central to the research but not yet described in published literature, software must be made available to editors and reviewers. We strongly encourage code deposition in a community repository (e.g. GitHub). See the Nature Portfolio guidelines for submitting code & software for further information.

## Data

Policy information about availability of data

All manuscripts must include a data availability statement. This statement should provide the following information, where applicable:
- Accession codes, unique identifiers, or web links for publicly available datasets
- A description of any restrictions on data availability
- For clinical datasets or third party data, please ensure that the statement adheres to our policy

We have included a Data availability statement and we share all the data produced by this study.

# Research involving human participants, their data, or biological material

Policy information about studies with human participants or human data. See also policy information about sex, gender (identity/presentation), and sexual orientation and race, ethnicity and racism.

| | |
|---|---|
| Reporting on sex and gender | N/A |
| Reporting on race, ethnicity, or other socially relevant groupings | N/A |
| Population characteristics | N/A |
| Recruitment | N/A |
| Ethics oversight | We are not dealing with any humnan participants, their data or biological material. Our study is limited to the analysis of policies that are developed by instituions and may impact humans but do not directly relate to them. |

Note that full information on the approval of the study protocol must also be provided in the manuscript.

# Field-specific reporting

Please select the one below that is the best fit for your research. If you are not sure, read the appropriate sections before making your selection.

☐ Life sciences   ☒ Behavioural & social sciences   ☐ Ecological, evolutionary & environmental sciences

For a reference copy of the document with all sections, see nature.com/documents/nr-reporting-summary-flat.pdf

# Behavioural & social sciences study design

All studies must disclose on these points even when the disclosure is negative.

| | |
|---|---|
| Study description | We conducted a cross-sectional quantitative analysis by examining the assessment criteria used in promotion policies. We systematically identified and analysed promotion criteria, comparing differences and similarities across disciplines, fields, tracks, types of institutions, and countries, considering their socioeconomic contexts. We solely recorded the presence or absence of subjectively chosen distinct assessment criteria in available documents describing promotion policies, without any regard to the process of hiring or promotion. |
| Research sample | The study analyzed 532 promotion policies from 190 academic institutions and 58 government agencies across 121 countries. The sample was chosen to maximize global diversity and capture a wide range of institutional practices, focusing on full professor promotion policies as is the most senior position that can be comparable between institutions in different countries. While not representative for all academic and national organisations, our sample has the power to sustain our claims (see SI sect 2.3) |
| Sampling strategy | The documents were obtained by global snowball sampling originating from the GYA network rather than as a randomised sample across all relevant institutions or authorities in the world, and our sample is not representative at country level. Our sample size was informed by power analysis, which as seen in SI section 2.3 confirm that our sample has the power to sustain our claims. |
| Data collection | Policies were obtained from public websites, academic networks, and official requests. The collected documents were analyzed to identify the presence or absence of specific criteria related to the promotion of full professors. Data coding was performed by members of the team using a standardised template, definitions, protocols and supporting materials. |
| Timing | Data study was initiated in 2016 with a pilot study to define the methodology. Documents were collected since then. However, 83% documents in the version published of this manuscript were obtained in 2022/2023 to ensure that they are still valid. |
| Data exclusions | We excluded data from documents that did not provide clear or comprehensive information about promotion criteria. Additionally, we limited our analysis to promotion policies for full professors, excluding documents that did not cover this role or focused on career tracks unrelated to both research and teaching, to ensure consistency and comparability of different data points. We excluded a total of 24 policies from our study. Throughout the process we documented the exclusion of 59 policies. |
| Non-participation | n/a |
| Randomization | n/a |

# Reporting for specific materials, systems and methods

We require information from authors about some types of materials, experimental systems and methods used in many studies. Here, indicate whether each material, system or method listed is relevant to your study. If you are not sure if a list item applies to your research, read the appropriate section before selecting a response.

## Materials & experimental systems

| n/a | Involved in the study |
|-----|------------------------|
| ☒ ☐ | Antibodies |
| ☒ ☐ | Eukaryotic cell lines |
| ☒ ☐ | Palaeontology and archaeology |
| ☒ ☐ | Animals and other organisms |
| ☒ ☐ | Clinical data |
| ☒ ☐ | Dual use research of concern |
| ☒ ☐ | Plants |

## Methods

| n/a | Involved in the study |
|-----|------------------------|
| ☒ ☐ | ChIP-seq |
| ☒ ☐ | Flow cytometry |
| ☒ ☐ | MRI-based neuroimaging |

## Plants

| Seed stocks | n/a |
|-------------|-----|

| Novel plant genotypes | n/a |
|-----------------------|-----|

| Authentication | n/a |
|----------------|-----|

