## [Peer Review File · Nature]

Regional and institutional trends in assessment for academic promotion

Corresponding Author: Ms Yensi Flores Bueso

Version 1:

Reviewer comments:

Referee #1

(Remarks to the Author)
Please see attached document.

Referee #2

(Remarks to the Author)

Thank you for the opportunity to review the manuscript “A global assessment of academic promotion criteria: What really counts?”, which I read with great interest. The study offers an examination of promotion criteria to full professor (or equivalent) written into national and institutional policies and guidelines from 55 countries. It is, to my knowledge, the most global study of its kind and presents valuable evidence over which assessment criteria are codified into policies and guidelines. The evidence presented largely affirms what is expected from the literature and from the common understanding of promotion criteria while providing an empirical basis for those common assertions. Most notable, for me, is the evidence highlight that the Global South is more likely to use number of publications and citation metrics than the Global North. While aligning with expectations, this study provides solid evidence on the matter.

While there is much to praise from the study, there are also a handful of areas where further clarity is needed and should be provided ahead of publication. These areas are all connected to the wide range of documents that were surely collected and the similarly wide-ranging set of academic and researcher systems to which they apply. While the authors acknowledge that “the structure and style of the policy documents varied substantially,” they provide little description of this variability in relations to the dimensions analyzed. For example, were certain kind of documents more prevalent in some countries than others?

Similarly, the collection of both institutional and national policies is a strength of the study, but the lack of analysis of where national policies were more prominent makes it difficult to disentangle the national/institutional analysis from the Global North/South analysis. This relates to a broader issue of the diversity of systems that exist for researchers around the world (how were equivalencies of full professor found for systems around the world?). Was there any consideration for policies from countries where researchers hold dual affiliations (one with the national system and another within in their institution)? While the manuscript acknowledges that such differences exist (similarly with different career tracks), the study design, analysis, or discussion fails to take them into account. I also found no mention of how many of the documents found could be cleanly mapped to particular career stages/tracks. In my experience, there is a mix of documents that specify career stages/tacks and those that do not.

Again, similarly, there is little detail provided on how much overlap there is between institutional and national level policies, or how much overlap there is between institutional policies (i.e., multiple documents from the same institution, potentially for different disciplines or different academic units). Was having a single document from a specific academic unit enough to include that country in the country-level analysis?

In short, the study is lacking a more completely explanation of how the differences in academic careers and the diversity of

documents were handled. Both of these elements should be expanded upon in Methods and in the Discussion. The differences in academic careers across countries, especially the extent to which academic career progressions can (and cannot!) be compared, should be addressed in the literature review and again in Point 1 of the discussion (noting, for example, that researcher mobility varies greatly by country). (For what it's worth, it was this complexity that held us back from analysing documents globally and made us decide to focus on the US and Canada, where the Review, Tenure, and Promotion systems are largely comparable).

For the most part, I don't think much of the above will affect the result of the analysis, but it would add some detail and present some limitations and caveats that should be addressed. The one area where, I suspect, it might matter the most is in the disciplinary analysis. Where those disciplinary documents are sourced from is likely to matter and some checks should be done to describe the geographic representation of that subset of documents. (Any further detail should be considered in Point 4 of the discussion).

A few other comments to consider:

- The analysis of "outputs" lacks detail on the types of outputs that are being solicited. Are only traditional outputs considered? In our own work (Alperin et al., 2022), we found 127 different types of outputs listed. While I'm not suggesting authors use this level of categorization, it might be helpful to note which types of outputs were included.
- The estimate of number of researchers covered by the study seems inflated. If documents were selected for those that apply promotion to (full) professor rank, then using the more generic "researcher per million" value as an approximation is not accurate. This speaks to the point raised above about the need for further details and discussion on career stages.
- I did not find much value in the flow described in Figure 1 (this flow is described in the text and is sufficiently obvious as to not warrant a visualization). The map is more helpful, although perhaps it could be coupled with further details on number of documents by type, as requested above.
- Section 4 of discussion: possible typo: "Communalities" "Commonalities"?
- I could see an argument made that the latter half of the discussion (after the five summary points) is not sufficiently rooted in the study results or the literature. Perhaps because I find myself agreeing with the perspective provided, I found the section to be a helpful conclusion to the article, but I would not be opposed for further linkages to the study findings and to previous literature.

Despite the questions and comments raised, I believe this to be a valuable study. I find the factor analysis to be a useful way of adding some depth to binary indicators and the conclusions drawn to be reasonably drawn from that analysis. A study of this nature will always have limitations owing to the impossibility of neatly capturing the global diversity in research systems and cultures. I appreciate the ambition of the study and believe that the scholarly community will benefit from being presented with this global comparative view, despite its minor shortcomings.

Alperin, J.P., Schimanski, L.A., La, M., Niles, M.T., & McKiernan, E.C. (2022). The value of data and other non-traditional scholarly outputs in academic review, promotion, and tenure. In *Open Handbook of Linguistic Data Management*. MIT Press

Referee #3

(Remarks to the Author)

The most significant contribution of this study is the global nature of the analysis allowing for comparisons regarding review for academic promotion to be made between Global North vs. South and between countries with different income levels. Similar analyses have been conducted in the past but have not comprehensively analyzed policy/procedure from across the globe. Also interesting are comparisons between institutional level and national policies governing assessment of research outcomes. It is interesting for researchers to gain understanding about what is valued in different regions, countries, and levels of administration. The insights gained have potential to guide efforts toward creating greater consistency in academic procedures globally, and may inspire leaders to apply approaches different from those commonly employed in their own region.

It is helpful that the authors displayed the number of countries from which the documents were collected in Figure 1. However, it would be more helpful to display the number of documents collected from each region (e.g., a visual based on the data used to generate table S1) to understand the composition of the dataset better. This concern applies across all data presented - n (number of documents) should be provided for each bar/category in each figure/table, or in legends/captions, as appropriate.

A potential weakness is the crowd-sourced nature of the collected documents which may be biased by the membership of the GYA and doesn't necessarily include all countries with eligible academic institutions. Another concern is the lack of stratification of the sample - some regions/income levels are likely overrepresented, and others underrepresented. This limits the ability to conclude about the truly global state of academic promotion practices.

Can the authors further clarify the implications of stratification by career tracks in the method section? Perhaps examples would be helpful.

What types of national documents were used? The description in the method section merely states the use of published documents - can examples be given?

Document analysis: From what method/process did the mapped 18 categories/4 groups originate? Interesting that teaching

is represented as a singular category whereas research and service are broken down into numerous detailed sub-categories - is there a reason for grouping all teaching themes (and even placing teaching within service)? Teaching often comprises the majority of the academic workload, despite its value usually weighted less in academic promotion documents.

Where possible, it would improve the quality of the manuscript to include display of statistically significant findings (as shown in tables in the supplementary information) as symbols on the figures in the manuscript itself (e.g., Table S9 <-> Figure 5) and/or written in the text of the article.

Thoughts on Conclusions

Point 1: It seems that differences in insitutional requirements may also produce situations in which researcher mobility is limited (i.e., research priorities and valued achievements at current university may not be valued at a different university). More emphasis on the idea that evaluation committees generally have flexibility in their consideration beyond what is listed in the documentation is warranted.

It is concerning that 44% of documents in this study did refer specifically to journal impact factor when impact factor is such a poor metric for research quality - this might be emphasized as a greater concern.

Point 2: What factors do the authors think cause higher income countries to rely less on scientometrics? Perhaps consider the increased labour required for qualitative analysis.

Point 3: Do the authors have any comments regarding the effects of national vs. institutional policies being out of alignment with each other in both Global North and Global South? What impact does this have on institutions and researchers?

Point 4: Also the JIF was not created for the purpose of evaluating the impact of individual articles:

[https://www.frontiersin.org/articles/10.3389/frma.2018.00002/full#:~:text=The journal impact factor \(JIF,known invention of Eugene Garfield](https://www.frontiersin.org/articles/10.3389/frma.2018.00002/full#:~:text=The journal impact factor (JIF,known invention of Eugene Garfield)

Also, this may be a good section in which to emphasize the value of regional/community engagement and acknowledge the variability this may produce in scientometrics despite increasing real-world impact of the research.

Although the Global North can perhaps "afford" an inefficient research ecosystem, it is still wasteful and inequitable and should also be addressed with change. It may be worth proposing a global initiative (inclusive of North and South, and all levels of income) rather than placing responsibility on the Global South to lead the way. Perhaps such a global initiative can be flexible to allow for regional strengths to shine and meaningful collaborations across regions and income levels to flourish. The authors may make practical suggestions for bodies already working towards these ideals (as mentioned in the introduction), based on the findings of this study.

Version 2:

Reviewer comments:

Referee #1

(Remarks to the Author)

I want to thank the authors for the responses to the earlier comments and the revised manuscript.

I have attached some notes to the peer reviewer rebuttal document.

Here are a few additional comments.

The authors may want to revisit the title of the paper. Given that very close to 75% of the documents are from non-European and North American jurisdictions, perhaps a title more reflective of this sampling frame would be more declarative to readers.

I think some readers will not well understand the idea of government policies concerning promotion and tenure. For example, in Most of Canada and the United States, universities (or other research institutions) would be making these decisions. Governments would not.

There are many authors of the paper. Are most of the authors included because they were able to access the documentation (at their respective university) or did they contribute in other ways? This information might be best achieved using the CREDIT system <https://credit.niso.org/> for authorship.

Referee #2

(Remarks to the Author)

With apologies for the delay in reviewing this second version of the study, to which I wanted to give my undivided attention.

I am quite satisfied with the revisions by the authors. I was initially concerned that a simple edit would not be sufficient, but the additional data collected and analysis presented do a fine job of speaking to my major concerns. Most notably, the collection of additional documents to allow you to disentangle the national and institutional policy analysis from the Global South and North analysis. The additional methodological details provided, such as the breakdown of document types and the check of whether policies are for full professor only were particularly helpful.

Just as importantly, the edits with additional methodological details, nuanced phrasing, and explicit acknowledgement of the limitations offer greater confidence in the results.

Referee #3

(Remarks to the Author)

The authors have significantly updated their manuscript to incorporate additional data, information, and changes consistent with the requests of the reviewers.

The method used for data collection and analysis is clarified and appears to now be transparent.

I appreciate the substantial efforts of the authors.

Major comments:

1. The weighting of data by number of FTE researchers per country makes sense and is an improvement compared to the original analysis because it more proportionately represents the policies influencing researchers across the globe. However, it is now clear that a handful of high-population countries in both Global North and Global South are dominating the analysis due to their researcher numbers dwarfing those of many other countries. For instance, countries with 1+ million researchers (e.g., USA, China) will have three orders of magnitude the influence on the reported outcomes compared to countries with <1000 researchers (e.g., much of Africa). Thus, the presented data seem to make a potential waste of going to the effort of collecting policies from the (many) countries with relatively few researchers. Perhaps the authors would like to think about additional analyses or methods to report on the policies affecting these smaller populations of researchers.

Minor comments:

1. There should be a citation in the caption for Figure 1, and within lines 131-134, to identify the source of the data on number of researchers within each country as used for weighting of the policy data.
2. Should mention in the article about leaving out Art and Creative Works as a discipline.
3. Some technical analysis description in lines 184-191 may be simplified and details left to the supplementary section for improved readability.
4. Figure 5 caption - list reference categories in same order as bars are given top to bottom to aid readability. Or, overlay reference category directly on the chart at the top of each set of bars.

Version 3:

Reviewer comments:

Referee #1

(Remarks to the Author)

I am still concerned about the methods. Without robust methods, that are replicable, the results become tenuous.

There is still no reporting of whether a protocol exists. This is an example of what a research protocol is ([https://icahn.mssm.edu/files/ISMMS/Assets/Research/IHCDS/Guidelines for Writing the research protocol by WHO.pdf](https://icahn.mssm.edu/files/ISMMS/Assets/Research/IHCDS/Guidelines%20for%20Writing%20the%20research%20protocol%20by%20WHO.pdf)). Research protocols are important because they allow peer reviewers, and others, to examine for the possibility of reporting biases (<https://www.bmj.com/content/362/bmj.k3802>) in completed research. While the example provided is from randomized trials, reporting biases are common in other types of research.

The authors start the methods section with data collection & analysis. Typically, readers like to read something about the study design. It looks like the authors have conducted a cross-sectional study. If I'm correct, this information should be included in the methods section. Similarly, what were the eligibility criteria for selecting documents.

How were the documents accessed? What processes were used. A few sentences describing the process in the methods section would be of value to readers and others interested in replicating what the authors did.

The sampling time – May 2016 to November 2023. Were some documents sampled in 2016 and others in 2023? A document sampled in 2016 might be out of date in 2023. This is particularly relevant given the prominence in researcher assessment over the last 5 years.

As best as I can tell the authors developed 18 categories and 11 sub-categories for mapping against the documents/policies. As a reader it would be invaluable to know how the categories and sub-categories were developed? Are they evidence-based?

It looks like each document was reviewed by members of the author team. Did two people review each document and come to some form of agreement? If so, how was this done and can the authors report measures of agreement?

Referee #3

(Remarks to the Author)

The authors addressed my major concern by conducting a robustness check of their results in which the data from all countries except the 10 with the most researchers were analyzed. These new analyses are reported in the Supplementary Information, and in two sentences added to the "Data Collection and Analysis" section of the manuscript:

"However, in a robustness check described in the SI (section 2.6), we find that our main results are replicated in a subsample of policies that refers to the remaining 28% of the population of researchers. Thereby, our global picture is also a reasonable description with regard to geographical spread of countries and less skewed than the population distribution."

I appreciate the efforts of the authors in conducting this additional analysis. I wonder if the text included in the main text of the manuscript overreaches in terms of interpretation. A comparison of significant findings in Table S13 (all data) versus Table S16 (restricted robustness check data) shows that approximately 1/3 of the reported regression results differ between the tables. It may be more transparent to revise the text in the manuscript to acknowledge not only the similarities, but also that there are detailed differences described in the Supplementary Information, so readers are not led to falsely assume the robustness check exactly replicated the full analysis.

Otherwise, for future analyses, it may be worth reconsidering the approach of removing the top 10 countries. Perhaps a different method of selection could be justified more meaningfully.

Version 4:

Reviewer comments:

Referee #1

(Remarks to the Author)

Please see attached document.

Referee #3

(Remarks to the Author)

The additional details regarding protocol, analysis, and results that were added to the manuscript and supplementary information have increased the clarity and transparency of the reported research.

Regional and institutional trends in assessment for academic promotion

Corresponding Author: Ms Yensi Flores Bueso

Version 1:

Reviewer comments:

Referee #1

(Remarks to the Author)
Please see attached document.

Referee #2

(Remarks to the Author)

Thank you for the opportunity to review the manuscript “A global assessment of academic promotion criteria: What really counts?”, which I read with great interest. The study offers an examination of promotion criteria to full professor (or equivalent) written into national and institutional policies and guidelines from 55 countries. It is, to my knowledge, the most global study of its kind and presents valuable evidence over which assessment criteria are codified into policies and guidelines. The evidence presented largely affirms what is expected from the literature and from the common understanding of promotion criteria while providing an empirical basis for those common assertions. Most notable, for me, is the evidence highlight that the Global South is more likely to use number of publications and citation metrics than the Global North. While aligning with expectations, this study provides solid evidence on the matter.

While there is much to praise from the study, there are also a handful of areas where further clarity is needed and should be provided ahead of publication. These areas are all connected to the wide range of documents that were surely collected and the similarly wide-ranging set of academic and researcher systems to which they apply. While the authors acknowledge that “the structure and style of the policy documents varied substantially,” they provide little description of this variability in relations to the dimensions analyzed. For example, were certain kind of documents more prevalent in some countries than others?

Similarly, the collection of both institutional and national policies is a strength of the study, but the lack of analysis of where national policies were more prominent makes it difficult to disentangle the national/institutional analysis from the Global North/South analysis. This relates to a broader issue of the diversity of systems that exist for researchers around the world (how were equivalencies of full professor found for systems around the world?). Was there any consideration for policies from countries where researchers hold dual affiliations (one with the national system and another within in their institution)? While the manuscript acknowledges that such differences exist (similarly with different career tracks), the study design, analysis, or discussion fails to take them into account. I also found no mention of how many of the documents found could be cleanly mapped to particular career stages/tracks. In my experience, there is a mix of documents that specify career stages/tacks and those that do not.

Again, similarly, there is little detail provided on how much overlap there is between institutional and national level policies, or how much overlap there is between institutional policies (i.e., multiple documents from the same institution, potentially for different disciplines or different academic units). Was having a single document from a specific academic unit enough to include that country in the country-level analysis?

In short, the study is lacking a more completely explanation of how the differences in academic careers and the diversity of

documents were handled. Both of these elements should be expanded upon in Methods and in the Discussion. The differences in academic careers across countries, especially the extent to which academic career progressions can (and cannot!) be compared, should be addressed in the literature review and again in Point 1 of the discussion (noting, for example, that researcher mobility varies greatly by country). (For what it's worth, it was this complexity that held us back from analysing documents globally and made us decide to focus on the US and Canada, where the Review, Tenure, and Promotion systems are largely comparable).

For the most part, I don't think much of the above will affect the result of the analysis, but it would add some detail and present some limitations and caveats that should be addressed. The one area where, I suspect, it might matter the most is in the disciplinary analysis. Where those disciplinary documents are sourced from is likely to matter and some checks should be done to describe the geographic representation of that subset of documents. (Any further detail should be considered in Point 4 of the discussion).

A few other comments to consider:

- The analysis of "outputs" lacks detail on the types of outputs that are being solicited. Are only traditional outputs considered? In our own work (Alperin et al., 2022), we found 127 different types of outputs listed. While I'm not suggesting authors use this level of categorization, it might be helpful to note which types of outputs were included.
- The estimate of number of researchers covered by the study seems inflated. If documents were selected for those that apply promotion to (full) professor rank, then using the more generic "researcher per million" value as an approximation is not accurate. This speaks to the point raised above about the need for further details and discussion on career stages.
- I did not find much value in the flow described in Figure 1 (this flow is described in the text and is sufficiently obvious as to not warrant a visualization). The map is more helpful, although perhaps it could be coupled with further details on number of documents by type, as requested above.
- Section 4 of discussion: possible typo: "Communalities" "Commonalities"?
- I could see an argument made that the latter half of the discussion (after the five summary points) is not sufficiently rooted in the study results or the literature. Perhaps because I find myself agreeing with the perspective provided, I found the section to be a helpful conclusion to the article, but I would not be opposed for further linkages to the study findings and to previous literature.

Despite the questions and comments raised, I believe this to be a valuable study. I find the factor analysis to be a useful way of adding some depth to binary indicators and the conclusions drawn to be reasonably drawn from that analysis. A study of this nature will always have limitations owing to the impossibility of neatly capturing the global diversity in research systems and cultures. I appreciate the ambition of the study and believe that the scholarly community will benefit from being presented with this global comparative view, despite its minor shortcomings.

Alperin, J.P., Schimanski, L.A., La, M., Niles, M.T., & McKiernan, E.C. (2022). The value of data and other non-traditional scholarly outputs in academic review, promotion, and tenure. In *Open Handbook of Linguistic Data Management*. MIT Press

Referee #3

(Remarks to the Author)

The most significant contribution of this study is the global nature of the analysis allowing for comparisons regarding review for academic promotion to be made between Global North vs. South and between countries with different income levels. Similar analyses have been conducted in the past but have not comprehensively analyzed policy/procedure from across the globe. Also interesting are comparisons between institutional level and national policies governing assessment of research outcomes. It is interesting for researchers to gain understanding about what is valued in different regions, countries, and levels of administration. The insights gained have potential to guide efforts toward creating greater consistency in academic procedures globally, and may inspire leaders to apply approaches different from those commonly employed in their own region.

It is helpful that the authors displayed the number of countries from which the documents were collected in Figure 1. However, it would be more helpful to display the number of documents collected from each region (e.g., a visual based on the data used to generate table S1) to understand the composition of the dataset better. This concern applies across all data presented - n (number of documents) should be provided for each bar/category in each figure/table, or in legends/captions, as appropriate.

A potential weakness is the crowd-sourced nature of the collected documents which may be biased by the membership of the GYA and doesn't necessarily include all countries with eligible academic institutions. Another concern is the lack of stratification of the sample - some regions/income levels are likely overrepresented, and others underrepresented. This limits the ability to conclude about the truly global state of academic promotion practices.

Can the authors further clarify the implications of stratification by career tracks in the method section? Perhaps examples would be helpful.

What types of national documents were used? The description in the method section merely states the use of published documents - can examples be given?

Document analysis: From what method/process did the mapped 18 categories/4 groups originate? Interesting that teaching

is represented as a singular category whereas research and service are broken down into numerous detailed sub-categories - is there a reason for grouping all teaching themes (and even placing teaching within service)? Teaching often comprises the majority of the academic workload, despite its value usually weighted less in academic promotion documents.

Where possible, it would improve the quality of the manuscript to include display of statistically significant findings (as shown in tables in the supplementary information) as symbols on the figures in the manuscript itself (e.g., Table S9 <-> Figure 5) and/or written in the text of the article.

Thoughts on Conclusions

Point 1: It seems that differences in insitutional requirements may also produce situations in which researcher mobility is limited (i.e., research priorities and valued achievements at current university may not be valued at a different university). More emphasis on the idea that evaluation committees generally have flexibility in their consideration beyond what is listed in the documentation is warranted.

It is concerning that 44% of documents in this study did refer specifically to journal impact factor when impact factor is such a poor metric for research quality - this might be emphasized as a greater concern.

Point 2: What factors do the authors think cause higher income countries to rely less on scientometrics? Perhaps consider the increased labour required for qualitative analysis.

Point 3: Do the authors have any comments regarding the effects of national vs. institutional policies being out of alignment with each other in both Global North and Global South? What impact does this have on institutions and researchers?

Point 4: Also the JIF was not created for the purpose of evaluating the impact of individual articles:

[https://www.frontiersin.org/articles/10.3389/frma.2018.00002/full#:~:text=The journal impact factor \(JIF,known invention of Eugene Garfield](https://www.frontiersin.org/articles/10.3389/frma.2018.00002/full#:~:text=The journal impact factor (JIF,known invention of Eugene Garfield)

Also, this may be a good section in which to emphasize the value of regional/community engagement and acknowledge the variability this may produce in scientometrics despite increasing real-world impact of the research.

Although the Global North can perhaps "afford" an inefficient research ecosystem, it is still wasteful and inequitable and should also be addressed with change. It may be worth proposing a global initiative (inclusive of North and South, and all levels of income) rather than placing responsibility on the Global South to lead the way. Perhaps such a global initiative can be flexible to allow for regional strengths to shine and meaningful collaborations across regions and income levels to flourish. The authors may make practical suggestions for bodies already working towards these ideals (as mentioned in the introduction), based on the findings of this study.

Version 2:

Reviewer comments:

Referee #1

(Remarks to the Author)

I want to thank the authors for the responses to the earlier comments and the revised manuscript.

I have attached some notes to the peer reviewer rebuttal document.

Here are a few additional comments.

The authors may want to revisit the title of the paper. Given that very close to 75% of the documents are from non-European and North American jurisdictions, perhaps a title more reflective of this sampling frame would be more declarative to readers.

I think some readers will not well understand the idea of government policies concerning promotion and tenure. For example, in Most of Canada and the United States, universities (or other research institutions) would be making these decisions. Governments would not.

There are many authors of the paper. Are most of the authors included because they were able to access the documentation (at their respective university) or did they contribute in other ways? This information might be best achieved using the CREDIT system <https://credit.niso.org/> for authorship.

Referee #2

(Remarks to the Author)

With apologies for the delay in reviewing this second version of the study, to which I wanted to give my undivided attention.

I am quite satisfied with the revisions by the authors. I was initially concerned that a simple edit would not be sufficient, but the additional data collected and analysis presented do a fine job of speaking to my major concerns. Most notably, the collection of additional documents to allow you to disentangle the national and institutional policy analysis from the Global South and North analysis. The additional methodological details provided, such as the breakdown of document types and the check of whether policies are for full professor only were particularly helpful.

Just as importantly, the edits with additional methodological details, nuanced phrasing, and explicit acknowledgement of the limitations offer greater confidence in the results.

Referee #3

(Remarks to the Author)

The authors have significantly updated their manuscript to incorporate additional data, information, and changes consistent with the requests of the reviewers.

The method used for data collection and analysis is clarified and appears to now be transparent.

I appreciate the substantial efforts of the authors.

Major comments:

1. The weighting of data by number of FTE researchers per country makes sense and is an improvement compared to the original analysis because it more proportionately represents the policies influencing researchers across the globe. However, it is now clear that a handful of high-population countries in both Global North and Global South are dominating the analysis due to their researcher numbers dwarfing those of many other countries. For instance, countries with 1+ million researchers (e.g., USA, China) will have three orders of magnitude the influence on the reported outcomes compared to countries with <1000 researchers (e.g., much of Africa). Thus, the presented data seem to make a potential waste of going to the effort of collecting policies from the (many) countries with relatively few researchers. Perhaps the authors would like to think about additional analyses or methods to report on the policies affecting these smaller populations of researchers.

Minor comments:

1. There should be a citation in the caption for Figure 1, and within lines 131-134, to identify the source of the data on number of researchers within each country as used for weighting of the policy data.
2. Should mention in the article about leaving out Art and Creative Works as a discipline.
3. Some technical analysis description in lines 184-191 may be simplified and details left to the supplementary section for improved readability.
4. Figure 5 caption - list reference categories in same order as bars are given top to bottom to aid readability. Or, overlay reference category directly on the chart at the top of each set of bars.

Version 3:

Reviewer comments:

Referee #1

(Remarks to the Author)

I am still concerned about the methods. Without robust methods, that are replicable, the results become tenuous.

There is still no reporting of whether a protocol exists. This is an example of what a research protocol is ([https://icahn.mssm.edu/files/ISMMS/Assets/Research/IHCDS/Guidelines for Writing the research protocol by WHO.pdf](https://icahn.mssm.edu/files/ISMMS/Assets/Research/IHCDS/Guidelines%20for%20Writing%20the%20research%20protocol%20by%20WHO.pdf)). Research protocols are important because they allow peer reviewers, and others, to examine for the possibility of reporting biases (<https://www.bmj.com/content/362/bmj.k3802>) in completed research. While the example provided is from randomized trials, reporting biases are common in other types of research.

The authors start the methods section with data collection & analysis. Typically, readers like to read something about the study design. It looks like the authors have conducted a cross-sectional study. If I'm correct, this information should be included in the methods section. Similarly, what were the eligibility criteria for selecting documents.

How were the documents accessed? What processes were used. A few sentences describing the process in the methods section would be of value to readers and others interested in replicating what the authors did.

The sampling time – May 2016 to November 2023. Were some documents sampled in 2016 and others in 2023? A document sampled in 2016 might be out of date in 2023. This is particularly relevant given the prominence in researcher assessment over the last 5 years.

As best as I can tell the authors developed 18 categories and 11 sub-categories for mapping against the documents/policies. As a reader it would be invaluable to know how the categories and sub-categories were developed? Are they evidence-based?

It looks like each document was reviewed by members of the author team. Did two people review each document and come to some form of agreement? If so, how was this done and can the authors report measures of agreement?

Referee #3

(Remarks to the Author)

The authors addressed my major concern by conducting a robustness check of their results in which the data from all countries except the 10 with the most researchers were analyzed. These new analyses are reported in the Supplementary Information, and in two sentences added to the "Data Collection and Analysis" section of the manuscript:

"However, in a robustness check described in the SI (section 2.6), we find that our main results are replicated in a subsample of policies that refers to the remaining 28% of the population of researchers. Thereby, our global picture is also a reasonable description with regard to geographical spread of countries and less skewed than the population distribution."

I appreciate the efforts of the authors in conducting this additional analysis. I wonder if the text included in the main text of the manuscript overreaches in terms of interpretation. A comparison of significant findings in Table S13 (all data) versus Table S16 (restricted robustness check data) shows that approximately 1/3 of the reported regression results differ between the tables. It may be more transparent to revise the text in the manuscript to acknowledge not only the similarities, but also that there are detailed differences described in the Supplementary Information, so readers are not led to falsely assume the robustness check exactly replicated the full analysis.

Otherwise, for future analyses, it may be worth reconsidering the approach of removing the top 10 countries. Perhaps a different method of selection could be justified more meaningfully.

Version 4:

Reviewer comments:

Referee #1

(Remarks to the Author)

Please see attached document.

Referee #3

(Remarks to the Author)

The additional details regarding protocol, analysis, and results that were added to the manuscript and supplementary information have increased the clarity and transparency of the reported research.

RESPONSE TO REVIEWERS

Editor's comment

Your manuscript, "A global assessment of academic promotion criteria: What really counts?", has now been seen by 3 referees, whose comments are attached below. While they find your work of potential interest, as do we, they have raised important concerns that in our view need to be addressed before we can consider publication in Nature. More specifically, they are concerned that the sampling may be biased (as you only include institutions that are GYA members) and that you neither look at different career tracks or provide sufficient methodological details to assess the strength of the results.

Should you be able to show that the GYA member institutions do not reflect a biased sample, include policies for different career tracks, and have the power to disentangle the effect of policies (national vs academic) from location, we would be happy to consider a revised manuscript (unless something similar has been accepted at Nature or appeared elsewhere in the meantime).

We are thankful for the opportunity to present this revised manuscript and for the constructive feedback provided by the reviewers and editor. We have given your advice thorough consideration and implemented your valuable suggestions in order to strengthen our work. We believe that you will find that this revised version of the manuscript adequately addresses all of your requests and resolves any concerns regarding our sample and methodological approach. Among the changes made, are:

- 1) We undertook a major data collection effort oriented at updating and extending the global representation of our sample, resulting in the incorporation of 419 policies into our study. Our sample now includes research institutions from 121 of the 195 countries worldwide, mitigating any potential biases in our sample selection and achieving a wider global representation. This extended sample enabled us to :
 - Run analyses of the power for the hypothesis tests reported. Since the global population of assessment policies is unknown, these tests are based on the observed mean values and standard deviations in our sample. The results are indicative of very high levels of power (above 95% in all relevant cases), please refer to Supplementary Information (SI) Sect. 2.3.
 - Significantly strengthened our analysis, providing a robust foundation to substantiate our claims. (Please refer to sect. 2.1 and 2.2 in the SI for detailed insights)
 - Conduct a multivariate regression analysis to disentangle the combined effects of multiple independent variables, allowing for a more comprehensive understanding of underlying dynamics.(Please see section 2.4 of the SI, L 226 – 254 and Figure 5 of the results).
 - Evaluate the influence of different career tracks in the choice of criteria evaluated. As seen in Fig 5 of results and SI sections 1.2.3. and Table S1.

- Ensured that the large majority (>85%) of the policies included in the study are the most recent versions and are still applicable. We have integrated this information into our dataset. See L: 40-47 in the SI sect 1.1.
 - We sourced relevant information suggested by reviewers, such as the evaluation of interdisciplinarity and ethics as assessment criteria, and whether a policy is exclusive for the role of full professor (see SI Table S2 in sect. 1.3.1, and all the Figures in the results).
- 2) In response to the observations made about the weaknesses in our methodology, we conducted a thorough review of our processes of data coding, implemented controls, and gathered additional data, which has notably improved the transparency, reproducibility and reusability of the data in our study. For example:
- Each criterion in the dataset now includes the evidence (in an adjacent cell) indicating the rationale for considering its presence, providing future users with a better understanding of the basis for scoring.
 - Policies underwent review by a minimum of three authors, as stipulated and detailed in the database (under the reviewer columns) .
 - The dataset provides URLs for over 80% of the policies collected, which were available online, alongside document attributes such as sourcing date, implementation date, and the latest year the policy was known to be active (see L: 41-48 in SI sect1.1).
 - We are sharing annotated code developed for our analyses, as well as the raw outputs of these analyses.
- 3) The methodology underwent a thorough rewrite, encompassing all details of each step of our processes.
- 4) We made significant revisions to our data analysis methods to enhance their representativeness, reproducibility, and impartiality (more in Supplementary Information, section 1.7). Among the key changes implemented:
- We introduced an intra institutional weighting system to ensure that each institution, regardless of the number of policies it contains, is assigned a value of 1 (see SI section 1.6).
 - We incorporated a global weighting system, calculated based on the number of researchers in full-time equivalents residing in each country included in the study. This approach enables us to refer our results to the population of potential job candidates affected (see SI section 1.6).
 - We minimized interventions in data categorization and stratification, which we thoroughly describe in the SI section 1.5 to maintain transparency and consistency throughout the analysis process.

Referee #1 (Remarks to the Author):

1. I do find some irony in the fact that this paper is limited to policies/documents for researcher assessments to full professorship carried out by the Global Young Academy. Wouldn't early career researchers be most interested in policy/assessment for hiring and early career progression (e.g., to associate professor)?

We thank the reviewer for raising this point. We now include the rationale for this decision in paragraph 1 of the data collection and analysis (Lines 84 – 94). In summary, we adopted this profile as it provides us with a basic role that can be meaningfully compared across various countries. While defining this study, we identified that the most senior academic position is more clearly defined and relatively similar internationally, whereas the entry- and mid-level positions are more differentiated across countries and sometimes even across institutions in a single country, with career pathways differing substantially. For example, “associate professor” is predominantly a US term and e.g. in Germany such a career profile does not exist. This topic was also addressed by Reviewer #2, albeit from a different perspective, underscoring the high complexity of comparing academic career progression. Moreover, the aspiration for many if not all ECRs is to one day ascend to the highest ranks of the profession, and this affects their incentives and potentially their behaviour already at the early stages of their career.

2. My main concerns about the paper are about the methodology.

We thank the reviewer for bringing this concern to our attention. This has been well noted, and we have extensively revised the methods to make them comprehensive. Please refer to the revised methodology described in the Supplementary Information.

3. Did the authors work from a written protocol?

Thank you for raising this point. Given the novelty of the project and the diversity of our sample, there were no studies or protocols that could meet our needs, and thus we empirically developed and tested different approaches at the outset of the study. Through this experience, we were able to standardise modes of practice in terms of data collection and tabulation. Similarly, we used the most statistically sound approaches in terms of data analysis. This is now thoroughly explained in our fundamentally rewritten methodology in Supplementary Information (SI) sect 1.

4. How were the documents accessed? What processes were used. For example, our team has just completed identifying hundreds of documents that mention policies/templates related to data sharing. This will likely require a few sentences describing the process in the methods section of our paper.

Thank you for bringing this to our attention. We have now included a section providing all these details in section 1.1 of the methodology (L34 - 40), including the following: “. We consider each distinct set of assessment criteria applied for a promotion process to constitute a “policy” for promotion. Some institutions or agencies distinguish different career tracks or academic disciplines, and therefore there can be several policies within the same institution or agency. In total, our dataset covers 532 policies of which 427 policies (80%) were sourced from public websites, 105 policies (20%) were sourced through the co-authors’ networks and 6 policies (1%) were obtained via an official request made through the GYA office. We share the respective URLs of policies found online in our dataset. .”

5. I think Figure 1 (Methods outline) insufficient. It does not allow interested readers to replicate the methods. Please provide more detailed methods. The results are only trustworthy if the methods are rigorous, comprehensive, and transparent.

Thank you for this remark. We have removed this figure and provided more substantial detail in the methodology section. Please refer to the Supplementary Information Sect 1.

6. How were the documents sampled. Where the documents taken from countries the Global Young Academy member was living in, some sort of purposive sampling and something else. Clarity of this issue is important.

Thank you for pointing this out, we have now included a detailed section of our snowballing sampling approach, in Section 1.1 of the Supplementary Information L10-19: *"We leveraged on members and alumni of the Global Young Academy (GYA) to source documents not only from their own institutions, but in a way of snowball sampling also from other institutions they know or have a relation with, or by involving their academic network.."*. While members of the Global Young Academy (GYA) played a crucial role in sourcing the documents (knowledge of regional infrastructure and languages), those are not restricted to their own institutions. We are happy to report that we have greatly increased our coverage across the globe with a further sourcing effort, we have updated all policy documents that were in the original dataset, and we applied post-sampling weighting, L 28-31: *"..Our sampling specifically aimed at maximising the reach and scope, so that our sample is as wide and diversified as possible, in terms of countries covered, areas of the world, and economic status. Consequently, as highlighted in the main text, we obtained the largest data set, to our knowledge, of promotion criteria globally to date."*.

7. The sampling time – May 2016 to July 2021. Were some documents sampled in 2016 and others in 2021? A document sampled in 2016 might be out of date in 2023. This is particularly relevant given the prominence in researcher assessment over the last 5 years. Did the authors update their sampling to be current as of the middle of 2023?

Following the reviewers advice in this matter we performed an extensive round of review of our sampling, which concluded in the update of all policies. This is now expanded in Section 1.1, L 40-47: *"...drove a second round of sourcing in 2023, to check for updates or new policies. From this exercise, we sourced 440 policies, of which 300 (56%) were implemented after 2020, 148 (28%) after 2015, 62 (12%), before 2015 and for 22 (4%) policies this information was not disclosed."* Further, for the small fraction of policies that we could not confirm whether they are still applicable, we wish to point out that we have observed that policy and institutional change is typically rather slow, and while we agree with the reviewer, that change in the practices at the global level is moving forward, this does not necessarily apply at the level of the single institutions or even single countries, which are the units of our analysis. With this in mind, we believe that our dataset is up to date and that to our knowledge, most of the policies are still applicable.

8. Be more precise throughout the document. For example, line 141, the authors state "About 60% of these policies". This is a little vague.

We thank the reviewer for highlighting this. We now have rephrased this to: *"Most of these institutions and agencies (73%) are based"* in Line 100 of the manuscript. We took notice and have refrained from vague language. However, we have refrained from providing decimal points in the findings to avoid confusing or possibly misleading the reader, as the hypothesis tests show that small differences in our results are not statistically significant. The differences that we comment upon are typically of a different

order of magnitude (10% or more) and indeed, our powers analysis shown in section 2.3 of the SI confirm that at this order of magnitude our tests have sufficient power.

9. As best as I can tell the authors developed 18 categories and 11 sub-categories for mapping against the documents/policies. As a reader it would be invaluable to know how the categories and sub-categories were developed? Are they evidence-based?

Thank you for highlighting this. We have now included a section in the SI (sections: “1.3.1. Definition of criteria” and “1.3.2. Categorisation of criteria”) expanding this point, see lines 156 -180 and tables S3 and S4, For example, L169-171: *“At the onset of this project, we performed an empirical exercise to understand and develop a method to quantitatively capture the qualitative data embedded in a policy. Through this exercise, we identified common keywords or clusters of keywords relating to similar criteria”*. L182-171: *“We found that the majority of policies are explicitly structured around the three broad categories of “research”, “teaching”, and “service”... ”*

10. Where any of the categories related to open science or equity, diversity, and inclusiveness? I’m I correct in assuming neither domains were examined?

While we acknowledge the importance of these topics, there were several limitations that prevented us from including these categories in our study. This is primarily due to the nature of these topics, which predominantly relate to the process of hiring or promotion (e.g. non-discrimination etc.), and to a smaller extent to the criteria evaluating the single candidates or their work. Since our analysis is based solely on policy criteria, we wouldn’t be able to comprehensively capture this information.

Furthermore the criteria evaluating these subjects are highly complex and globally inhomogeneous. We concluded that performing statistical analysis on criteria that have very different meanings across different national or regional policy frameworks would likely be more confusing than helpful and therefore of little value. We have now clarified this stance and rationale in the first paragraph of the Data Collection and analysis section, lines 123-130: *“Our analysis is solely based on the presence or absence of these criteria, we are not considering the assessment process. Specifically, we did not consider the role or weight of criteria, nor did we capture how assessors interpret the policies, and how an assessment panel ultimately arrives at its decision, potentially taking into account further criteria that are not explicitly mentioned in the policies”* .

To expand on this, properly capturing the relation between open science factors in promotion criteria with national and regional open science policy frameworks, benefits to various actors, and a globally equitable open science system warrants its own study, and goes much beyond capturing key commonalities and differences and arising implications for researchers, research managers, and national governments, as discussed here. One of the authors of this study explicitly discussed global policy divergence and misalignment with global goals (DOI: 10.5334/dsj-2022-001). Open Science must not be reduced to a box-ticking exercise to meet policy requirements. Similarly, equity, diversity, and inclusiveness are crucial goals of science policy and we are aware that research evaluation can impact them (for example another author has personally worked on this topic: <https://doi.org/10.1016/j.respol.2019.103820>). However, more than any other criterion covered in our study, these policy goals take different meanings in various national contexts, sometimes in response to

specific shortcomings and/or meeting national legal requirements, with incompatibilities across countries. We elaborate on this in the SI.

- 11.** It looks like each document was reviewed by members of the author team. Did two authors review each document and come to some form of agreement? If so, how was this done and can the authors report measures of agreement?

Thank you for bringing this to our attention, we have expanded on the process of data capture in section 1.5 of the SI L 266-281. In brief, we adopted several measures to ensure consistency on the interpretation, including the provision of the evidence that led to a scoring decision and a thorough procedure involving a two-stage verification process in which each document was separately reviewed by at least two co-authors. Whenever disagreements arose, a 3rd author or more (depending on the level of disagreement) were involved to review the text and reach an agreement. In those cases, we documented the incident by including the decision into the definition of the criteria (Shown in Table S2), so that it was available to other authors working parallelly for future reference.

- 12.** The authors outline 5 strategies. On line 302, the authors state "...should be valuable by institutions and researchers ...". Do the authors mean 'valued'?

Thank you for spotting this, we have changed this and the sentence now reads: "*We anticipate that our findings will prove valuable to both researchers and research managers for understanding career options and opportunities, and provide guidance on how to engage in building a strong research ecosystem that embraces diversity amongst responsible actors who contribute efficiently with their various strengths.*"

- 13.** The authors do not appear to propose any option for 'promotion criteria are not'.

We apologise to the reviewer, but we failed to understand what the concrete suggestion is. We asked several members of the team and could not reach a sound consensus interpretation. Therefore, we felt unable to address this point.

- 14.** The authors appear surprised by the use of quantitative metrics used by lower-income countries. Might this be because the resources (fiscal and qualitative expertise) required to implement more qualitative assessments are not available? Even in my higher income country, our university has limited expertise in the qualitative methods experts.

Thank you, as it turns out, some of the authors were surprised and others less so. In this revised manuscript, we have a substantially larger dataset that allowed us to run multivariate analysis. This now shows that it is not generically the global south to rely more on metrics, but rather specific cases such as upper-middle income countries. Our original claim has now been specified.

However, it remains true that, descriptively, metrics are more frequently adopted in the Global South than in the Global North. While we agree with the reviewer's suggestion about resources being a potential underlying reason, we did not include this in the discussion as we didn't find tangible evidence supporting this hypothesis and we found alternative explanations, such as that introduced by the "Scoping Group report" of the International Academies Partnership, which suggests that institutions in these countries may have "learned the game" and are actively trying to climb the rankings, although

they perceive that higher-income countries are currently changing the rules of the game by moving towards qualitative criteria. We hope that this study provides a basis by which future research can disentangle this See: <https://www.interacademies.org/publication/future-research-evaluation-synthesis-current-debates-and-developments>).

15. A limitation section would be valuable for readers.

Thank you for the suggestion, we have now included upfront limitations of our study in the main text (first paragraph on data collection, lines 84 -94) and we have rewritten the methods (SI Section 1) to contextualise our methodology, providing the rationale for approaches taken and highlighting any limitations in an upfront manner. We believe that this will be liked by the reviewer as it will enhance the clarity for the reader, preventing any misinterpretations or misguided expectations.

Referee #2 (Remarks to the Author):

Thank you for the opportunity to review the manuscript “A global assessment of academic promotion criteria: What really counts?”, which I read with great interest. The study offers an examination of promotion criteria to full professor (or equivalent) written into national and institutional policies and guidelines from 55 countries. It is, to my knowledge, the most global study of its kind and presents valuable evidence over which assessment criteria are codified into policies and guidelines. The evidence presented largely affirms what is expected from the literature and from the common understanding of promotion criteria while providing an empirical basis for those common assertions. Most notable, for me, is the evidence highlighting that the Global South is more likely to use number of publications and citation metrics than the Global North. While aligning with expectations, this study provides solid evidence on the matter.

Thank you. We appreciate your perspective, appreciation of the study and suggestions.

1. While there is much to praise from the study, there are also a handful of areas where further clarity is needed and should be provided ahead of publication. These areas are all connected to the wide range of documents that were surely collected and the similarly wide-ranging set of academic and researcher systems to which they apply. While the authors acknowledge that “the structure and style of the policy documents varied substantially,” they provide little description of this variability in relations to the dimensions analyzed. For example, were certain kinds of documents more prevalent in some countries than others?

We thank the reviewer for the constructive feedback. In response to this suggestion, we have now captured and analysed policy document features. Here, we surveyed the types of documents encountered across different policies, noted whether they were exclusively specified for the full professor role and their level of detail(see Table S5). In this way, we wish to provide a better understanding of the general structural differences that may exist between policies. Additionally, we performed an analysis to assess the influence of factors such as the geographical origin of the policy, economic background, type of institution, disciplines, and tracks on the development of highly detailed policies that include rubrics (see table S6). All of these can be found in section 1.4 of the supplementary information (SI) in lines 215-243.

2. Similarly, the collection of both institutional and national policies is a strength of the study, but the lack of analysis of where national policies were more prominent makes it difficult to disentangle the national/institutional analysis from the Global North/South analysis.

We thank the reviewer for highlighting this. Following the reviewer's advice, we expanded the description of the sub-samples by region, type of institution, and other factors in section 1.2 of the SI. Here, we now include a comprehensive table (Table S1) listing the different proportions of policies used for each analysis. Furthermore, in section 2.3 of the supplementary information, Table S12 (power analyses) we presents evidence of the robustness of our data to support our analysis. Additionally, we have included a multivariate analysis to help us disentangle the effects of confounding variables, such as those relating to the number of institutional versus national policies sourced for each sub-dataset, the geographic and socio-demographic context, and the nature of the policy (Please see Figure 5 and lines 226 -241 of the results).

While we cannot accurately determine where national or institutional policies are geographically more prominent, as this is beyond of what we could possibly achieve (now stated in L 123-130 of our data collection & analysis), including these analyses clarify the role of each independent variable in our analyses.

Furthermore, despite the limitations of our scope, we endeavoured to develop, to our knowledge, the largest dataset of up-to-date policy criteria, which we have made publicly available. We have prioritised the reusability and reproducibility of this dataset, so that it facilitates future research endeavours. This data is readily available for studies aiming to unravel the connections between the characteristics of a country or institution and the criteria adopted. We hope that this data is valuable to support these and other studies.

3. This relates to a broader issue of the diversity of systems that exist for researchers around the world (how were equivalencies of full professor found for systems around the world?). Was there any consideration for policies from countries where researchers hold dual affiliations (one with the national system and another within their institution)? While the manuscript acknowledges that such differences exist (similarly with different career tracks), the study design, analysis, or discussion fails to take them into account.

We thank the reviewer for bringing this to our attention. We now include the text clarifying these very relevant points. Text regarding the full professorship equivalences can be found in Lines 84-94 of the data collection and analysis section, where we specify: *"..career progression can take various paths, and types of posts that exist in one country might be unknown in others. Mapping or analysing the variety of career paths would be beyond the scope that we can reasonably cover. We therefore specifically decided to focus on policies for promotion to (full) "Professor", i.e. the most senior academic role, which has a profile that is omnipresent and can be meaningfully compared across countries"*.

Furthermore, in response to the reviewer's request, we conducted a survey of the policies to gather information regarding whether the position was exclusive, or if the policy was specific for the role of professor. This criterion was included in our analyses and presented in Fig 5 and described in L 223-226 of Sect 1.4 of the SI, and tables S1, S5 and S6. In brief, 85% of the policies specified that the role was

full-time and exclusive (prospective winning candidates cannot hold dual affiliation), while 79% specified that the policy had criteria exclusive to the role of full professor.

We also addressed the issue of "dual affiliations" by restricting the inclusion of policies to academic institutions. This decision was made on the premise that only policies from academic institutions applied in this context. Because, despite the fact that Professors may hold affiliations beyond academic institutions, the title is usually associated with a position at an academic institution, with only the academic institution having the authority to grant a "professorship".

Finally, we noted that in some countries, institutions follow national policies (e.g. South Africa or the Philippines), while defining further criteria in addition. In that case, we consider the national policy to be incorporated into the institutional policy (See section 1.2.1 of SI).

4. I also found no mention of how many of the documents found could be cleanly mapped to particular career stages/tracks. In my experience, there is a mix of documents that specify career stages/tracks and those that do not.

We thank the reviewer for highlighting this. We have now provided this information in section 1.2.3 of the SI and the breakdown is in Table S1.

5. Again, similarly, there is little detail provided on how much overlap there is between institutional and national level policies, or how much overlap there is between institutional policies (i.e., multiple documents from the same institution, potentially for different disciplines or different academic units). Was having a single document from a specific academic unit enough to include that country in the country-level analysis?

Thank you for bringing this to our attention. In response to the reviewers questions, we have:

1. Provided details of our sub samples in SI section 1.2 and Table S1, where we list information, such as L109-114: *"Some institutional policies were found to explicitly refer to national policies. In several cases, a national policy outlines basic requirements for promotion, while institutions can add additional criteria.. ."* In this section we also define more precisely that the unit of analysis in our study is policies (that is, a set of criteria applying to candidates) and not documents. We do not refer to documents because some policies are spelled out across multiple documents, some documents contain instead multiple policies, and as the reviewer notices, in some cases there are repetitions of overlap (for example, a national policy being simply reiterated by single institutions). For more details, please our enlarged discussion in section 1.2.1 of SI.
2. Implemented a post-sampling weighting system (See SI sect 1.6) where L 288-291: *"..all policies within the same institution or agency have the same weight, all institutions or agencies within the same country have the same weight, and each country has a weight proportional to the number of researchers active within that country."*
3. Further, we have provided more clear limitations about the scope of our analysis in the main text (Lines 123-135) where we state that any analysis at the country level or below, is beyond the scope of our project, and we clarify that the data set is not a representative sample at the country level.

6. In short, the study is lacking a more complete explanation of how the differences in academic careers and the diversity of documents were handled. Both of these elements should be expanded upon in Methods and in the Discussion. The differences in academic careers across countries, especially the extent to which academic career progressions can (and cannot!) be compared, should be addressed in the literature review and again in Point 1 of the discussion (noting, for example, that researcher mobility varies greatly by country). (For what it's worth, it was this complexity that held us back from analysing documents globally and made us decide to focus on the US and Canada, where the Review, Tenure, and Promotion systems are largely comparable).

We thank the reviewer for this suggestion, we hope that section 1.2 and 1.4 of the SI now include the detailed information of methodology requested. We also emphasised our limitations in L 84-94 of our main text and as requested, addressed this point in the discussion (see lines 266-271): *"The observed variations in adopted criteria across policies moreover show that institutions have substantial freedom of choice rather than having to assess researchers in a predetermined way. This gives room to both the diversity of institutions and the diversity of career paths of researchers. In the absence of a uniform research assessment system, institutions can adapt assessment criteria to their needs..."*

Furthermore, to address concerns of the influence of different tracks, we purposely increased the size of our samples to have the statistical power to include the influence of career tracks in our analysis. This is now shown in Figure 5 of the results, and outlined in L 220, 233.

7. For the most part, I don't think much of the above will affect the result of the analysis, but it would add some *detail* and present some *limitations and caveats* that should be addressed. The one area where, I suspect, it might matter the most is in the disciplinary analysis. Where those disciplinary documents are sourced from is likely to matter and some checks should be done to describe the geographic representation of that subset of documents. (Any further detail should be considered in Point 4 of the discussion).

We agree with the reviewer, and we now provide the information regarding the origin of policies by disciplines in Table S1 of the SI. We have also stressed our limitations and caveats in the manuscript and in the SI, such as that presented in lines 84 -94 of the data collection and analysis, .

Lines 123-130: *"Our analysis is solely based on the presence or absence of these criteria, we are not considering the assessment process. Specifically, we did not consider the role or weight of criteria, nor did we capture how assessors interpret the policies, and how an assessment panel ultimately arrives at its decision..."*

Lines 131-135: *"In order to obtain a global picture that reflects the population of researchers, we used post-sampling weights, so that each policy for the same institution or agency has the same weight, each institution or agency within a country has the same weight, and each country has a weight proportional to the number of researchers active there."*

A few other comments to consider:

8. The analysis of “outputs” lacks detail on the types of outputs that are being solicited. Are only traditional outputs considered? In our own work (Alperin et al., 2022), we found 127 different types of outputs listed. While I’m not suggesting authors use this level of categorization, it might be helpful to note which types of outputs were included.

Given the wide diversity of policies and our aim to provide a global understanding of the evaluated criteria, we focused our study on capturing the benchmarks commonly used to assess outcomes, which predominantly revolve around publications. The only other distinction we made is between publications and registered intellectual property (such as patents), which are also included in a significant subset of the policies. We have now explained this in supplementary information section 1.3.1 L 169-178 as follows: *“At the onset of this project, we performed an empirical exercise to understand and develop a method to quantitatively capture the qualitative data embedded in a policy. Through this exercise, we identified common keywords or clusters of keywords relating to similar criteria.... Furthermore, while there may be various outputs listed in different policies, we focused our study on those that were consistently mentioned across the policies. These primarily refer to “publications” or “intellectual property.”*

9. The estimate of the number of researchers covered by the study seems inflated. If documents were selected for those that apply for promotion to (full) professor rank, then using the more generic “researcher per million” value as an approximation is not accurate. This speaks to the point raised above about the need for further details and discussion on career stages.

Thank you. We have dropped these statements in this context and used the number of researchers in full time equivalents for each country in our sample to apply a global weight, as outlined in lines 131-135 and sect. 1.6 of the SI. The main assumption behind our work is that the number of candidates for full professorship in a country is roughly proportional to the number of FTE researchers in the country. But as a further note of caution, we now consistently refer to “potential candidates”.

10. I did not find much value in the flow described in Figure 1 (this flow is described in the text and is sufficiently obvious as to not warrant a visualization). The map is more helpful, although perhaps it could be coupled with further details on number of documents by type, as requested above.

As suggested by the reviewer, we have dropped the figure and replaced it with a map composite illustrating the number of researchers per country included in the sample and the number of policies and institutions sampled in each country.

11. Section 4 of discussion: possible typo: “Communalities” à “Commonalities”?

Indeed, thank you for spotting this. We have corrected it.

12. I could see an argument made that the latter half of the discussion (after the five summary points) is not sufficiently rooted in the study results or the literature. Perhaps because I find myself agreeing with the perspective provided, I found the section to be a helpful conclusion to the article, but I would not be opposed for further linkages to the study findings and to previous literature.

Thank you for this suggestion, we took the opportunity to add a few points that explore further linkages with the literature and how our study findings compare with it. For example see those included in lines 376 – 383.

13. Despite the questions and comments raised, I believe this to be a valuable study. I find the factor analysis to be a useful way of adding some depth to binary indicators and the conclusions drawn to be reasonably drawn from that analysis. A study of this nature will always have limitations owing to the impossibility of neatly capturing the global diversity in research systems and cultures. I appreciate the ambition of the study and believe that the scholarly community will benefit from being presented with this global comparative view, despite its minor shortcomings.

Thank you. We appreciate your thoughts and feedback. We have clarified our limitations in this regard in lines 84-94.

Referee #3 (Remarks to the Author):

The most significant contribution of this study is the global nature of the analysis allowing for comparisons regarding review for academic promotion to be made between Global North vs. South and between countries with different income levels. Similar analyses have been conducted in the past but have not comprehensively analysed policy/procedure from across the globe. Also interesting are comparisons between institutional level and national policies governing assessment of research outcomes. It is interesting for researchers to gain understanding about what is valued in different regions, countries, and levels of administration. The insights gained have potential to guide efforts toward creating greater consistency in academic procedures globally, and may inspire leaders to apply approaches different from those commonly employed in their own region.

Thank you. We value and appreciate your perspective.

1. It is helpful that the authors displayed the number of countries from which the documents were collected in Figure 1. However, it would be more helpful to display the number of documents collected from each region (e.g., a visual based on the data used to generate table S1) to understand the composition of the dataset better. This concern applies across all data presented - n (number of documents) should be provided for each bar/category in each figure/table, or in legends/captions, as appropriate.

Thank you, following the reviewer's advice, we drafted maps that have overlapping information of researchers in a country and the number of policies and institutions sampled in each country. Additionally, we now include a very detailed breakdown of our subsamples in table S1 and also include further information about the documents in section 1.4.3, tables S5 and S6.

2. A potential weakness is the crowd-sourced nature of the collected documents which may be biased by the membership of the GYA and doesn't necessarily include all countries with eligible academic

institutions. Another concern is the lack of stratification of the sample - some regions/income levels are likely overrepresented, and others underrepresented. This limits the ability to conclude about the truly global state of academic promotion practices.

Thank you for this feedback, we have addressed the problem of potential underreach by more than doubling our sample size, which now covers 121 countries (as compared to 55 countries before). However, it is true as the reviewer suggests, that the data set does not include every country in the world with at least one academic institution. Our sample contains the absolute majority of countries in the world, including all the most populous and/or research-intensive, and to our knowledge, it is the largest and most comprehensive data set to date.

We also provide a clearer description of our sampling strategy and GYA members involvement in lines 11-20 of section 1.1 of the SI, clarifying the sampling was not limited to the countries where GYA members reside: *“We thus leveraged on members and alumni of the Global Young Academy (GYA) to source documents not only from their own institutions, but in a way of snowball sampling also from other institutions they know or have a relation with, or by involving their academic network... ”*

The detail provided now clarifies that our snowball sampling with GYA members (from 5 continents) did not limit our extent, but instead expanded it, as it provided key knowledge of regional (continental and subcontinental) infrastructures, connections, and languages that were crucial in expanding our sample. Furthermore, in addressing concerns about potential bias in our sample, we acknowledge that our dataset is not a "random" sample of policies, but we strongly emphasize that our sample is "unbiased" for the purposes of our study, in the sense that the inclusion of a policy was never influenced by the criteria contained within the policy itself. We include this in lines 21-27 of the SI: *“While the only possibility in practice, we do not regard this snowball sampling as engendering any specific bias. Specifically, neither the composition of the group of people sourcing the policy documents nor any filtering of documents was informed by the promotion criteria that are the object of our study. We included each and every sourced document that was sufficiently clear and comprehensive for us to identify the absence or presence of specific criteria (see Sects. 1.2 and 1.3). While our set of policy documents does not constitute a random sample, it can reasonably be considered as an unbiased one for the purposes of this study.”*

Finally, we have introduced a weighting system that allows us to adjust for the potential impact and influence that a policy has, considering the population it may affect. As detailed in lines 292-299 of section 1.6 of the SI: *“Consequently, weighted averages or proportions in our data set become indicative of the global fraction of researchers potentially affected, provided that averaged over many countries the following assumptions hold: (1) the share of potential candidates for full professorship is proportional to the share of total researchers; (2) the policies set by every institution or agency within a country cover a comparable number of researchers (that is, a comparable number of potential candidates are subject to them); and (3) policies within an institution or agency cover comparable numbers of researchers.”*

3. Can the authors further clarify the implications of stratification by career tracks in the method section? Perhaps examples would be helpful.

Thank you for this observation, we have now provided a detailed full section addressing this in SI section 1.2. *“Our dataset was not obtained through a stratified sampling procedure, as we collected all documents that spell out the criteria for promotion to “full professor”. For the purpose of differential analysis, we classified the policies by their country and related region, their scope (“institutional” or “national” policies), career tracks, and academic disciplines”.* For academic tracks (section 1.2.3; L135-139): *“We identified this standard academic track for each of the policies and adopted its set of criteria as the default for our analysis, thereby allowing and capturing some diversity on what is understood as the role of “professor” on standard academic track. Besides the standard academic track (research & teaching), a substantial number of policies (48%) consider a research-focused track, a teaching-focused track, or a clinical track”.*

As explained in the SI section 1.6, we consider a document to contain more than one policy, where the criteria spelled out differ by discipline or track. To normalize the influence of policies across institutions, we implemented a weighting system where each institution has weight equal to 1, regardless of the number of policies that we codes from its documents.

4. What types of national documents were used? The description in the method section merely states the use of published documents - can examples be given?

Thank you for highlighting this, we took the opportunity to list concrete examples of government agencies that have issued national policies. We agree that it is insightful to see what kind of agencies are involved in this. We list these in section 1.2.1. For example: L96: *“i. Government agencies under a universities, education, or higher education portfolio: National Agency for Quality Assessment and Accreditation of Spain (ANECA), National Universities Commission (Nigeria).”* L102. *“ii. Government agencies under a research, innovation, science, and/or technology portfolio: National Research Foundation (NRF, South Africa), Department of Scientific Research and Technological Development....”* L107: *“iii. University Grants Commission (Sri Lanka), Department of Budget and Management (DBM, The Philippines).”* Furthermore, we share the URL of all the publicly available policies and the codes we used to classified them in our dataset.

5. Document analysis: From what method/process did the mapped 18 categories/4 groups originate? Interesting that teaching is represented as a singular category whereas research and service are broken down into numerous detailed sub-categories - is there a reason for grouping all teaching themes (and even placing teaching within service)? Teaching often comprises the majority of the academic workload, despite its value usually weighted less in academic promotion documents.

Thank you for raising this point. We have followed the suggestion and included rationale for the categorisation and the grouping of teaching activities. We now provide a detailed section of the definition of criteria in our section 1.3.1, lines 169-178 and explaining how they originated: *“At the onset of this project, we performed an empirical exercise to understand and develop a method to quantitatively capture the qualitative data embedded in a policy. Through this exercise, we identified common keywords or clusters of keywords relating to similar criteria. This gave us 30 distinct typical evaluation criteria that occurred consistently across a substantial number of policies”*

We also provide the rationale for the categorization in section 1.3.2. lines 182-193: *“ We found that the majority of policies are explicitly structured around the three broad categories of “research”, “teaching”, and “service”, the latter covering the contributions and impact to the wider profession and society”.* Additional information is also provided now in Tables S3 and S4.

In addition, our rationale for not subcategorize teaching can be found in section 1.3.1. L 173-178: *“We specifically considered teaching as a single criterion rather than delving into the complex details of teaching assessment, because it was found to be highly variable across policies and difficult to categorise with our current methods and scope.”* The conflation of teaching and service arose from the criterion of “mentoring”, which was seen as overlapping. We renamed the category “Teaching & Service”. Incidentally, in the factor analysis the criterion of whether the policy considers teaching, does not enter with a large loading in any factor and retains a high uniqueness, suggesting that this variable is indeed separate from the others (and if our analysis was driven by purely empirical considerations the variable should probably be dropped). In our view, this would call for a separate, in depth analysis of the inclusion of teaching among the evaluation criteria (which, as the reviewer suggests, is unfortunately often discounted), but such an investigation is beyond the scope of the present work.

Thoughts on Conclusions:

6. Point 1: It seems that differences in institutional requirements may also produce situations in which researcher mobility is limited (i.e., research priorities and valued achievements at current university may not be valued at a different university).

We agree with the reviewer, and we added the following to the discussion, lines 330-335: *“but they need to be aware that institutions are not the same and therefore not every institution will be a good match.”* We also now further elaborate on this with *“Pushes to increasingly conform to standardised research profiles are detrimental to diversity of backgrounds and ideas, already adversely affected by adherence to bibliometric research evaluation [43], and hamper inter-sectoral mobility across academia, industry, government, and not-for-profit organisations. Capable and well-skilled researchers can face a-priori exclusion not only by inflexible policies, but also by norms enshrined in the minds of assessors.”.* An important fact to keep in mind is that academic institutions are not identical (which makes a case for assessment criteria to differ). In fact, a very common reason for rejecting applications for an academic post is that it is generic and not tailored to the specific institution.

7. More emphasis on the idea that evaluation committees generally have flexibility in their consideration beyond what is listed in the documentation is warranted.

We agree that this is an important point to keep in mind, and we now state this more clearly and prominently in lines 123 – 130 the text of the main document as follows: *“Our analysis is solely based on the presence or absence of these criteria, we are not considering the assessment process. Specifically, we did not consider the role or weight of criteria, nor did we capture how assessors interpret the policies, and how an assessment panel ultimately arrives at its decision, potentially taking into account further criteria that are not explicitly mentioned in the policies.”*

8. It is concerning that 44% of documents in this study did refer specifically to journal impact factor when impact factor is such a poor metric for research quality - this might be emphasized as a greater concern.

Thank you. To clarify, under the respective tabulated criterion, we cover both direct references to the Journal Impact Factor as well as references to criteria that derive from it, such as journal indexing (see Item 5 in Table S2 - Definitions of Criteria). We have relabelled this criterion in order to clarify this point. We agree that the Journal Impact Factor is a very poor metric indicator of research quality and we make emphasis of it in lines 277 -87 of our discussion, and expressed our explicit concern that upper middle-income countries have the strongest affinity with output metrics in point 2 of the discussion *“Scientometrics are most popular in upper middle-income countries”*.

However, on the bright side we think that it is important to also point to the fact that many documents do not make any (direct or indirect) reference to it (point 5 of the discussion: *“A bibliometric profile is not a key to success everywhere”*), while many people believe that it is now of universal relevance all across academia.

9. Point 2: What factors do the authors think cause higher income countries to rely less on scientometrics? Perhaps consider the increased labour required for qualitative analysis.

We much like this question, which was raised by another reviewer too, and are happy that our findings raise this kind of questions. Properly answering it beyond some speculations, however, requires further hard evidence that we do not have (yet), and we wanted to avoid making speculative statements in the discussion of our current study, given that there is some risk of drifting off and diluting the messages that directly derive from evidence. May we take this as a suggestion for our future research directions?

10. Point 3: Do the authors have any comments regarding the effects of national vs. institutional policies being out of alignment with each other in both Global North and Global South? What impact does this have on institutions and researchers?

Thank you, we actually started speculating about this point, and arrived at the conclusion that we do not understand this outcome. A related (and rather complex) question is to what extent we actually would want an “alignment” (rather than freedom as responsible actors and diversity), and we noticed quite different views about this across representatives e.g. of CoARA member organisations. We ultimately decided not to engage in speculation in our discussion session beyond the supporting evidence.

11. Point 4: Also the JIF was not created for the purpose of evaluating the impact of individual articles: <https://www.frontiersin.org/articles/10.3389/frma.2018.00002/full> #:~:text=The journal impact factor (JIF, known invention of Eugene Garfield)

Yes, we agree with the reviewer. We are aware that JIF was not meant to be used as a metric to evaluate researchers, and experts on scientometrics tend to point that out all the time. Unfortunately, we can document that many people do not listen to them. We mention the misapplication of metric indicators to the evaluation of scientific performance, and we now explicitly refer to the Journal Impact Factor as a prominent example, citing the suggested reference. See lines 51-54 of the introduction: *“experts on*

scientometrics have raised concerns that evaluation has increasingly become led by data rather than by judgement and that the misapplication of indicators to the evaluation of scientific performance has become pervasive, the Journal Impact Factor being a prominent example [20, 21]."

12. Also, this may be a good section in which to emphasize the value of regional/community engagement and acknowledge the variability this may produce in scientometrics despite increasing real-world impact of the research.

Joining this with a point made by another referee, we are now elaborating a bit further on the three-way dissonance between what counts, what is perceived to count, and what should count. Notably, this frequently forces researchers to make a tough choice between positively contributing to society and advancing their career. See discussion lines 381-383: *"On the assessment of researchers, one finds a three-way dissonance between what counts, what is perceived to count, and what should count [60, 61]. Researchers often face a tough choice between wanting to make a positive difference to society and advancing their career."*

13. Although the Global North can perhaps "afford" an inefficient research ecosystem, it is still wasteful and inequitable and should also be addressed with change. It may be worth proposing a global initiative (inclusive of North and South, and all levels of income) rather than placing responsibility on the Global South to lead the way. Perhaps such a global initiative can be flexible to allow for regional strengths to shine and meaningful collaborations across regions and income levels to flourish. The authors may make practical suggestions for bodies already working towards these ideals (as mentioned in the introduction), based on the findings of this study.

We are very supportive of global initiatives and do not want to suggest that the Global South breaks away. It is our intention to stress that the Global South is well-placed to take a leading role in such global initiatives, we have rephrased the concluding paragraph to: *"The obsession with frequently ill-suited metrics has created a research ecosystem that is highly inefficient. While many countries in the Global North can afford such inefficiency (but should not), it becomes more important for the Global South to adopt strategies for success that take a different approach. Our study challenges South-North catch-up strategies based on unsuitable performance indicators. Taking the opportunity to build research environments that better meet their purpose is not primarily a matter of funds available, but mostly about fostering a different kind of culture, as shown by Latin America's world-leading model for Open Access publishing [62]. Such kind of initiatives can provide key input to platforms like CoARA [33] that aim at building a global community. Rather than letting the Global North sort out things which the Global South then adapts to, actors from the Global South are well-suited to take the lead on global initiatives that show the way forward"* to improve the clarity of our message.

Response to Referees

Editor's comment

Thank you for your patience while we waited for our referees to re-review your paper entitled "A global assessment of academic promotion criteria: What really counts?" It has now been seen again by our original three referees. You will see from their comments below that while they find your work of interest, Reviewer 3 still raises an important part about the generalizability of the results, given that they are driven by the highest populated countries. We remain very interested in the possibility of publishing your study in Nature but would like to consider your response to these concerns in the form of a revised manuscript before we make a final decision on publication.

We therefore invite you to revise your manuscript taking into account the following points:

- 1) Please add any analyses possible to determine how much the results are influenced by the most populated countries / change (or not) if you look at different regional groupings.

Thank you for the comment and the appreciation for the important effort in data collection behind our study.

To make the influence of various countries more transparent, we now explicitly provide the country weights as a table in the SI. It turns out that the ten countries with the largest number of researchers contribute 72% of the total number of researchers globally. To measure the effect of these very large countries on our results, as compared to the remaining 28% of the population of researchers in the 121 countries that form part of a study, we carried out a further analysis solely on the subsample of the 111 countries with relatively fewer researchers.

Picking a job offer or promotion opportunity at random on a global scale, there is a high probability that it will be regulated by norms or standards set in a large, more resourceful research system such as China's or the US, than in a Latin American, South Asian, or Sub-Saharan African country. As the reviewer correctly emphasises, this implies that a handful of countries acquire an overwhelming relevance globally. We now highlight this more clearly in new **section 2.6** of the SI (lines 470 – 530, Tables S14 – S16), where we report the full distribution of country weights (Table S14) and comment on its skewness. We make mention of this analysis in lines 138-145 of the manuscript.

This leaves open the question raised by the reviewer, i.e. to try and investigate which specific results are driven by one or very few big countries. We believe an exhaustive answer to this question will require a separate in-depth study, with more data representative at the national level. However, a first answer can come from some robustness checks. In the new revision of the SI (section 2.6), we

now report the results of the same factor and regression analyses reported in the main manuscript, on a reduced dataset obtained by excluding from the analysis the 10 countries with the highest weights. Any difference in results is bound to arise from some systematic difference(s) between one or more of these top 10 countries and all the rest. To our surprise, despite the large difference in the scale (and presumably characteristics) of very large and very small research systems, most of our results seem to replicate in the subsample. We find that four factors best explain the data on co-occurrence of criteria, and these factors have the same interpretation as those of the main analysis. These factors also largely correlate (or not) with the same variables, with only minor differences in some cases. As mentioned, we believe more in-depth analysis should be carried out in future work. However, the robustness of our results shows that our study is 'global' also in the sense of highlighting apparently consistent worldwide trends.

- 2) Regarding R1's point about your sample including many countries outside of Europe and North America: we suggest that you change the title from "global" to multinational (as we do not think that there is room in the title for more detail). We also ask that you change "global" to "multinational" in the summary.

We appreciate your feedback. In this case, we strongly believe that "Global" is an appropriate description for this study, as it accurately reflects the representation of research institutions worldwide. We qualify our work as "global" in the sense that it considers active researchers (who are not already full professors) globally as a single community, and we try to depict world-wide trends.

As stated by UNESCO (SI Ref. 6-9), the number of full-time researchers in a country is indicative of its investment in science and overall scientific activity. Table S14 highlights that countries/regions such as China, India, South Korea, Japan, and Taiwan are collectively more influential in the research ecosystem than the US and many EU countries together. Additionally, countries like Australia, Brazil, Mexico, South Africa, and Thailand exhibit comparable research activity to countries such as Canada, the United Kingdom, Spain, Italy, and France. Furthermore, the number of researchers per million inhabitants in countries like South Korea, Japan, and Israel surpasses that of the US, Canada, and Germany (see table "Number_Researchers" in our datasets). Thus, we believe that having a significant representation of data points from non-EU and non-US countries strengthens our manuscript and makes it more representative of the global research landscape.

To further substantiate our position, UniRank.org states that in 2023, Europe and North America have 2,706 and 1,856 officially recognized higher-education institutions (universities), respectively. This totals 4,562 universities, while globally there are 13,837 universities. This means that 67% of universities are outside of Europe and North America. Given that 72% of our policies come from non-European and non-North American institutions, we believe our data reflects the global reality as accurately as possible. Following on this point we feel that focusing on the US and EU realities can be considered as a form of introducing bias to the study and that having a higher geographical diversity can provide richer and broader insights of the different research cultures worldwide, which can be of

far more value as it is more difficult to understand and gather. In our experience, multiple organisations would love for this data to be public, as a dataset as diverse as the one we present in this study, is not yet available and can help inform global initiatives.

Additionally, the feedback provided by referees 2 and 3 can be perceived in agreement with our global stance, especially the request from Referee 3 to show trends for those countries who are not dominant seem to be contradicting this request. Furthermore, we have found many other studies, such as the UNESCO Higher Education Global Report, and various from this publisher, have with similar or lower number of countries in their samples, with a comparable ratio of non-US, non-EU data points to be defined as “Global studies” For example: <https://www.nature.com/articles/s41398-018-0148-0>, <https://www.nature.com/articles/s41893-021-00842-z>).

We hope this information demonstrates that our title is well justified. We appreciate your consideration and look forward to your feedback.

- 3) Finally, we strongly suggest that your revised manuscript has tracked changes, which is increasingly requested by referees to aid in their re-review.

Thank you for pointing this out, we have kept tracked changes on the manuscript.

Referee #1:

I want to thank the authors for the responses to the earlier comments and the revised manuscript

- 1) This is not what I mean. Proposed research typically has a protocol. This is document that outlines the pre-study plans and proposed data analysis. This is typically done so that peer reviewers (and others) can examine what was proposed against what was done. This is an important way to detect reporting biases. If you did not have a protocol, this is something that needs to be indicated in a section of the paper - strengths and limitations.

Thank you for the suggestion, we now clarify this on lines 151-152: “The methodology for the coding and analysis of the data was developed after a pilot first wave of data collection and not at the onset of the project, as explained in section 1.1 of SI.”

- 2) Most people agree on the UNESCO definition of open science. There are fairly standard definitions of equity, diversity and inclusiveness. That said, I accept you need to assess for EDI or OS.

Thank you. Yes, the multidimensionality of OS is also pointed out in the UNESCO definition, which acknowledges that there is no clear definition and just refers to a conglomerate of various movements and practices: “For the purpose of this Recommendation, open science is defined as an inclusive construct that combines various movements and practices...” The proper assessment would require an elaborate analysis that is beyond the scope of this study.

- 3) The authors do not appear to propose any option for 'promotion criteria not met'.

Apologies, we still fail to understand the question adequately. If this is about the evaluation process, we have added in a "Limitations Summary" (L146-152) that we are solely looking at whether criteria are mentioned or not, not at the evaluation process.

- 4) A "strengths and limitation" section is typically placed towards the end of the discussion section, with a separate subheading. This is a place for readers to read about the main strengths and limitations of the paper - as viewed by the authors. For example, you should include something about the sampling approach. It is not systematic, but opportunistic.

Thank you for the suggestion. This may be a discipline-specific requirement to which many of us are unfamiliar. We agree that it is important to be clear about the limitations of the applied methodology, and therefore we point to these already when we introduce the methodology and provide a motivation for our choice of methodology. But following the suggestion of a "strengths and limitations" sections and to provide further guidance to the reader, we have added a clear summary of our limitations in Lines 146-152 of the main text.

- 5) Similarly, while the paper does not include a country level analysis, the paper does not include a continent or regional level analysis (e.g., Africa).

Thank you. We wish to clarify to the reviewer that we refer to "continent – and/or regional- level" analysis to the analysis presented in Fig 5, where we perform a multivariate regression to look for tendencies between the Global north vs Global South (regional) and between continents. We added "With very few exceptions, we also did not find significant differences across continents." in L254-255 to note on the results of the continental comparison. The regional comparison is described in L257 – L268.

A few additional comments.

- 6) The authors may want to revisit the title of the paper. Given that very close to 75% of the documents are from non-European and North American jurisdictions, perhaps a title more reflective of this sampling frame would be more declarative to readers.

Thank you for the suggestion. We value the feedback provided by the reviewer but disagree in this request. This is more elaborated in our reply to the editor (Request No 2).

- 7) I think some readers will not well understand the idea of government policies concerning promotion and tenure. For example, in most of Canada and the United States, universities (or other research institutions) would be making these decisions. Governments would not.

Thank you for this suggestion, to clarify this we have added: *"This distinction reflects the fact that academic institutions have different degrees of autonomy across various countries"* in lines 98-100 of the main text.

- 8) There are many authors of the paper. Are most of the authors included because they were able to access the documentation (at their respective university) or did they contribute in other ways? This information might be best achieved using the CREDIT system <https://credit.niso.org/> for authorship.

We based our definition of authorship following the ICJME guidelines, and McNutt et al recommendations. At the onset of the study a minimum contribution threshold was established, and the authorship order reflects the amount of effort placed by each of the authors on this manuscript. No authorship was awarded just for sourcing policies, as per McNutt et al. these and other non-substantial contributions were acknowledged in the acknowledgement section. All authors sourced, studied and coded the data in policies, and reviewed each other's work so that each policy was evaluated by 3 authors. Therefore, each co-author reviewed and/or tabulated a minimum of 25 policies. All authors were involved in the elaboration of the manuscript, approved their content and have signed a form agreeing to be personally accountable for their contributions, as stated in the author contribution section and in the dataset shared. In this regard, the dataset clearly shows which policies were tabulated and revised by which author (see columns CR to CU). The authors contribution section was written following the CREDIT model as a base and adjusted to the requirement of this study. Please refer to columns CR-CU of our table "Data of Tabulation" in the dataset made available to the reviewers and the authors contribution section (L639-658 of the main manuscript).

Referee #2:

With apologies for the delay in reviewing this second version of the study, to which I wanted to give my undivided attention.

I am quite satisfied with the revisions by the authors. I was initially concerned that a simple edit would not be sufficient, but the additional data collected, and analysis presented do a fine job of speaking to my major concerns. Most notably, the collection of additional documents to allow you to disentangle the national and institutional policy analysis from the Global South and North analysis. The additional methodological details provided, such as the breakdown of document types and the check of whether policies are for full professor only were particularly helpful. Just as importantly, the edits with additional methodological details, nuanced phrasing, and explicit acknowledgement of the limitations offer greater confidence in the results.

We thank the reviewer for the positive remarks and the appreciation for our work. We value the reviewer feedback and are happy to know that the revised version satisfied all doubts.

Referee #3:

The authors have significantly updated their manuscript to incorporate additional data, information, and changes consistent with the requests of the reviewers. The method used for data collection and analysis is clarified and appears to now be transparent. I appreciate the substantial efforts of the authors.

Major comments:

The weighting of data by number of FTE researchers per country makes sense and is an improvement compared to the original analysis because it more proportionately represents the policies influencing researchers across the globe. However, it is now clear that a handful of high-population countries in both Global North and Global South are dominating the analysis due to their researcher numbers dwarfing those of many other countries. For instance, countries with 1+ million researchers (e.g., USA, China) will have three orders of magnitude the influence on the reported outcomes compared to countries with <1000 researchers (e.g., much of Africa). Thus, the presented data seem to make a potential waste of going to the effort of collecting policies from the (many) countries with relatively few researchers. Perhaps the authors would like to think about additional analyses or methods to report on the policies affecting these smaller populations of researchers.

We thank the reviewer for the suggestion. To clarify this point, we performed a robustness check, which is now presented in section 2.6 of the SI. In this analysis, we replicated our analyses using a sub-dataset representing the bottom quartile of countries, which we did by removing 10 countries which carried ~75% of the weight. For more details, please refer to the response to the editor, and SI section 2.6 (L470-531).

Specifically, we believe that using weights does not imply a 'waste' or unused data as the reviewer suggests. We are aware that yet more analyses could be carried out on our dataset, and we regard its publication (along with the publication of the article) as a substantive contribution per se. However, data from a plurality of contexts is necessary already for our analyses of separate subsamples (e.g. when comparing Global North and Global South), and in particular, adequate heterogeneity along several dimensions is a requirement of the multivariate analysis, where we single out the different roles of geography (considering the main Continents, and the Global North/South) and simultaneously of economic classification of countries, institutional vs. national character of policies, as well as general vs. discipline-specific policies, track-specific, and rank-specific policies vs. those that only apply for full professorship. While weighting the data impacts on the results (as documented in the SI), it does not suppress the diversity within our dataset.

Minor comments:

- 1) There should be a citation in the caption for Figure 1, and within lines 131-134 (In order to obtain a global picture that reflects the population of researchers, we used post-sampling weights...), to identify the source of the data on the number of researchers within each country as used for weighting of the policy data.

We thank the reviewer for pointing this out, we have now added references 36-42 and the text: *“The number of researchers was obtained from the UNESCO Institute for Statistics and UNESCO Science Reports (see SI for details)”* to Figure 1 legend and main text and have also added references to the sections of SI describing this in L 136: *“see more details in SI sections 1.6, 2.6 and Table S16) [36-42].”*

- 2) Should be mentioned in the article about leaving out Art and Creative Works as a discipline.

Thank you for the suggestion. Although this was already mentioned in lines 74-77 of the SI, to improve the clarity, we have also added the following: *“We did not consider scholarship in the Arts and Creative works given substantial differences in the nature and relevant outputs of this field in comparison to other disciplines”* in lines 152-154 of the SI section (1.2.4. Disciplines).

- 3) Some technical analysis descriptions in lines 184-191 may be simplified and details left to the supplementary section for improved readability.

We thank the reviewer for the suggestion. We have made the change requested and the paragraph now reads: *“We found that four main factors explain 65% of the combined variance in the data (see Figure S1), each factor representing common information that is shared among criteria that tend to be used in the same policies. Figure 4 reports the factor loadings. As it emerges, the quantitative and qualitative criteria for assessing research outputs were clearly separated by our factors, demonstrating that they were a most distinctive characteristic across policies, whereas other groups of criteria that we defined ex-ante were somewhat entangled (See Figure S2).”* Now in lines 201-208.

- 4) Figure 5 caption - list reference categories in the same order as bars are given top to bottom to aid readability. Or overlay a reference category directly on the chart at the top of each set of bars.

We thank the reviewer for highlighting this. We added the reference categories in the figure legends, below the dimension categories and have modified the caption as follows: **“Figure 5. Coefficients of the regression analyses.** Relation between the assessment factors and various dimensions of the policies specified in the legend, measured as deviations from the reference category presented in brackets in the figure legend”.

Editor remarks:

Your manuscript, "A global assessment of academic promotion criteria: What really counts?", has now been seen again by Reviewers 1 and 3. You will see from their comments below that they appreciate your revisions; however, they still have some methodological questions (R1) and questions about the interpretation (R3). We are very interested in the possibility of publishing your study in Nature but would like to consider your response to these concerns in the form of a revised manuscript before we make a final decision on publication.

We therefore invite you to revise your manuscript to include the additional methodological details requested by R1 and to include the differences highlighted by R3. We also strongly suggest that your revised manuscript has tracked changes, which is increasingly requested by referees to aid in their re-review.

We thank the editor for the opportunity to present a revised and improved version of our manuscript. We believe we have completely clarified the queries from Reviewer 3 and addressed all concerns raised by Reviewer 1 by incorporating additional documentation and background information that will strengthen our methodology. We are confident that our methodological design is now clearer.

This project represents nearly a decade of work, with the first four years dedicated to defining the methodology and the subsequent three years focused on consolidating it to ensure its reproducibility and scalability for it to be performed by a large, global team. We have also greatly benefited from this peer-review process and sincerely thank the reviewers for their time and input. Their contributions have become an integral part of this manuscript, enhancing the presentation of our methods and data, and suggesting analyses that have improved the quality, transparency, and reproducibility of the study.

We are grateful to the editor for guiding this review process and to the referees for being key contributors to this work. This manuscript is the result of a concerted effort by the authors, editor, and peer reviewers, and we are personally impressed by the high quality of the final product. We hope our fellow contributors share this positive view.

Referee #1 Remarks:

I am still concerned about the methods. Without robust methods that are replicable, the results become tenuous.

There is still no reporting of whether a protocol exists. This is an example of what a research protocol is ([https://icahn.mssm.edu/files/ISMMS/Assets/Research/IHCDS/Guidelines for Writing the research protocol by WHO.pdf](https://icahn.mssm.edu/files/ISMMS/Assets/Research/IHCDS/Guidelines%20for%20Writing%20the%20research%20protocol%20by%20WHO.pdf)). Research protocols are important because they allow peer reviewers, and others, to examine for the possibility of reporting biases (<https://www.bmj.com/content/362/bmj.k3802>) in completed research. While the example provided is from randomised trials, reporting biases are common in other types of research.

We appreciate the concerns raised by the reviewer and wish to assure them that it is of utmost importance to us that the database shared with this manuscript—which we consider a most valuable contribution of this study—serves as a baseline for further research and informs policy development. Ensuring our data's compliance with the FAIR principles was crucial to guarantee its reproducibility, interoperability, and reusability. This commitment informed the structure of our data frame, our working procedures, and the data and code shared.

We also thank the reviewer for suggesting the inclusion of a protocol with the publication. We recognize the value of this suggestion and have now included a protocol in our reproducibility package (please see Data & Code). Before the collection and analysis of data we did not have a document *structured* precisely in the form the reviewer suggests, but we did have documents and other material (including written guidelines, documents templates, video recordings of online meetings, and a recorded demonstration session) defining our empirical strategy and aimed at facilitating collaboration among our large, international, and interdisciplinary team. However, following the reviewer's suggestion we have now put all this information together (most of which was scattered throughout the SI) in a single document with the suggested protocol format, which we include in the submission as extended data. We also share the following: (1) *General and* (2) *coding guidelines*, (3) *criteria definitions*, (4) *a standardised matrix for coding policy documents*, and (5) *examples for coding and reviewing data* that were instrumental in completing this work in the reproducibility package. The only documentation we are unable to share are the videos produced during team meetings, due to data privacy mandates. We wish to highlight that the most critical guideline was the definition of criteria, shared in Table S2. All these documents have been also uploaded as supporting information to our FigShare And CodeOcean capsule (facilitated by the journal, to share our data and data analysis code and to verify that all our computational analyses are reproducible). We believe the reviewer has access to the capsule in this link: <https://codeocean.com/capsule/0942594/tree>

The authors start the methods section with data collection & analysis. Typically, readers like to read something about the study design. It looks like the authors have conducted a cross-sectional study. If I'm correct, this information should be included in the methods section. Similarly, what were the eligibility criteria for selecting documents.

We thank the reviewer for the suggestion, we have changed the label of the old section data collection & analysis to **“study design and limitations”**. Here we describe the rationale for our methodology and limitations. We have also included the following paragraph in the main text:

“Understanding what matters for institutions globally is crucial for informing policies and strategies that address counterproductive assessment practices. To study how researchers are evaluated worldwide, we conducted a cross-sectional analysis by examining the assessment criteria used in promotion policies. We systematically identified and analysed promotion criteria, comparing differences and similarities across disciplines, fields, tracks, types of institutions, and countries, considering their socioeconomic contexts. The methodology was informed by an initial pilot study and subsequently developed into a comprehensive framework that allows for the comparison and quantification of qualitative data encoded in documents sourced from institutions around the world (see Section 1.1 of the supplementary material).”

We have also expanded and provided further detail of our study design and how it came to be in Sect 1.1 of the Supplementary information:

“This study was initiated by some of the authors who, in 2016, sought to understand what matters for institutions globally in terms of scientific excellence and to which extent our global research culture is influenced by metrics. The team hypothesised that these factors could be objectively studied by looking at the assessment criteria used in institutional promotion policies. Leveraging the global platform provided by the Global Young Academy (GYA), the team launched a call to all members and alumni (representing 100 countries) to request promotion policies from their respective institutions or countries.

As a result of the call, the team collected 42 promotion documents from various world regions and began defining a methodological framework to capture and quantify data embedded on the documents, enabling global comparison. This process required clear definitions of which data to include, as well as how these data should be coded to ensure reproducibility and reusability. These details are expanded in our protocol, presented in a point-by-point manner and throughout Section 1 of the Supplementary material. However, here we briefly narrate the rationale and context.

With this empirical exercise, it became evident that academic careers are not uniform worldwide and that the working conditions of researchers are significantly different. To address these differences, we imposed certain limitations on the way we sourced, interpreted, and recorded data. First, we limited our data sourcing to promotion policies for full professors, as this role was the only one universally present in virtually all documents describing policies (see Section 1.2). We retained only documents that offer academic paths or include posts covering both research and teaching, while acknowledging the fact that some institutions offer different career tracks that may not be comparable. Additionally, after conducting a round of interviews with representatives from various institutions to explore data sourcing options, the team concluded that a strict and consistent interpretation of the data was essential, limiting our study to accounting for the presence or absence of specific assessment criteria. The definition of our data units and the adopted procedures for their inclusion and categorisation are detailed in Section 1.2.3...”

How were the documents accessed? What processes were used. A few sentences describing the process in the methods section would be of value to readers and others interested in replicating what the authors did.

We thank the reviewer for highlighting that this section was not readily noticeable on our previous section 1.1 of the Supplementary Information. We have restructured the section to improve its visibility. We separated the paragraph into a section **“1.2.2. Sample composition and access”** (lines 86-94).

“We consider each distinct set of assessment criteria applied for a promotion process to constitute a “policy” for promotion. Some institutions or agencies distinguish different career tracks or academic disciplines, and therefore there can be several policies within the same institution or agency. In total, our dataset covers 532 policies of which 426 policies (78%) were sourced from public websites, 106 policies (20%) were sourced through the co-authors’ networks and 12 policies (2%) were obtained via an official request made through the GYA office. We share the respective URLs of policies found online in our dataset.”

We also added the following line (92-94 in supplementary information) to improve clarity:

“All publicly available documents (78%) were accessed through corresponding websites, and the others were obtained and managed as hard copies in PDF files.”

The sampling time – May 2016 to November 2023. Were some documents sampled in 2016 and others in 2023? A document sampled in 2016 might be out of date in 2023. This is particularly relevant given the prominence in researcher assessment over the last 5 years.

We thank the reviewer for the remark. We have restructured data from Lines 41-48 SI section 1.1 into the new subsection **“1.2.3 Sampling periods”**. We have also included a more detailed description of the sampling periods as follows:

“The first sampling wave took place between 2016 and 2018, during which 46 policies were collected and then used to inform our methodological framework. A second wave of sampling occurred from 2018 to 2021, yielding 159 policies from 56 countries, which were analysed to produce the first version of this study, available as a pre-print [2].”

From these sample collection waves, we observed that the most common update cycle is 5-10 years as the reviewer correctly points out. After the first round of revision, it was suggested that we update the dataset to ensure it is up to date and we did. This may have not been clear before, and we hope the restructuring helps. The information is in Lines 99-107, as follows:

“After the reviewer advice and noticing that many policies have an updating cycle of 5 or 10 years, we drove a third round of sourcing in 2023 for the present analysis, to check for updates in the policies collected and new policies to expand our global reach. From this exercise, we sourced 440 policies, of which 300 (56%) were implemented after 2020, 148 (28%) after 2015, 62 (12%), before 2015 and for 22 (4%) policies this information was not disclosed. Furthermore, based on the evidence gathered, 460 (87%) of the policies in our sample were applicable as of Dec 2023. While we could not gather any evidence of updates or revisions from the remaining policies, most of these policies were known to apply in 2020 and are very likely to still apply in 2024, as no replacement was found. Therefore, we are confident that our sample closely reflects the present state (2024).”

As best as I can tell the authors developed 18 categories and 11 sub-categories for mapping against the documents/policies. As a reader it would be invaluable to know how the categories and subcategories were developed? Are they evidence-based?

Thank you for bringing this up to our attention. We have 30 criteria (sect 1.4.1) that are grouped into 3 broad categories (research, teaching, services), as was found in many policies. Some categories were used in different combinations, and to further categorise the criteria we conducted a survey between authors where we voted for where we thought these traits would fit best. This is detailed in section **“1.4.2. Categorisation of criteria”**, as below. We now make mention of this section in section 1.1, hoping that it will improve its visibility.

*“We found that the majority of policies are explicitly structured around the three broad categories of “research”, “teaching”, and “service”, the latter covering the contributions and impact to the wider profession and society, as well as to the institution and its operations. Moreover, some criteria are frequently used in combination to provide evidence of key traits such as professional standing, accomplishments, or leadership. For further categorisation, we conducted a poll amongst all authors on whether each of our defined criteria resonates with some key trait, with the results shown in **Table S4**. These criteria do not include “interdisciplinarity” and “ethics & integrity”, which are understood to be “general traits” not clearly in the domain of either “research”, “teaching”, or “service”. This led*

to sub-categories that best cover all criteria without overlap, where 21 criteria related to “research” are grouped into “research outputs” (11), “career development” (8), and “recognition” (2), while 7 criteria relate to “teaching & service” (see Table S3). We combined the latter given that “mentoring” was perceived to relate to both.”

It looks like each document was reviewed by members of the author team. Did two people review each document and come to some form of agreement? If so, how was this done and can the authors report measures of agreement?

As we enforced an explicit mention of the definition of each specific criteria this facilitated the process. Each data coder and reviewer encoded their interpretations together with the quote that enforced the definition or the reason it didn't apply (in the corresponding evidence cell). A third, more experienced author (also data compiler) will note differences and notes left by each and resolve with consent of both, based on the answer that more closely relates to our stated definitions. In cases where the definition was not clear, the project leaders were involved. On a couple of occasions, this led to amendments of our definitions and a communication to all authors.

This was briefly explained in section 1.6 *“Furthermore, we established a two-step verification process: In the first step, each policy was tabulated by a team member and then reviewed by at least one other author. Discrepancies were noted and resolved with advice from team leaders, and if required, the project leaders (see more detail in the protocol).”*

Now, the reference to section 1.6 in section 1.1 and expand the information provided *“To improve our global coverage and incorporate reviewer advice (including policy updates), we recruited additional team members to participate in data sourcing, coding, and interpretation of the criteria. We established geographical teams and developed a standardised matrix template as well as data coding guidelines. Several meetings were held, and recordings exemplifying the interpretation and coding of data were distributed (see metadata in our replication package for examples; videos cannot be shared due to data privacy). Each author was given a package with the necessary information and dedicated space to store and code the data. Team leaders were responsible for distributing tasks among members and consolidating the data into regional sub-datasets. To ensure accurate coding, each policy was reviewed by two co-authors. Team leaders mediated any differences in interpretation and, if necessary, engaged with project leads to resolve discrepancies or amend the definitions of criteria. To facilitate this process, an evidence cell was included next to each encoded criterion, allowing authors to document the location of the reference and quote the exact text extracted (See section 1.6).”*

We also refer to this in the drafted protocol. Although we do not share the uncompiled data, where notes were made, we provide an example of this on our replication package. See the file “example criteria report”.

Referee #3 Remarks:

The authors addressed my major concern by conducting a robustness check of their results in which the data from all countries except the 10 with the most researchers were analyzed. These

new analyses are reported in the Supplementary Information, and in two sentences added to the "Data Collection and Analysis" section of the manuscript:

"However, in a robustness check described in the SI (section 2.6), we find that our main results are replicated in a subsample of policies that refers to the remaining 28% of the population of researchers. Therefore, our global picture is also a reasonable description with regard to geographical spread of countries and less skewed than the population distribution."

I appreciate the efforts of the authors in conducting this additional analysis. I wonder if the text included in the main text of the manuscript overreaches in terms of interpretation. A comparison of significant findings in Table S13 (all data) versus Table S16 (restricted robustness check data) shows that approximately 1/3 of the reported regression results differ between the tables. It may be more transparent to revise the text in the manuscript to acknowledge not only the similarities, but also that there are detailed differences described in the Supplementary Information, so readers are not led to falsely assume the robustness check exactly replicated the full analysis.

Otherwise, for future analyses, it may be worth reconsidering the approach of removing the top 10 countries. Perhaps a different method of selection could be justified more meaningfully.

Thanks for the comment. Indeed, in the previous version of the manuscript we have been extremely short, leaving most relevant information in the SI. We have now tried to add some details in the main text too. As the reviewer notices, the results of the factor analysis in the two datasets (complete and reduced) are very similar but not identical, and this could explain some differences in the regression results, besides possibly different correlates of the predicted factors.

In the new text added, we refrain from counting the instances in which we obtain different results, for two reasons: we preferred to focus on the qualitative changes that may have implications in terms of research policy; and most explanatory variables in the regressions are actually groups of indicator variables (dichotomous dummy variables) that denote a difference with respect to a same baseline, so that quite likely when one changes others will change too.

What we tried to highlight in the new text are the following points:

1. The different results are limited to changes in the statistical significance of coefficients, while no coefficient has a statistically significant sign that is reversed into an opposite statistically significant coefficient from one dataset to the other;
2. The differences are not a sign of lack of robustness but rather of the overwhelming relevance that a few countries' research systems have on the global level;
3. In terms of substantive results that we emphasise in the main text, the main differences in the restricted sample are: **a)** we find fewer differences between Global North and Global South and national and institutional policies, but we confirm the preference for metrics among national policies; **b)** we now find even fewer differences across career tracks, thus reinforcing our previous finding, and the same is true for policies specific for access to full professorship; **c)** we now find some more differences across disciplines, mostly concerning engineering or the fourth factor (outcomes), but overall most discipline-specific policies continue to be not too different from general policies (and among them); and **d)**

we no longer find that policies from upper-middle income countries tend to be different from those from higher-income countries, which is one of the notable results obtained on the full sample.

We have summarised these main points in the main text while leaving the corresponding SI section unaltered, and we hope this answers the reviewer's concerns.

Finally, concerning different approaches to investigating the robustness of the results, we considered the editor's suggestion to split the sample by geographical area, but we felt this would have implied working on too small subsamples. Moreover, we felt the article was already quite long and shifting focus from the global perspective would have pushed us in a totally different direction from our original aims. We will treasure the reviewer's and the editor's comments for future works, as well as we hope that some other researchers will use (and expand on) the dataset we produced and shared in the most imaginative ways.

Response to Referees:

We sincerely thank the referees for their valuable input and feedback, which have significantly strengthened our manuscript in many ways. We greatly appreciate the time and effort devoted to reviewing our work, and the resulting quality is a testament to their valuable contributions. Please find below our responses to referee 1 and a thank-you note for referee 3.

Referee 1

1. The authors have now reported the study design.

Thank you, yes.

2. In the body of the paper (study design and limitations), the authors state “We systematically identified and analysed promotion criteria”. The authors do not provide additional details here and refer readers to - (see Section 1.1 of the supplementary material). How the promotion criteria were systematically identified is critical methodological information. This information needs to be included in the body of the paper.

Thank you for highlighting this. We have now clarified the text in line 74-76: *“We systematically identified and analysed selected promotion criteria by capturing their presence or absence in promotion policies and comparing differences and similarities across disciplines, fields, tracks, types of institutions, and countries, considering their socioeconomic contexts.”* In the interest of space, we make reference to the thoroughly described methods in the SI.

3. Early in the “Study Design” section the authors need to explicitly state they did not have a formal protocol prior to beginning the project although various components were drafted. The authors have also included “Limitations” as part of the title of this section (Study Design and limitations”. I think ‘limitations” is ill advised here.

We now state *“Rather than following a predefined protocol, we developed our methodology through an initial pilot study, which then evolved into a comprehensive framework”* in lines 77-80. We have also separated the section of Limitations to be independent in Lines 351-357 -After the discussion as previously suggested by the reviewer.

4. From section 1.1 (supplementary material), I read “First, we limited our data sourcing to promotion policies for full professors, as this role was the only one universally present in virtually all documents describing policies (see Section 1.2). We retained only documents that offer academic paths or include posts covering both research and teaching, while acknowledging the fact that some institutions offer different career tracks that may not be comparable. Additionally, after conducting a round of interviews with representatives from various institutions to explore data sourcing options, the team concluded that a strict and consistent interpretation of the data was essential, limiting our study to accounting for the presence or absence of specific assessment criteria. The definition of our data units and the adopted procedures for their inclusion and categorisation are detailed in Section 1.2.3...”. From reading this section, I interpret the inclusion criteria to be the yellow highlighted text above. The exclusion criteria are ?? Please include a sentence or two of the eligibility criteria (inclusion and exclusion criteria) in the

methodology of the paper. Readers should not have to dig around supplementary materials to find such critical information (eligibility criteria).

We have included this information in section 2.2. of the Methodology (L680-687): *“ Data Cleaning (inclusion criteria). The documents varied significantly in structure and level of detail (see SI sect. 1.5, Table E2). We included only documents with clear, measurable criteria, excluding duplicates such as national policies reported by multiple institutions. Documents relying on vague terms like “excellence in research,” “leadership,” or “international visibility” without specific, measurable achievements were excluded. Additionally, only documents addressing the role of full Professor and covering both research and teaching responsibilities were considered. We focused on policies related to the sciences and humanities, excluding Arts and Creative Works due to substantial differences in outputs and achievements.”*

5. Similarly, the authors state *““The first sampling wave took place between 2016 and 2018, during which 46 policies were collected and then used to inform our methodological framework. A second wave of sampling occurred from 2018 to 2021, yielding 159 policies from 56 countries, which were analysed to produce the first version of this study, available as a pre-print [2].”*. This information (about sampling in yellow highlight) needs to be in the body of the paper as it is critical information about the methodology. *We have included this information to Lines 89-94: “Between May 2016 and November 2023 we drew on the GYA membership and alumni network to collect documents outlining promotion policies, including criteria and procedures. In a study pilot phase (2016–2018), we sourced 46 policies to inform our methodological framework. After, in a first data collection phase (2018–2021) we built a dataset of 196 policies from 55 countries. In 2023, we updated the data for all policies collected, and broadened the scope of our effort to include policies from underrepresented Global South regions, adding further 440 policies”*

Referee #3 (Remarks to the Author):

The additional details regarding protocol, analysis, and results that were added to the manuscript and supplementary information have increased the clarity and transparency of the reported research.

We thank the referee for their comment and the valuable feedback provided. We thank them for their time devoted to this study

Precis

The 21 authors examine research assessment (promotion to full professor) in what they describe as a “truly global outlook” perspective. The authors examined 159 policies from academic institutions and 37 policies from government between 2016 and 2021. Documents from 55 countries were included. Each document was assessed on 18 categories and 11 sub-categories. The authors note several comparisons between the global ‘north’ and south’. The authors also note the popularity of quantitative assessments. In the discussion, the authors note five criteria that should be considered for developing researcher assessment policies.

Assessment

The topic the authors address is obviously an important one and there is plenty of push for change, lots of proposals to reimagine researcher assessment but limited action as of yet.

1. I do find some irony in the fact that this paper is limited to polices/documents for researcher assessments to full professorship carried out by the Global Young Academy. Wouldn't early career researchers be most interested in polices/assessment for hiring and early career progression (e.g., to associate professor)?
2. My main concerns about the paper are about the methodology.
3. Did the authors work from a written protocol?
4. How were the documents accessed? What processes were used. For example, our team have just completed identifying hundreds of documents that mention policies/templates related to data sharing. This will likely require a few sentences describing the process in the methods section of our paper.
5. I think Figure 1 (Methods outline) insufficient. It does not allow interested readers to replicate the methods. Please provide more detailed methods. The results are only trustworthy if the methods are rigorous, comprehensive, and transparent.
6. How were the documents sampled. Where the documents taken from countries the Global Young Academy member was living in, some sort of purposive sampling and something else. Clarity of this issue is important.
7. The sampling time – May 2016 to July 2021. Were some documents sampled in 2016 and others in 2021? A document sampled in 2016 might be out of date in 2023. This is particularly relevant given the prominence in researcher assessment over the last 5 years. Did the authors update their sampling to be current as of the middle of 2023?
8. Be more precise throughout the document. For example, line 141, the authors state “About 60% of these policies”. This is a little vague.

9. As best as I can tell the authors developed 18 categories and 11 sub-categories for mapping against the documents/policies. As a reader it would be invaluable to know how the categories and sub-categories were developed? Are they evidence-based?
10. Where any of the categories related to open science or equity, diversity, and inclusiveness? I'm I correct in assuming neither domains were examined?
11. It looks like each document was reviewed by members of the author team. Did two authors review each document and come to some form of agreement? If so, how was this done and can the authors report measures of agreement?
12. The authors outline 5 strategies. On line 302, the authors state "...should be valuable by institutions and researchers ...". Do the authors mean 'valued'?
13. The authors do not appear to propose any option for 'promotion criteria are not'.
14. The authors appear surprised by the use of quantitative metrics used by lower-income countries. Might this be because the resources (fiscal and qualitative expertise) required to implement more qualitative assessments are not available? Even in my higher income country, our university has limited expertise in the qualitative methods experts.
15. A limitation section would be valuable for readers.

RESPONSE TO REVIEWERS

Editor's comment

Your manuscript, "A global assessment of academic promotion criteria: What really counts?", has now been seen by 3 referees, whose comments are attached below. While they find your work of potential interest, as do we, they have raised important concerns that in our view need to be addressed before we can consider publication in Nature. More specifically, they are concerned that the sampling may be biased (as you only include institutions that are GYA members) and that you neither look at different career tracks or provide sufficient methodological details to assess the strength of the results.

Should you be able to show that the GYA member institutions do not reflect a biased sample, include policies for different career tracks, and have the power to disentangle the effect of policies (national vs academic) from location, we would be happy to consider a revised manuscript (unless something similar has been accepted at Nature or appeared elsewhere in the meantime).

We are thankful for the opportunity to present this revised manuscript and for the constructive feedback provided by the reviewers and editor. We have given your advice thorough consideration and implemented your valuable suggestions in order to strengthen our work. We believe that you will find that this revised version of the manuscript adequately addresses all of your requests and resolves any concerns regarding our sample and methodological approach. Among the changes made, are:

- 1) We undertook a major data collection effort oriented at updating and extending the global representation of our sample, resulting in the incorporation of 419 policies into our study. Our sample now includes research institutions from 121 of the 195 countries worldwide, mitigating any potential biases in our sample selection and achieving a wider global representation. This extended sample enabled us to :
 - Run analyses of the power for the hypothesis tests reported. Since the global population of assessment policies is unknown, these tests are based on the observed mean values and standard deviations in our sample. The results are indicative of very high levels of power (above 95% in all relevant cases), please refer to Supplementary Information (SI) Sect. 2.3.
 - Significantly strengthened our analysis, providing a robust foundation to substantiate our claims. (Please refer to sect. 2.1 and 2.2 in the SI for detailed insights)
 - Conduct a multivariate regression analysis to disentangle the combined effects of multiple independent variables, allowing for a more comprehensive understanding of underlying dynamics.(Please see section 2.4 of the SI, L 226 – 254 and Figure 5 of the results).
 - Evaluate the influence of different career tracks in the choice of criteria evaluated. As seen in Fig 5 of results and SI sections 1.2.3. and Table S1.

- Ensured that the large majority (>85%) of the policies included in the study are the most recent versions and are still applicable. We have integrated this information into our dataset. See L: 40-47 in the SI sect 1.1.
 - We sourced relevant information suggested by reviewers, such as the evaluation of interdisciplinarity and ethics as assessment criteria, and whether a policy is exclusive for the role of full professor (see SI Table S2 in sect. 1.3.1, and all the Figures in the results).
- 2) In response to the observations made about the weaknesses in our methodology, we conducted a thorough review of our processes of data coding, implemented controls, and gathered additional data, which has notably improved the transparency, reproducibility and reusability of the data in our study. For example:
- Each criterion in the dataset now includes the evidence (in an adjacent cell) indicating the rationale for considering its presence, providing future users with a better understanding of the basis for scoring.
 - Policies underwent review by a minimum of three authors, as stipulated and detailed in the database (under the reviewer columns) .
 - The dataset provides URLs for over 80% of the policies collected, which were available online, alongside document attributes such as sourcing date, implementation date, and the latest year the policy was known to be active (see L: 41-48 in SI sect1.1).
 - We are sharing annotated code developed for our analyses, as well as the raw outputs of these analyses.
- 3) The methodology underwent a thorough rewrite, encompassing all details of each step of our processes.
- 4) We made significant revisions to our data analysis methods to enhance their representativeness, reproducibility, and impartiality (more in Supplementary Information, section 1.7). Among the key changes implemented:
- We introduced an intra institutional weighting system to ensure that each institution, regardless of the number of policies it contains, is assigned a value of 1 (see SI section 1.6).
 - We incorporated a global weighting system, calculated based on the number of researchers in full-time equivalents residing in each country included in the study. This approach enables us to refer our results to the population of potential job candidates affected (see SI section 1.6).
 - We minimized interventions in data categorization and stratification, which we thoroughly describe in the SI section 1.5 to maintain transparency and consistency throughout the analysis process.

Referee #1 (Remarks to the Author):

1. I do find some irony in the fact that this paper is limited to policies/documents for researcher assessments to full professorship carried out by the Global Young Academy. Wouldn't early career researchers be most interested in policy/assessment for hiring and early career progression (e.g., to associate professor)?

We thank the reviewer for raising this point. We now include the rationale for this decision in paragraph 1 of the data collection and analysis (Lines 84 – 94). In summary, we adopted this profile as it provides us with a basic role that can be meaningfully compared across various countries. While defining this study, we identified that the most senior academic position is more clearly defined and relatively similar internationally, whereas the entry- and mid-level positions are more differentiated across countries and sometimes even across institutions in a single country, with career pathways differing substantially. For example, “associate professor” is predominantly a US term and e.g. in Germany such a career profile does not exist. This topic was also addressed by Reviewer #2, albeit from a different perspective, underscoring the high complexity of comparing academic career progression. Moreover, the aspiration for many if not all ECRs is to one day ascend to the highest ranks of the profession, and this affects their incentives and potentially their behaviour already at the early stages of their career.

2. My main concerns about the paper are about the methodology.

We thank the reviewer for bringing this concern to our attention. This has been well noted, and we have extensively revised the methods to make them comprehensive. Please refer to the revised methodology described in the Supplementary Information.

3. Did the authors work from a written protocol?

Thank you for raising this point. **Given the novelty of the project and** the diversity of our sample, there were no studies or protocols that could meet our needs, and thus we empirically developed and tested different approaches at the outset of the study. Through this experience, we were able to standardise modes of practice in terms of data collection and tabulation. Similarly, we used the most statistically sound approaches in terms of data analysis. This is now thoroughly explained in our fundamentally rewritten methodology in Supplementary Information (SI) sect 1.

4. How were the documents accessed? What processes were used. For example, our team has just completed identifying hundreds of documents that mention policies/templates related to data sharing. This will likely require a few sentences describing the process in the methods section of our paper.

Thank you for bringing this to our attention. We have now included a section providing all these details in section 1.1 of the methodology (L34 - 40), including the following: “. We consider each distinct set of assessment criteria applied for a promotion process to constitute a “policy” for promotion. Some institutions or agencies distinguish different career tracks or academic disciplines, and therefore there can be several policies within the same institution or agency. In total, our dataset covers 532 policies of which 427 policies (80%) were sourced from public websites, 105 policies (20%) were sourced through the co-authors’ networks and 6 policies (1%) were obtained via an official request made through the GYA office. We share the respective URLs of policies found online in our dataset. .”

5. I think Figure 1 (Methods outline) insufficient. It does not allow interested readers to replicate the methods. Please provide more detailed methods. The results are only trustworthy if the methods are rigorous, comprehensive, and transparent.

Thank you for this remark. We have removed this figure and provided more substantial detail in the methodology section. Please refer to the Supplementary Information Sect 1.

6. How were the documents sampled. Where the documents taken from countries the Global Young Academy member was living in, some sort of purposive sampling and something else. Clarity of this issue is important.

Thank you for pointing this out, we have now included a detailed section of our snowballing sampling approach, in Section 1.1 of the Supplementary Information L10-19: *"We leveraged on members and alumni of the Global Young Academy (GYA) to source documents not only from their own institutions, but in a way of snowball sampling also from other institutions they know or have a relation with, or by involving their academic network.."*. While members of the Global Young Academy (GYA) played a crucial role in sourcing the documents (knowledge of regional infrastructure and languages), those are not restricted to their own institutions. We are happy to report that we have greatly increased our coverage across the globe with a further sourcing effort, we have updated all policy documents that were in the original dataset, and we applied post-sampling weighting, L 28-31: *"..Our sampling specifically aimed at maximising the reach and scope, so that our sample is as wide and diversified as possible, in terms of countries covered, areas of the world, and economic status. Consequently, as highlighted in the main text, we obtained the largest data set, to our knowledge, of promotion criteria globally to date."*.

7. The sampling time – May 2016 to July 2021. Were some documents sampled in 2016 and others in 2021? A document sampled in 2016 might be out of date in 2023. This is particularly relevant given the prominence in researcher assessment over the last 5 years. Did the authors update their sampling to be current as of the middle of 2023?

Following the reviewers advice in this matter we performed an extensive round of review of our sampling, which concluded in the update of all policies. This is now expanded in Section 1.1, L 40-47: *"...drove a second round of sourcing in 2023, to check for updates or new policies. From this exercise, we sourced 440 policies, of which 300 (56%) were implemented after 2020, 148 (28%) after 2015, 62 (12%), before 2015 and for 22 (4%) policies this information was not disclosed."* Further, for the small fraction of policies that we could not confirm whether they are still applicable, we wish to point out that we have observed that policy and institutional change is typically rather slow, and while we agree with the reviewer, that change in the practices at the global level is moving forward, this does not necessarily apply at the level of the single institutions or even single countries, which are the units of our analysis. With this in mind, we believe that our dataset is up to date and that to our knowledge, most of the policies are still applicable.

8. Be more precise throughout the document. For example, line 141, the authors state "About 60% of these policies". This is a little vague.

We thank the reviewer for highlighting this. We now have rephrased this to: *"Most of these institutions and agencies (73%) are based"* in Line 100 of the manuscript. We took notice and have refrained from vague language. However, we have refrained from providing decimal points in the findings to avoid confusing or possibly misleading the reader, as the hypothesis tests show that small differences in our results are not statistically significant. The differences that we comment upon are typically of a different

order of magnitude (10% or more) and indeed, our powers analysis shown in section 2.3 of the SI confirm that at this order of magnitude our tests have sufficient power.

9. As best as I can tell the authors developed 18 categories and 11 sub-categories for mapping against the documents/policies. As a reader it would be invaluable to know how the categories and sub-categories were developed? Are they evidence-based?

Thank you for highlighting this. We have now included a section in the SI (sections: “1.3.1. Definition of criteria” and “1.3.2. Categorisation of criteria”) expanding this point, see lines 156 -180 and tables S3 and S4, For example, L169-171: *“At the onset of this project, we performed an empirical exercise to understand and develop a method to quantitatively capture the qualitative data embedded in a policy. Through this exercise, we identified common keywords or clusters of keywords relating to similar criteria”*. L182-171: *“We found that the majority of policies are explicitly structured around the three broad categories of “research”, “teaching”, and “service”... ”*

10. Where any of the categories related to open science or equity, diversity, and inclusiveness? I’m I correct in assuming neither domains were examined?

While we acknowledge the importance of these topics, there were several limitations that prevented us from including these categories in our study. This is primarily due to the nature of these topics, which predominantly relate to the process of hiring or promotion (e.g. non-discrimination etc.), and to a smaller extent to the criteria evaluating the single candidates or their work. Since our analysis is based solely on policy criteria, we wouldn’t be able to comprehensively capture this information.

Furthermore the criteria evaluating these subjects are highly complex and globally inhomogeneous. We concluded that performing statistical analysis on criteria that have very different meanings across different national or regional policy frameworks would likely be more confusing than helpful and therefore of little value. We have now clarified this stance and rationale in the first paragraph of the Data Collection and analysis section, lines 123-130: *“Our analysis is solely based on the presence or absence of these criteria, we are not considering the assessment process. Specifically, we did not consider the role or weight of criteria, nor did we capture how assessors interpret the policies, and how an assessment panel ultimately arrives at its decision, potentially taking into account further criteria that are not explicitly mentioned in the policies”* .

To expand on this, properly capturing the relation between open science factors in promotion criteria with national and regional open science policy frameworks, benefits to various actors, and a globally equitable open science system warrants its own study, and goes much beyond capturing key commonalities and differences and arising implications for researchers, research managers, and national governments, as discussed here. One of the authors of this study explicitly discussed global policy divergence and misalignment with global goals (DOI: 10.5334/dsj-2022-001). Open Science must not be reduced to a box-ticking exercise to meet policy requirements. Similarly, equity, diversity, and inclusiveness are crucial goals of science policy and we are aware that research evaluation can impact them (for example another author has personally worked on this topic: <https://doi.org/10.1016/j.respol.2019.103820>). However, more than any other criterion covered in our study, these policy goals take different meanings in various national contexts, sometimes in response to

specific shortcomings and/or meeting national legal requirements, with incompatibilities across countries. We elaborate on this in the SI.

11. It looks like each document was reviewed by members of the author team. Did two authors review each document and come to some form of agreement? If so, how was this done and can the authors report measures of agreement?

Thank you for bringing this to our attention, we have expanded on the process of data capture in section 1.5 of the SI L 266-281. In brief, we adopted several measures to ensure consistency on the interpretation, including the provision of the evidence that led to a scoring decision and a thorough procedure involving a two-stage verification process in which each document was separately reviewed by at least two co-authors. Whenever disagreements arose, a 3rd author or more (depending on the level of disagreement) were involved to review the text and reach an agreement. In those cases, we documented the incident by including the decision into the definition of the criteria (Shown in Table S2), so that it was available to other authors working parallelly for future reference.

12. The authors outline 5 strategies. On line 302, the authors state "...should be valuable by institutions and researchers ...". Do the authors mean 'valued'?

Thank you for spotting this, we have changed this and the sentence now reads: "*We anticipate that our findings will prove valuable to both researchers and research managers for understanding career options and opportunities, and provide guidance on how to engage in building a strong research ecosystem that embraces diversity amongst responsible actors who contribute efficiently with their various strengths.*"

13. The authors do not appear to propose any option for 'promotion criteria are not'.

We apologise to the reviewer, but we failed to understand what the concrete suggestion is. We asked several members of the team and could not reach a sound consensus interpretation. Therefore, we felt unable to address this point.

14. The authors appear surprised by the use of quantitative metrics used by lower-income countries. Might this be because the resources (fiscal and qualitative expertise) required to implement more qualitative assessments are not available? Even in my higher income country, our university has limited expertise in the qualitative methods experts.

Thank you, as it turns out, some of the authors were surprised and others less so. In this revised manuscript, we have a substantially larger dataset that allowed us to run multivariate analysis. This now shows that it is not generically the global south to rely more on metrics, but rather specific cases such as upper-middle income countries. Our original claim has now been specified.

However, it remains true that, descriptively, metrics are more frequently adopted in the Global South than in the Global North. While we agree with the reviewer's suggestion about resources being a potential underlying reason, we did not include this in the discussion as we didn't find tangible evidence supporting this hypothesis and we found alternative explanations, such as that introduced by the "Scoping Group report" of the International Academies Partnership, which suggests that institutions in these countries may have "learned the game" and are actively trying to climb the rankings, although

they perceive that higher-income countries are currently changing the rules of the game by moving towards qualitative criteria. We hope that this study provides a basis by which future research can disentangle this See: <https://www.interacademies.org/publication/future-research-evaluation-synthesis-current-debates-and-developments>).

15. A limitation section would be valuable for readers.

Thank you for the suggestion, we have now included upfront limitations of our study in the main text (first paragraph on data collection, lines 84 -94) and we have rewritten the methods (SI Section 1) to contextualise our methodology, providing the rationale for approaches taken and highlighting any limitations in an upfront manner. We believe that this will be liked by the reviewer as it will enhance the clarity for the reader, preventing any misinterpretations or misguided expectations.

Referee #2 (Remarks to the Author):

Thank you for the opportunity to review the manuscript “A global assessment of academic promotion criteria: What really counts?”, which I read with great interest. The study offers an examination of promotion criteria to full professor (or equivalent) written into national and institutional policies and guidelines from 55 countries. It is, to my knowledge, the most global study of its kind and presents valuable evidence over which assessment criteria are codified into policies and guidelines. The evidence presented largely affirms what is expected from the literature and from the common understanding of promotion criteria while providing an empirical basis for those common assertions. Most notable, for me, is the evidence highlighting that the Global South is more likely to use number of publications and citation metrics than the Global North. While aligning with expectations, this study provides solid evidence on the matter.

Thank you. We appreciate your perspective, appreciation of the study and suggestions.

1. While there is much to praise from the study, there are also a handful of areas where further clarity is needed and should be provided ahead of publication. These areas are all connected to the wide range of documents that were surely collected and the similarly wide-ranging set of academic and researcher systems to which they apply. While the authors acknowledge that “the structure and style of the policy documents varied substantially,” they provide little description of this variability in relations to the dimensions analyzed. For example, were certain kinds of documents more prevalent in some countries than others?

We thank the reviewer for the constructive feedback. In response to this suggestion, we have now captured and analysed policy document features. Here, we surveyed the types of documents encountered across different policies, noted whether they were exclusively specified for the full professor role and their level of detail(see Table S5). In this way, we wish to provide a better understanding of the general structural differences that may exist between policies. Additionally, we performed an analysis to assess the influence of factors such as the geographical origin of the policy, economic background, type of institution, disciplines, and tracks on the development of highly detailed policies that include rubrics (see table S6). All of these can be found in section 1.4 of the supplementary information (SI) in lines 215-243.

2. Similarly, the collection of both institutional and national policies is a strength of the study, but the lack of analysis of where national policies were more prominent makes it difficult to disentangle the national/institutional analysis from the Global North/South analysis.

We thank the reviewer for highlighting this. Following the reviewer's advice, we expanded the description of the sub-samples by region, type of institution, and other factors in section 1.2 of the SI. Here, we now include a comprehensive table (Table S1) listing the different proportions of policies used for each analysis. Furthermore, in section 2.3 of the supplementary information, Table S12 (power analyses) we presents evidence of the robustness of our data to support our analysis. Additionally, we have included a multivariate analysis to help us disentangle the effects of confounding variables, such as those relating to the number of institutional versus national policies sourced for each sub-dataset, the geographic and socio-demographic context, and the nature of the policy (Please see Figure 5 and lines 226 -241 of the results).

While we cannot accurately determine where national or institutional policies are geographically more prominent, as this is beyond of what we could possibly achieve (now stated in L 123-130 of our data collection & analysis), including these analyses clarify the role of each independent variable in our analyses.

Furthermore, despite the limitations of our scope, we endeavoured to develop, to our knowledge, the largest dataset of up-to-date policy criteria, which we have made publicly available. We have prioritised the reusability and reproducibility of this dataset, so that it facilitates future research endeavours. This data is readily available for studies aiming to unravel the connections between the characteristics of a country or institution and the criteria adopted. We hope that this data is valuable to support these and other studies.

3. This relates to a broader issue of the diversity of systems that exist for researchers around the world (how were equivalencies of full professor found for systems around the world?). Was there any consideration for policies from countries where researchers hold dual affiliations (one with the national system and another within their institution)? While the manuscript acknowledges that such differences exist (similarly with different career tracks), the study design, analysis, or discussion fails to take them into account.

We thank the reviewer for bringing this to our attention. We now include the text clarifying these very relevant points. Text regarding the full professorship equivalences can be found in Lines 84-94 of the data collection and analysis section, where we specify: *"..career progression can take various paths, and types of posts that exist in one country might be unknown in others. Mapping or analysing the variety of career paths would be beyond the scope that we can reasonably cover. We therefore specifically decided to focus on policies for promotion to (full) "Professor", i.e. the most senior academic role, which has a profile that is omnipresent and can be meaningfully compared across countries"*.

Furthermore, in response to the reviewer's request, we conducted a survey of the policies to gather information regarding whether the position was exclusive, or if the policy was specific for the role of professor. This criterion was included in our analyses and presented in Fig 5 and described in L 223-226 of Sect 1.4 of the SI, and tables S1, S5 and S6. In brief, 85% of the policies specified that the role was

full-time and exclusive (prospective winning candidates cannot hold dual affiliation), while 79% specified that the policy had criteria exclusive to the role of full professor.

We also addressed the issue of "dual affiliations" by restricting the inclusion of policies to academic institutions. This decision was made on the premise that only policies from academic institutions applied in this context. Because, despite the fact that Professors may hold affiliations beyond academic institutions, the title is usually associated with a position at an academic institution, with only the academic institution having the authority to grant a "professorship".

Finally, we noted that in some countries, institutions follow national policies (e.g. South Africa or the Philippines), while defining further criteria in addition. In that case, we consider the national policy to be incorporated into the institutional policy (See section 1.2.1 of SI).

4. I also found no mention of how many of the documents found could be cleanly mapped to particular career stages/tracks. In my experience, there is a mix of documents that specify career stages/tracks and those that do not.

We thank the reviewer for highlighting this. We have now provided this information in section 1.2.3 of the SI and the breakdown is in Table S1.

5. Again, similarly, there is little detail provided on how much overlap there is between institutional and national level policies, or how much overlap there is between institutional policies (i.e., multiple documents from the same institution, potentially for different disciplines or different academic units). Was having a single document from a specific academic unit enough to include that country in the country-level analysis?

Thank you for bringing this to our attention. In response to the reviewers questions, we have:

1. Provided details of our sub samples in SI section 1.2 and Table S1, where we list information, such as L109-114: *"Some institutional policies were found to explicitly refer to national policies. In several cases, a national policy outlines basic requirements for promotion, while institutions can add additional criteria.. ."* In this section we also define more precisely that the unit of analysis in our study is policies (that is, a set of criteria applying to candidates) and not documents. We do not refer to documents because some policies are spelled out across multiple documents, some documents contain instead multiple policies, and as the reviewer notices, in some cases there are repetitions of overlap (for example, a national policy being simply reiterated by single institutions). For more details, please our enlarged discussion in section 1.2.1 of SI.
2. Implemented a post-sampling weighting system (See SI sect 1.6) where L 288-291: *"..all policies within the same institution or agency have the same weight, all institutions or agencies within the same country have the same weight, and each country has a weight proportional to the number of researchers active within that country."*
3. Further, we have provided more clear limitations about the scope of our analysis in the main text (Lines 123-135) where we state that any analysis at the country level or below, is beyond the scope of our project, and we clarify that the data set is not a representative sample at the country level.

6. In short, the study is lacking a more complete explanation of how the differences in academic careers and the diversity of documents were handled. Both of these elements should be expanded upon in Methods and in the Discussion. The differences in academic careers across countries, especially the extent to which academic career progressions can (and cannot!) be compared, should be addressed in the literature review and again in Point 1 of the discussion (noting, for example, that researcher mobility varies greatly by country). (For what it's worth, it was this complexity that held us back from analysing documents globally and made us decide to focus on the US and Canada, where the Review, Tenure, and Promotion systems are largely comparable).

We thank the reviewer for this suggestion, we hope that section 1.2 and 1.4 of the SI now include the detailed information of methodology requested. We also emphasised our limitations in L 84-94 of our main text and as requested, addressed this point in the discussion (see lines 266-271): *"The observed variations in adopted criteria across policies moreover show that institutions have substantial freedom of choice rather than having to assess researchers in a predetermined way. This gives room to both the diversity of institutions and the diversity of career paths of researchers. In the absence of a uniform research assessment system, institutions can adapt assessment criteria to their needs..."*

Furthermore, to address concerns of the influence of different tracks, we purposely increased the size of our samples to have the statistical power to include the influence of career tracks in our analysis. This is now shown in Figure 5 of the results, and outlined in L 220, 233.

7. For the most part, I don't think much of the above will affect the result of the analysis, but it would add some *detail* and present some *limitations and caveats* that should be addressed. The one area where, I suspect, it might matter the most is in the disciplinary analysis. Where those disciplinary documents are sourced from is likely to matter and some checks should be done to describe the geographic representation of that subset of documents. (Any further detail should be considered in Point 4 of the discussion).

We agree with the reviewer, and we now provide the information regarding the origin of policies by disciplines in Table S1 of the SI. We have also stressed our limitations and caveats in the manuscript and in the SI, such as that presented in lines 84 -94 of the data collection and analysis, .

Lines 123-130: *"Our analysis is solely based on the presence or absence of these criteria, we are not considering the assessment process. Specifically, we did not consider the role or weight of criteria, nor did we capture how assessors interpret the policies, and how an assessment panel ultimately arrives at its decision..."*

Lines 131-135: *"In order to obtain a global picture that reflects the population of researchers, we used post-sampling weights, so that each policy for the same institution or agency has the same weight, each institution or agency within a country has the same weight, and each country has a weight proportional to the number of researchers active there."*

A few other comments to consider:

8. The analysis of “outputs” lacks detail on the types of outputs that are being solicited. Are only traditional outputs considered? In our own work (Alperin et al., 2022), we found 127 different types of outputs listed. While I’m not suggesting authors use this level of categorization, it might be helpful to note which types of outputs were included.

Given the wide diversity of policies and our aim to provide a global understanding of the evaluated criteria, we focused our study on capturing the benchmarks commonly used to assess outcomes, which predominantly revolve around publications. The only other distinction we made is between publications and registered intellectual property (such as patents), which are also included in a significant subset of the policies. We have now explained this in supplementary information section 1.3.1 L 169-178 as follows: *“At the onset of this project, we performed an empirical exercise to understand and develop a method to quantitatively capture the qualitative data embedded in a policy. Through this exercise, we identified common keywords or clusters of keywords relating to similar criteria.... Furthermore, while there may be various outputs listed in different policies, we focused our study on those that were consistently mentioned across the policies. These primarily refer to “publications” or “intellectual property.”*

9. The estimate of the number of researchers covered by the study seems inflated. If documents were selected for those that apply for promotion to (full) professor rank, then using the more generic “researcher per million” value as an approximation is not accurate. This speaks to the point raised above about the need for further details and discussion on career stages.

Thank you. We have dropped these statements in this context and used the number of researchers in full time equivalents for each country in our sample to apply a global weight, as outlined in lines 131-135 and sect. 1.6 of the SI. The main assumption behind our work is that the number of candidates for full professorship in a country is roughly proportional to the number of FTE researchers in the country. But as a further note of caution, we now consistently refer to “potential candidates”.

10. I did not find much value in the flow described in Figure 1 (this flow is described in the text and is sufficiently obvious as to not warrant a visualization). The map is more helpful, although perhaps it could be coupled with further details on number of documents by type, as requested above.

As suggested by the reviewer, we have dropped the figure and replaced it with a map composite illustrating the number of researchers per country included in the sample and the number of policies and institutions sampled in each country.

11. Section 4 of discussion: possible typo: “Communalities” à “Commonalities”?

Indeed, thank you for spotting this. We have corrected it.

12. I could see an argument made that the latter half of the discussion (after the five summary points) is not sufficiently rooted in the study results or the literature. Perhaps because I find myself agreeing with the perspective provided, I found the section to be a helpful conclusion to the article, but I would not be opposed for further linkages to the study findings and to previous literature.

Thank you for this suggestion, we took the opportunity to add a few points that explore further linkages with the literature and how our study findings compare with it. For example see those included in lines 376 – 383.

13. Despite the questions and comments raised, I believe this to be a valuable study. I find the factor analysis to be a useful way of adding some depth to binary indicators and the conclusions drawn to be reasonably drawn from that analysis. A study of this nature will always have limitations owing to the impossibility of neatly capturing the global diversity in research systems and cultures. I appreciate the ambition of the study and believe that the scholarly community will benefit from being presented with this global comparative view, despite its minor shortcomings.

Thank you. We appreciate your thoughts and feedback. We have clarified our limitations in this regard in lines 84-94.

Referee #3 (Remarks to the Author):

The most significant contribution of this study is the global nature of the analysis allowing for comparisons regarding review for academic promotion to be made between Global North vs. South and between countries with different income levels. Similar analyses have been conducted in the past but have not comprehensively analysed policy/procedure from across the globe. Also interesting are comparisons between institutional level and national policies governing assessment of research outcomes. It is interesting for researchers to gain understanding about what is valued in different regions, countries, and levels of administration. The insights gained have potential to guide efforts toward creating greater consistency in academic procedures globally, and may inspire leaders to apply approaches different from those commonly employed in their own region.

Thank you. We value and appreciate your perspective.

1. It is helpful that the authors displayed the number of countries from which the documents were collected in Figure 1. However, it would be more helpful to display the number of documents collected from each region (e.g., a visual based on the data used to generate table S1) to understand the composition of the dataset better. This concern applies across all data presented - n (number of documents) should be provided for each bar/category in each figure/table, or in legends/captions, as appropriate.

Thank you, following the reviewer's advice, we drafted maps that have overlapping information of researchers in a country and the number of policies and institutions sampled in each country. Additionally, we now include a very detailed breakdown of our subsamples in table S1 and also include further information about the documents in section 1.4.3, tables S5 and S6.

2. A potential weakness is the crowd-sourced nature of the collected documents which may be biased by the membership of the GYA and doesn't necessarily include all countries with eligible academic

institutions. Another concern is the lack of stratification of the sample - some regions/income levels are likely overrepresented, and others underrepresented. This limits the ability to conclude about the truly global state of academic promotion practices.

Thank you for this feedback, we have addressed the problem of potential underreach by more than doubling our sample size, which now covers 121 countries (as compared to 55 countries before). However, it is true as the reviewer suggests, that the data set does not include every country in the world with at least one academic institution. Our sample contains the absolute majority of countries in the world, including all the most populous and/or research-intensive, and to our knowledge, it is the largest and most comprehensive data set to date.

We also provide a clearer description of our sampling strategy and GYA members involvement in lines 11-20 of section 1.1 of the SI, clarifying the sampling was not limited to the countries where GYA members reside: *“We thus leveraged on members and alumni of the Global Young Academy (GYA) to source documents not only from their own institutions, but in a way of snowball sampling also from other institutions they know or have a relation with, or by involving their academic network... ”*

The detail provided now clarifies that our snowball sampling with GYA members (from 5 continents) did not limit our extent, but instead expanded it, as it provided key knowledge of regional (continental and subcontinental) infrastructures, connections, and languages that were crucial in expanding our sample. Furthermore, in addressing concerns about potential bias in our sample, we acknowledge that our dataset is not a "random" sample of policies, but we strongly emphasize that our sample is "unbiased" for the purposes of our study, in the sense that the inclusion of a policy was never influenced by the criteria contained within the policy itself. We include this in lines 21-27 of the SI: *“While the only possibility in practice, we do not regard this snowball sampling as engendering any specific bias. Specifically, neither the composition of the group of people sourcing the policy documents nor any filtering of documents was informed by the promotion criteria that are the object of our study. We included each and every sourced document that was sufficiently clear and comprehensive for us to identify the absence or presence of specific criteria (see Sects. 1.2 and 1.3). While our set of policy documents does not constitute a random sample, it can reasonably be considered as an unbiased one for the purposes of this study.”*

Finally, we have introduced a weighting system that allows us to adjust for the potential impact and influence that a policy has, considering the population it may affect. As detailed in lines 292-299 of section 1.6 of the SI: *“Consequently, weighted averages or proportions in our data set become indicative of the global fraction of researchers potentially affected, provided that averaged over many countries the following assumptions hold: (1) the share of potential candidates for full professorship is proportional to the share of total researchers; (2) the policies set by every institution or agency within a country cover a comparable number of researchers (that is, a comparable number of potential candidates are subject to them); and (3) policies within an institution or agency cover comparable numbers of researchers.”*

3. Can the authors further clarify the implications of stratification by career tracks in the method section? Perhaps examples would be helpful.

Thank you for this observation, we have now provided a detailed full section addressing this in SI section 1.2. *“Our dataset was not obtained through a stratified sampling procedure, as we collected all documents that spell out the criteria for promotion to “full professor”. For the purpose of differential analysis, we classified the policies by their country and related region, their scope (“institutional” or “national” policies), career tracks, and academic disciplines”.* For academic tracks (section 1.2.3; L135-139): *“We identified this standard academic track for each of the policies and adopted its set of criteria as the default for our analysis, thereby allowing and capturing some diversity on what is understood as the role of “professor” on standard academic track. Besides the standard academic track (research & teaching), a substantial number of policies (48%) consider a research-focused track, a teaching-focused track, or a clinical track”.*

As explained in the SI section 1.6, we consider a document to contain more than one policy, where the criteria spelled out differ by discipline or track. To normalize the influence of policies across institutions, we implemented a weighting system where each institution has weight equal to 1, regardless of the number of policies that we codes from its documents.

4. What types of national documents were used? The description in the method section merely states the use of published documents - can examples be given?

Thank you for highlighting this, we took the opportunity to list concrete examples of government agencies that have issued national policies. We agree that it is insightful to see what kind of agencies are involved in this. We list these in section 1.2.1. For example: L96: *“i. Government agencies under a universities, education, or higher education portfolio: National Agency for Quality Assessment and Accreditation of Spain (ANECA), National Universities Commission (Nigeria).”* L102. *“ii. Government agencies under a research, innovation, science, and/or technology portfolio: National Research Foundation (NRF, South Africa), Department of Scientific Research and Technological Development....”* L107: *“iii. University Grants Commission (Sri Lanka), Department of Budget and Management (DBM, The Philippines).”* Furthermore, we share the URL of all the publicly available policies and the codes we used to classified them in our dataset.

5. Document analysis: From what method/process did the mapped 18 categories/4 groups originate? Interesting that teaching is represented as a singular category whereas research and service are broken down into numerous detailed sub-categories - is there a reason for grouping all teaching themes (and even placing teaching within service)? Teaching often comprises the majority of the academic workload, despite its value usually weighted less in academic promotion documents.

Thank you for raising this point. We have followed the suggestion and included rationale for the categorisation and the grouping of teaching activities. We now provide a detailed section of the definition of criteria in our section 1.3.1, lines 169-178 and explaining how they originated: *“At the onset of this project, we performed an empirical exercise to understand and develop a method to quantitatively capture the qualitative data embedded in a policy. Through this exercise, we identified common keywords or clusters of keywords relating to similar criteria. This gave us 30 distinct typical evaluation criteria that occurred consistently across a substantial number of policies”*

We also provide the rationale for the categorization in section 1.3.2. lines 182-193: *“ We found that the majority of policies are explicitly structured around the three broad categories of “research”, “teaching”, and “service”, the latter covering the contributions and impact to the wider profession and society”.* Additional information is also provided now in Tables S3 and S4.

In addition, our rationale for not subcategorize teaching can be found in section 1.3.1. L 173-178: *“We specifically considered teaching as a single criterion rather than delving into the complex details of teaching assessment, because it was found to be highly variable across policies and difficult to categorise with our current methods and scope.”* The conflation of teaching and service arose from the criterion of “mentoring”, which was seen as overlapping. We renamed the category “Teaching & Service”. Incidentally, in the factor analysis the criterion of whether the policy considers teaching, does not enter with a large loading in any factor and retains a high uniqueness, suggesting that this variable is indeed separate from the others (and if our analysis was driven by purely empirical considerations the variable should probably be dropped). In our view, this would call for a separate, in depth analysis of the inclusion of teaching among the evaluation criteria (which, as the reviewer suggests, is unfortunately often discounted), but such an investigation is beyond the scope of the present work.

Thoughts on Conclusions:

6. Point 1: It seems that differences in institutional requirements may also produce situations in which researcher mobility is limited (i.e., research priorities and valued achievements at current university may not be valued at a different university).

We agree with the reviewer, and we added the following to the discussion, lines 330-335: *“but they need to be aware that institutions are not the same and therefore not every institution will be a good match.”* We also now further elaborate on this with *“Pushes to increasingly conform to standardised research profiles are detrimental to diversity of backgrounds and ideas, already adversely affected by adherence to bibliometric research evaluation [43], and hamper inter-sectoral mobility across academia, industry, government, and not-for-profit organisations. Capable and well-skilled researchers can face a-priori exclusion not only by inflexible policies, but also by norms enshrined in the minds of assessors.”.* An important fact to keep in mind is that academic institutions are not identical (which makes a case for assessment criteria to differ). In fact, a very common reason for rejecting applications for an academic post is that it is generic and not tailored to the specific institution.

7. More emphasis on the idea that evaluation committees generally have flexibility in their consideration beyond what is listed in the documentation is warranted.

We agree that this is an important point to keep in mind, and we now state this more clearly and prominently in lines 123 – 130 the text of the main document as follows: *“Our analysis is solely based on the presence or absence of these criteria, we are not considering the assessment process. Specifically, we did not consider the role or weight of criteria, nor did we capture how assessors interpret the policies, and how an assessment panel ultimately arrives at its decision, potentially taking into account further criteria that are not explicitly mentioned in the policies.”*

8. It is concerning that 44% of documents in this study did refer specifically to journal impact factor when impact factor is such a poor metric for research quality - this might be emphasized as a greater concern.

Thank you. To clarify, under the respective tabulated criterion, we cover both direct references to the Journal Impact Factor as well as references to criteria that derive from it, such as journal indexing (see Item 5 in Table S2 - Definitions of Criteria). We have relabelled this criterion in order to clarify this point. We agree that the Journal Impact Factor is a very poor metric indicator of research quality and we make emphasis of it in lines 277 -87 of our discussion, and expressed our explicit concern that upper middle-income countries have the strongest affinity with output metrics in point 2 of the discussion *“Scientometrics are most popular in upper middle-income countries”*.

However, on the bright side we think that it is important to also point to the fact that many documents do not make any (direct or indirect) reference to it (point 5 of the discussion: *“A bibliometric profile is not a key to success everywhere”*), while many people believe that it is now of universal relevance all across academia.

9. Point 2: What factors do the authors think cause higher income countries to rely less on scientometrics? Perhaps consider the increased labour required for qualitative analysis.

We much like this question, which was raised by another reviewer too, and are happy that our findings raise this kind of questions. Properly answering it beyond some speculations, however, requires further hard evidence that we do not have (yet), and we wanted to avoid making speculative statements in the discussion of our current study, given that there is some risk of drifting off and diluting the messages that directly derive from evidence. May we take this as a suggestion for our future research directions?

10. Point 3: Do the authors have any comments regarding the effects of national vs. institutional policies being out of alignment with each other in both Global North and Global South? What impact does this have on institutions and researchers?

Thank you, we actually started speculating about this point, and arrived at the conclusion that we do not understand this outcome. A related (and rather complex) question is to what extent we actually would want an “alignment” (rather than freedom as responsible actors and diversity), and we noticed quite different views about this across representatives e.g. of CoARA member organisations. We ultimately decided not to engage in speculation in our discussion session beyond the supporting evidence.

11. Point 4: Also the JIF was not created for the purpose of evaluating the impact of individual articles: <https://www.frontiersin.org/articles/10.3389/frma.2018.00002/full> #:~:text=The journal impact factor (JIF, known invention of Eugene Garfield)

Yes, we agree with the reviewer. We are aware that JIF was not meant to be used as a metric to evaluate researchers, and experts on scientometrics tend to point that out all the time. Unfortunately, we can document that many people do not listen to them. We mention the misapplication of metric indicators to the evaluation of scientific performance, and we now explicitly refer to the Journal Impact Factor as a prominent example, citing the suggested reference. See lines 51-54 of the introduction: *“experts on*

scientometrics have raised concerns that evaluation has increasingly become led by data rather than by judgement and that the misapplication of indicators to the evaluation of scientific performance has become pervasive, the Journal Impact Factor being a prominent example [20, 21]."

12. Also, this may be a good section in which to emphasize the value of regional/community engagement and acknowledge the variability this may produce in scientometrics despite increasing real-world impact of the research.

Joining this with a point made by another referee, we are now elaborating a bit further on the three-way dissonance between what counts, what is perceived to count, and what should count. Notably, this frequently forces researchers to make a tough choice between positively contributing to society and advancing their career. See discussion lines 381-383: *"On the assessment of researchers, one finds a three-way dissonance between what counts, what is perceived to count, and what should count [60, 61]. Researchers often face a tough choice between wanting to make a positive difference to society and advancing their career."*

13. Although the Global North can perhaps "afford" an inefficient research ecosystem, it is still wasteful and inequitable and should also be addressed with change. It may be worth proposing a global initiative (inclusive of North and South, and all levels of income) rather than placing responsibility on the Global South to lead the way. Perhaps such a global initiative can be flexible to allow for regional strengths to shine and meaningful collaborations across regions and income levels to flourish. The authors may make practical suggestions for bodies already working towards these ideals (as mentioned in the introduction), based on the findings of this study.

We are very supportive of global initiatives and do not want to suggest that the Global South breaks away. It is our intention to stress that the Global South is well-placed to take a leading role in such global initiatives, we have rephrased the concluding paragraph to: *"The obsession with frequently ill-suited metrics has created a research ecosystem that is highly inefficient. While many countries in the Global North can afford such inefficiency (but should not), it becomes more important for the Global South to adopt strategies for success that take a different approach. Our study challenges South-North catch-up strategies based on unsuitable performance indicators. Taking the opportunity to build research environments that better meet their purpose is not primarily a matter of funds available, but mostly about fostering a different kind of culture, as shown by Latin America's world-leading model for Open Access publishing [62]. Such kind of initiatives can provide key input to platforms like CoARA [33] that aim at building a global community. Rather than letting the Global North sort out things which the Global South then adapts to, actors from the Global South are well-suited to take the lead on global initiatives that show the way forward"* to improve the clarity of our message.

1. The authors have now reported the study design.
2. In the body of the paper (study design and limitations), the authors state “*We systematically identified and analysed promotion criteria*”. The authors do not provide additional details here and refer readers to - (see Section 1.1 of the supplementary material).
 - How the promotion criteria were systematically identified is critical methodological information. This information needs to be included in the body of the paper.
3. Early in the “Study Design” section the authors need to explicitly state they did not have a formal protocol prior to beginning the project although various components were drafted. The authors have also included “Limitations” as part of the title of this section (Study Design and limitations”. I think ‘limitations” is ill advised here.
4. From section 1.1 (supplementary material), I read “First, **we limited our data sourcing to promotion policies for full professors**, as this role was the only one universally present in virtually all documents describing policies (see Section 1.2). **We retained only documents that offer academic paths or include posts covering both research and teaching**, while acknowledging the fact that some institutions offer different career tracks that may not be comparable. Additionally, after conducting a round of interviews with representatives from various institutions to explore data sourcing options, the team concluded that a strict and consistent interpretation of the data was essential, limiting our study to accounting for the presence or absence of specific assessment criteria. The definition of our data units and the adopted procedures for their inclusion and categorisation are detailed in Section 1.2.3...”.
 - From reading this section, I interpret the inclusion criteria to be the yellow highlighted text above. The exclusion criteria are ?? Please include a sentence or two of the eligibility criteria (inclusion and exclusion criteria) in the methodology of the paper. Readers should not have to dig around supplementary materials to find such critical information (eligibility criteria).
5. Similarly, the authors state ““**The first sampling wave took place between 2016 and 2018**, during which 46 policies were collected and then used to inform our methodological framework. **A second wave of sampling occurred from 2018 to 2021**, yielding 159 policies from 56 countries, which were analysed to produce the first version of this study, available as a pre-print [2].””.
 - This information (about sampling in yellow highlight) needs to be in the body of the paper as it is critical information about the methodology.